# Beyond Adult and COMPAS:
# Fair Multi-Class Prediction via Information Projection

**Wael Alghamdi**[1*], **Hsiang Hsu**[1*], **Haewon Jeong**[1*],
**Hao Wang**[1], **P. Winston Michalak**[1], **Shahab Asoodeh**[2], **Flavio P. Calmon**[1]
[1]John A. Paulson School of Engineering and Applied Sciences, Harvard University
[2]Department of Computing and Software, McMaster University

## Abstract

We consider the problem of producing fair probabilistic classifiers for multi-class classification tasks. We formulate this problem in terms of "projecting" a pre-trained (and potentially unfair) classifier onto the set of models that satisfy target group-fairness requirements. The new, projected model is given by post-processing the outputs of the pre-trained classifier by a multiplicative factor. We provide a parallelizable, iterative algorithm for computing the projected classifier and derive both sample complexity and convergence guarantees. Comprehensive numerical comparisons with state-of-the-art benchmarks demonstrate that our approach maintains competitive performance in terms of accuracy-fairness trade-off curves, while achieving favorable runtime on large datasets. We also evaluate our method at scale on an open dataset with multiple classes, multiple intersectional groups, and over 1M samples.

## 1 Introduction

Machine learning (ML) algorithms are increasingly used to automate decisions that have significant social consequences. This trend has led to a surge of research on designing and evaluating fairness interventions that prevent discrimination in ML models. When dealing with *group fairness*, fairness interventions aim to ensure that a ML model does not discriminate against different groups determined by, for example, race, sex, and/or nationality. Extensive comparisons between discrimination control methods can be found in [BDH+18, FSV+19, WRC21]. As these studies demonstrate, there is still no "best" fairness intervention for ML, and the majority of existing approaches are tailored to either binary classification tasks, binary population groups, or both.[2] Moreover, discrimination control methods are often tested on overused datasets of modest sizes collected in either the US or Europe (e.g., UCI Adult [Lic13] and COMPAS [ALMK16]).

Most fairness interventions in ML focus on binary outcomes. In this case, the classification output is either positive or negative, and group-fairness metrics are tailored to binary decisions [HPS16]. While binary classification covers a range of ML tasks of societal importance (e.g., whether to approve a loan, whether to admit a student), there are many cases where the predicted variable is not binary. For example, in education, grading algorithms assign one out of several grades to students. In healthcare, predicted outcomes are frequently not binary (e.g., severity of disease).

We introduce a theoretically-grounded discrimination control method called `FairProjection`. This method ensures group fairness in multi-class classification for several, potentially overlapping population groups. We consider group fairness metrics that are natural multi-class extensions of their binary

---

*Equal contribution. Correspondence to: Wael Alghamdi and Flavio P. Calmon (`alghamdi@g.harvard.edu` and `flavio@seas.harvard.edu`).

[2]See Related Work and Table 1 for notable exceptions.

classification counterparts, such as statistical parity [FFM$^+$15], equalized odds [HPS16], and error rate imbalance [PRW$^+$17, Cho17]. When restricted to two predicted classes, `FairProjection` performs competitively against state-of-the-art fairness interventions tailored to binary classification tasks. `FairProjection` is model-agnostic (i.e., applicable to any model class) and scalable to datasets that are orders of magnitude larger than standard benchmarks found in the fair ML literature.

Our approach is based on an information-theoretic formulation called *information projection*. We show that this formulation is particularly well-suited for ensuring fairness in probabilistic classifiers with multi-class outputs. Given a probability distribution $P$ and a convex set of distributions $\mathcal{P}$, the goal of information projection is to find the "closest" distribution to $P$ in $\mathcal{P}$. The study of information projection can be traced back to [Csi75], which used KL-divergence to measure "distance" between distributions. Since then, information projection has been extended to other divergence measures, such as $f$-divergences [Csi95] and Rényi divergences [KS16, KS15]. Recently, [AAW$^+$20] studied how to project a probabilistic classifier, viewed as a conditional distribution, onto the set of classifiers that satisfy target group-fairness requirements. Remarkably, the projected classifier is obtained by multiplying (i.e., post-processing) the predictions of the original classifier by a factor that depends on the group-fairness constraints.

Prior work on information projection relies on a critical—and limiting—information-theoretic assumption: the underlying probability distributions are *known exactly*. This is infeasible in practical ML applications, where only a set of training examples sampled from the underlying data distribution is available. `FairProjection` fills this gap by using an efficient algorithm for computing the projected classifier with finite samples. We establish theoretical guarantees for this algorithm in terms of convergence and sample complexity.

Notably, our proposed fairness intervention is parallelizable (e.g., on a GPU). Hence, `FairProjection` scales to datasets with the number of samples comparable to the population of many US states ($> 10^6$ samples). We provide a TensorFlow [AAB$^+$15] implementation of `FairProjection` and apply it to post-process the outputs of probabilistic classifiers to ensure group fairness.

We benchmark our post-processing approach against several state-of-the-art fairness interventions selected based on the availability of reproducible code, and qualitatively compare it against many others. Our numerical results are among the most comprehensive comparisons of fairness interventions to date. We present performance results on the HSLS (High School Longitudinal Study, used in [JWC22]), Adult [Lic13], and COMPAS [ALMK16] datasets.

We also evaluate `FairProjection` on a dataset derived from open and anonymized data from Brazil's national high school exam—the *Exame Nacional do Ensino Médio* (ENEM)—with over 1 million samples. We made use of this dataset due to the need for large-scale benchmarks for evaluating fairness interventions in multi-class classification tasks. We also answer recent calls [BZZ$^+$21, DHMS21] for moving away from overused datasets such as Adult [Lic13] and COMPAS [ALMK16]. We hope that the ENEM dataset encourages researchers in the field of fair ML to test their methods within broader contexts.[3]

In summary, our main contributions are: **(i)** We introduce a post-processing fairness intervention for multi-class classification problems that can account for multiple protected groups and is scalable to large datasets; **(ii)** We derive finite-sample guarantees and convergence-rate results for our post-processing method. Importantly, `FairProjection` makes information projection practical without requiring exact knowledge of probability distributions; **(iii)** We demonstrate the favourable performance of our approach through comprehensive benchmarks against state-of-the-art fairness interventions; **(iv)** We put forth a new large-scale dataset (ENEM) for benchmarking discrimination control methods in multi-class classification tasks; this dataset may encourage researchers in fair ML to evaluate their methods beyond Adult and COMPAS.

**Related work.** We summarize key differentiating factors from prior work in Table 1 and provide a more in-depth discussion in Appendix A.2.5. The fairness interventions that are the most similar to ours are the FairScoreTransformer [WRC20, WRC21, FST] and the pre-processing method in [JN20].

---

[3] Since (to the best of our knowledge) the ENEM dataset has not been used in fair ML, we provide in Appendix A.3 a datasheet for the ENEM dataset. The data can be found at [INE20], and code for pre-processing the data and the implementation of `FairProjection` can be found at https://github.com/HsiangHsu/Fair-Projection.

| Method | Feature | | | | | | |
|---|---|---|---|---|---|---|---|
| | Multiclass | Multigroup | Scores | Curve | Parallel | Rate | Metric |
| Reductions [ABD+18] | ✗ | ✓ | ✓ | ✓ | ✗ | ✓ | SP, (M)EO |
| Reject-option [KKZ12] | ✗ | ✓ | ✗ | ✓ | ✗ | ✗ | SP, (M)EO |
| EqOdds [HPS16] | ✗ | ✓ | ✗ | ✗ | ✗ | ✓ | EO |
| LevEqOpp [CDH+19] | ✗ | ✗ | ✗ | ✗ | ✗ | ✗ | FNR |
| CalEqOdds [PRW+17] | ✗ | ✗ | ✓ | ✗ | ✗ | ✓ | MEO |
| FACT [KCT20] | ✗ | ✗ | ✗ | ✓ | ✗ | ✗ | SP, (M)EO |
| Identifying[4] [JN20] | ✓✗ | ✓ | ✓ | ✓ | ✗ | ✗ | SP, (M)EO |
| FST [WRC20, WRC21] | ✗ | ✓ | ✓ | ✓ | ✗ | ✓ | SP, (M)EO |
| Overlapping [YCK20] | ✓ | ✓ | ✓ | ✓ | ✗ | ✗ | SP, (M)EO |
| Adversarial [ZLM18] | ✓ | ✓ | N/A[5] | ✓ | ✓ | ✗ | SP, (M)EO |
| FairProjection (ours) | ✓ | ✓ | ✓ | ✓ | ✓ | ✓ | SP, (M)EO |

**Table 1:** Comparison between benchmark methods. **Multiclass/multigroup**: implementation takes datasets with multiclass/multigroup labels; **Scores**: processes raw outputs of probabilistic classifiers; **Curve**: outputs fairness-accuracy tradeoff curves (instead of a single point); **Parallel**: parallel implementation (e.g., on GPU) is available; **Rate**: convergence rate or sample complexity guarantee is proved; **Metric**: applicable fairness metric, with SP↔Statistical Parity, EO↔Equalized Odds, MEO↔Mean EO. Since FairProjection is a post-processing method, we focus our comparison on post-processing fairness intervention methods, except for Reductions [ABD+18], which is a representative in-processing method, and Adversarial [ZLM18], which we use to benchmark multi-class prediction. For comparing in-processing methods, see [LPB+21, Table 1].

The FST and [JN20] can be viewed as instantiations of FairProjection when restricted to the binary classification setting and to cross-entropy (for FST) or KL-divergence (for [JN20]) as the $f$-divergence of choice. Thus, our approach is a generalization of both methods to multiple $f$-divergences. Importantly, unlike our method, [JN20] requires retraining a classifier multiple times.

A reductions approach for fair classification was introduced in [ABD+18]. When restricted to binary classification, the benchmarks in Section 5 indicate that the reductions approach consistently achieves the most competitive fairness-accuracy trade-off compared to ours. FairProjection has two key differences from [ABD+18]: it is not restricted to binary classification tasks and does not require refitting a classifier several times over the training dataset. These are also key differentiating points from [CHKV19], which presented a meta-algorithm for fair classification that accounts for multiple constraints and groups. The reductions approach was later significantly generalized in the GroupFair method by [YCK20] to account for overlapping groups and multiple predicted classes. Unlike [YCK20], we do not require retraining classifiers.

Several other recent fairness intervention methods consider optimizing accuracy under group-fairness constraints. In [CJG+19], a "proxy-Lagrangian" formulation was proposed for incorporating non-differentiable rate constraints, including group fairness constraints. We avoid non-differentiability issues by considering the probabilities (scores) at the output of the classifier instead of thresholded decisions. In [ZVRG17], a fairness-constrained optimization was introduced that is applicable to margin-based classifiers (our approach can be used on any probabilistic classifier). In [CDPF+17] and [MW18], the fairness-accuracy trade-offs in binary classification tasks are characterized when the underlying distributions are known. A non-parity-based fairness notion was proposed in [KGZ19], called "multiaccuracy," which aims to ensure high accuracy for all subgroups even when the group information is not given in the data. We limit our analysis to parity notions of group fairness. To circumvent the non-differentiability of group-fairness constraints, approximate fairness constraints based on functionals found in information theory have been explored in [LPB+21, Rényi mutual information], [BNBR19, Rényi maximal correlation], and [PQC+19, maximum mean discrepancy]. We avoid such non-differentiability issues by casting group fairness constraints in the score domain.

---

[4][JN20] mention that their method can be applied to multi-class classification, but their reported benchmarks are only for binary classification tasks.

[5][ZLM18] is an in-processing method unlike other benchmarks in the table. It does not take a pre-trained classifier as an input.

| Fairness Criterion | Statistical parity | Equalized odds | Overall accuracy equality |
|---|---|---|---|
| **Expression** | $\left\| \dfrac{P_{\widehat{Y}\mid S=a}(c')}{P_{\widehat{Y}}(c')} - 1 \right\| \le \alpha$ | $\left\| \dfrac{P_{\widehat{Y}\mid Y=c,S=a}(c')}{P_{\widehat{Y}\mid Y=c}(c')} - 1 \right\| \le \alpha$ | $\left\| \dfrac{P(\widehat{Y}=Y \mid S=a)}{P(\widehat{Y}=Y)} - 1 \right\| \le \alpha$ |

**Table 2:** Standard multi-class group fairness criteria; one fixes $\alpha > 0$ and iterates over all $(a, c, c') \in [A] \times [C]^2$.

**Notation.** Boldface Latin letters will always refer to vectors or matrices. The entries of a vector $\boldsymbol{z}$ are denoted by $z_j$, and those of a matrix $\boldsymbol{G}$ by $G_{i,j}$. The all-1 and all-0 vectors are denoted by $\mathbf{1}$ and $\mathbf{0}$. We set $[N] \triangleq \{1, \cdots, N\}$ and $\mathbb{R}_+ \triangleq [0, \infty)$. The probability simplex over $[N]$ is denoted by $\boldsymbol{\Delta}_N \triangleq \{\boldsymbol{p} \in \mathbb{R}_+^N \; ; \; \mathbf{1}^T \boldsymbol{p} = 1\}$, and $\boldsymbol{\Delta}_N^+$ is its (relative) interior. If $P$ is a Borel probability measure over $\mathbb{R}^N$, $Z \sim P$ is a random variable, and $f : \mathbb{R}^N \to \mathbb{R}^K$ is Borel, then the expectation of $f(Z)$ is denoted by $\mathbb{E}[f(Z)] = \mathbb{E}_P[f] = \mathbb{E}_P[f(Z)] = \mathbb{E}_{Z \sim P}[f(Z)]$. We use the standard asymptotic notations $O, \Theta$, and $\Omega$.

## 2 Problem formulation and preliminaries

**Classification tasks.** The essential objects in classification are the input sample space $\mathcal{X}$, the predicted classes $\mathcal{Y}$, and the classifiers. We fix two random variables $X$ and $Y$, taking values in sets $\mathcal{X}$ and $\mathcal{Y} \triangleq [C]$. Here, $(X, Y)$ is a pair comprised of an input sample and corresponding class label randomly drawn from $\mathcal{X} \times \mathcal{Y}$ with distribution $P_{X,Y}$. A probabilistic classifier is a function $\boldsymbol{h} : \mathcal{X} \to \boldsymbol{\Delta}_C$, where $h_c(x)$ represents the probability of sample $x \in \mathcal{X}$ falling in class $c \in \mathcal{Y}$. Thus, $\boldsymbol{h}$ gives rise to a $\mathcal{Y}$-valued random variable $\widehat{Y}$ via the distribution $P_{\widehat{Y}\mid X=x}(c) \triangleq h_c(x)$.

**Group-fairness constraints.** Let $S$ be a group attribute (e.g., race and/or sex), taking values in $\mathcal{S} \triangleq [A]$. We consider multi-class generalization of three commonly used group fairness criteria in Table 2. As observed by existing works [see, e.g., ABD$^+$18, MW18, CHKV19, WRC20, AAW$^+$20], each of these fairness constraints[6] can be written in the vector-inequality form $\mathbb{E}_{P_X}[\boldsymbol{Gh}] \le \boldsymbol{0}$ for a closed-form matrix-valued function $\boldsymbol{G} : \mathcal{X} \to \mathbb{R}^{K \times C}$. For instance, for statistical parity, the $\boldsymbol{G}$ matrix evaluated at a fixed individual $x \in \mathcal{X}$ has $K = 2AC$ rows indexed by $(\delta, a, c') \in \{0, 1\} \times [A] \times [C]$, where the $(\delta, a, c')$-th row is equal to $\left( (-1)^\delta P_S(a)^{-1} \sum_{c \in [C]} P_{S\mid X=x, Y=c}(a) h_c^{\text{base}}(x) - (\alpha + (-1)^\delta) \right) \boldsymbol{e}_{c'}$, with $\boldsymbol{e}_1, \cdots, \boldsymbol{e}_C$ denoting the standard basis for $\mathbb{R}^C$. The expressions for the $\boldsymbol{G}$ matrix corresponding to the other fairness metrics are given in Appendix A.1.8, with a detailed derivation of statistical parity in Appendix A.1.9. Note that $\boldsymbol{G}$ depends on $P_{S\mid X,Y}$. If the group attribute $S$ is part of the input feature $X$, then $P_{S\mid X,Y}$ is simply replaced with an indicator function. Otherwise, we approximate this conditional distribution by training a probabilistic classifier.

**Goal.** Our goal is to design an efficient post-processing method that takes a pre-trained classifier $\boldsymbol{h}^{\text{base}}$ that may violate some target group-fairness criteria and finds a fair classifier that has the most similar outputs (i.e., closest utility performance) to that of $\boldsymbol{h}^{\text{base}}$.

**Fairness through information-projection.** We formulate the fairness intervention problem as follows. For a fixed search space $\mathcal{H} \subset \boldsymbol{\Delta}_C^{\mathcal{X}} \triangleq \{\boldsymbol{h} : \mathcal{X} \to \boldsymbol{\Delta}_C\}$, a loss function $\text{err} : \boldsymbol{\Delta}_C^{\mathcal{X}} \times \boldsymbol{\Delta}_C^{\mathcal{X}} \to \mathbb{R}$, and a base classifier $\boldsymbol{h}^{\text{base}} \in \boldsymbol{\Delta}_C^{\mathcal{X}}$, one seeks to solve:

$$\underset{\boldsymbol{h} \in \mathcal{H}}{\text{minimize}} \; \text{err}\left(\boldsymbol{h}, \boldsymbol{h}^{\text{base}}\right) \quad \text{subject to } \mathbb{E}_{P_X}[\boldsymbol{Gh}] \le \boldsymbol{0}. \tag{1}$$

The function err quantifies the "closeness" between the scores given by $\boldsymbol{h}$ and $\boldsymbol{h}^{\text{base}}$. The constraint on $\boldsymbol{h}$ can encode any arbitrary statistical information about the joint distribution induced on the pair $(X, \widehat{Y})$. Specifically, any constraint $\mathbb{E}_{P_{X,\widehat{Y}}}[\boldsymbol{g}(X, \widehat{Y})] \le \boldsymbol{0}$, where $\boldsymbol{g} : \mathcal{X} \times [C] \to \mathbb{R}^K$, may be recast in the form (1). Thus, solving the optimization (1) amounts to finding the minimal necessary perturbation to the base classifier $\boldsymbol{h}^{\text{base}}$ to make it satisfy a given on-average constraint. Since we

---

[6] We remark that our framework can be applied to other fairness constraints, e.g., the ones in [WRC20].

consider raw output scores, we measure "closeness" via $f$-divergences:

$$\mathrm{err}\left(\boldsymbol{h}, \boldsymbol{h}^{\mathrm{base}}\right) = D_f(\boldsymbol{h} \| \boldsymbol{h}^{\mathrm{base}} \mid P_X) \triangleq \mathbb{E}_{P_X}\left[\sum_{c \in [C]} h_c^{\mathrm{base}}(X) f\left(\frac{h_c(X)}{h_c^{\mathrm{base}}(X)}\right)\right] - f(1), \quad (2)$$

where $f$ is a convex function over $(0, \infty)$. By varying different choices of $f$, we can obtain e.g., cross-entropy (CE, $f(t) = -\log t$) and KL-divergence ($f(t) = t \log t$). For a chosen $f$-divergence, the optimization problem (1) becomes a generalization of *information projection* [Csi75].

**Preliminaries on information-projection.** In a recent work [AAW$^+$20], an optimal solution for the information projection formulation (1) was theoretically characterized. We briefly describe this result next. Let[7] $\mathcal{H} \triangleq \{\boldsymbol{h} \in \mathcal{C}(\mathcal{X}, \boldsymbol{\Delta}_C) \; ; \; \inf_{c,x} h_c(x) > 0\}$ and we introduce the following definition and assumption.

**Definition 1.** For $\boldsymbol{p} \in \boldsymbol{\Delta}_C$, let $D_f^{\mathrm{conj}}(\,\cdot\,, \boldsymbol{p})$ denote the convex conjugate of $D_f(\,\cdot\, \| \boldsymbol{p})$:

$$D_f^{\mathrm{conj}}(\boldsymbol{v}, \boldsymbol{p}) \triangleq \sup_{\boldsymbol{q} \in \boldsymbol{\Delta}_C} \boldsymbol{v}^T \boldsymbol{q} - D_f(\boldsymbol{q} \| \boldsymbol{p}). \quad (3)$$

**Assumption 1.** Assume that: **(i)** $f \in \mathcal{C}^2(\mathbb{R})$, $f(1) = 0$, $f'(0^+) = -\infty$, and $f''(t) > 0$ for all $t > 0$; **(ii)** each $G_{k,c}$ is bounded, differentiable, and has bounded gradient; **(iii)** $\boldsymbol{h}^{\mathrm{base}} \in \mathcal{H}$, and each $h_c^{\mathrm{base}}$ has bounded partial derivatives; and **(iv)** there is an $\boldsymbol{h} \in \mathcal{H}$ such that $\mathbb{E}_{P_X}[\boldsymbol{G}\boldsymbol{h}] < \boldsymbol{0}$.

Now, the solution for (1) can be obtained by a simple "tilting" of the base classifier's output, as stated in the next theorem.

**Theorem 1** ([AAW$^+$20]). *If $f, \boldsymbol{h}^{\mathrm{base}}$, and $\boldsymbol{G}$ satisfy Assumption 1, and $\mathcal{X} = \mathbb{R}^d$, then there is a unique solution $\boldsymbol{h}^{\mathrm{opt}}$ for the optimization problem (1) for the $f$-divergence objective (2). Furthermore, $\boldsymbol{h}^{\mathrm{opt}}$ is given by the tilt*

$$h_c^{\mathrm{opt}}(x) = h_c^{\mathrm{base}}(x) \cdot \phi\left(v_c(x; \boldsymbol{\lambda}^\star) + \gamma(x; \boldsymbol{\lambda}^\star)\right), \qquad (x, c) \in \mathcal{X} \times [C], \quad (4)$$

*where:* **(i)** *the function $\phi$ denotes the inverse of $f'$;* **(ii)** *the function $\boldsymbol{v} : \mathcal{X} \times \mathbb{R}^K \to \mathbb{R}^C$ is defined by $\boldsymbol{v}(x; \boldsymbol{\lambda}) \triangleq -\boldsymbol{G}(x)^T \boldsymbol{\lambda}$;* **(iii)** *the function $\gamma : \mathcal{X} \times \mathbb{R}^K \to \mathbb{R}$ is characterized by the equation $\mathbb{E}_{c \sim \boldsymbol{h}^{\mathrm{base}}(x)}[\phi\left(v_c(x; \boldsymbol{\lambda}) + \gamma(x; \boldsymbol{\lambda})\right)] = 1$; and* **(iv)** *$\boldsymbol{\lambda}^\star \in \mathbb{R}^K$ is any solution to the convex problem*

$$D^\star \triangleq \min_{\boldsymbol{\lambda} \in \mathbb{R}_+^K} \mathbb{E}\left[D_f^{\mathrm{conj}}\left(\boldsymbol{v}(X; \boldsymbol{\lambda}), \boldsymbol{h}^{\mathrm{base}}(X)\right)\right]. \quad (5)$$

If the underlying data distribution is known, Theorem 1 yields an expression for the projected classifier as a post-processing of the base classifier. However, in practice, we do not know the underlying distribution and have to approximate it from a finite number of i.i.d. samples. In Section 3, we first describe how we approximate the solution given in Theorem 1 with finite samples. We then propose a parallelizable algorithm to solve the approximation in Section 4.

## 3 A finite-sample approximation of information projection

In practice, $P_X$ is unknown and only data points $\mathbb{X} \triangleq \{X_i\}_{i \in [N]} \subset \mathcal{X}$, drawn from $P_X$, are available. Thus, we propose the following fairness optimization problem. We search for a (multi-class) classifier $\boldsymbol{h} : \mathbb{X} \to \boldsymbol{\Delta}_C$ that solves the following:

$$\begin{aligned} \underset{\substack{\boldsymbol{h}:\mathbb{X}\to\boldsymbol{\Delta}_C \\ \boldsymbol{a}:\mathbb{X}\to\mathbb{R}^C, \boldsymbol{b}\in\mathbb{R}^K}}{\mathrm{minimize}} \quad & D_f\left(\boldsymbol{h} \| \boldsymbol{h}^{\mathrm{base}} \mid \widehat{P}_X\right) + \tau_1 \cdot \left(\mathbb{E}_{X \sim \widehat{P}_X}\left[\|\boldsymbol{a}(X)\|_2^2\right] + \|\boldsymbol{b}\|_2^2\right) \\ \mathrm{subject\ to} \quad & \mathbb{E}_{\widehat{P}_X}[\boldsymbol{G} \cdot (\boldsymbol{h} + \tau_2 \boldsymbol{a})] \le \tau_2 \boldsymbol{b}, \end{aligned} \quad (6)$$

with $\widehat{P}_X$ being the empirical measure (e.g., obtained from a dataset), and $\tau_1, \tau_2 > 0$ prescribed constants. The terms $\boldsymbol{a}$ and $\boldsymbol{b}$ are added to circumvent infeasibility issues and aid convergence of our

---

[7]Here, $\mathcal{C}(\mathcal{X}, \boldsymbol{\Delta}_C)$ denotes the complete metric space of continuous functions from $\mathcal{X}$ to $\boldsymbol{\Delta}_C$, equipped with the sup-norm, i.e., $\|\boldsymbol{h}\| \triangleq \sup_{x \in \mathcal{X}} \|\boldsymbol{h}(x)\|_1$. In addition, we restrict attention to classifiers bounded away from the simplex boundary to simplify the proof of strong duality in Theorem 2 (see Remark 1 on our assumptions).

numerical procedure. We show in the following theorem that there is a unique solution for (6), and that it is given by a tilt (i.e., multiplicative factor) of $h^{\mathrm{base}}$. The tilting parameter is the solution of a finite-dimensional strongly convex optimization problem.

**Theorem 2.** *Suppose Assumption 1 holds, and set $\zeta \triangleq \tau_2^2/(2\tau_1)$. There exists a unique solution $h^{\mathrm{opt},N}$ to (6), and it is given by the formula*

$$h_c^{\mathrm{opt},N}(x) = h_c^{\mathrm{base}}(x) \cdot \phi\left(v_c(x; \boldsymbol{\lambda}_{\zeta,N}^\star) + \gamma(x; \boldsymbol{\lambda}_{\zeta,N}^\star)\right), \quad (x,c) \in \mathbb{X} \times [C], \tag{7}$$

*with $\boldsymbol{v}, \phi, \gamma$ as in Theorem 1, and $\boldsymbol{\lambda}_{\zeta,N}^\star \in \mathbb{R}^K$ is the unique solution to the strongly convex problem*

$$D_{\zeta,N}^\star \triangleq \min_{\boldsymbol{\lambda} \in \mathbb{R}_+^K} \mathbb{E}_{\widehat{P}_X}\left[D_f^{\mathrm{conj}}\left(\boldsymbol{v}(X; \boldsymbol{\lambda}), h^{\mathrm{base}}(X)\right)\right] + \frac{\zeta}{2}\left\|\boldsymbol{\mathcal{G}}_N^T \boldsymbol{\lambda}\right\|_2^2 \tag{8}$$

*where $\boldsymbol{\mathcal{G}}_N \triangleq \left(\boldsymbol{G}(X_1)/\sqrt{N}, \cdots, \boldsymbol{G}(X_N)/\sqrt{N}, \boldsymbol{I}_K\right) \in \mathbb{R}^{K \times (NC+K)}$.*

*Proof.* See Appendix A.1.1. $\qquad\qquad\qquad\qquad\qquad\qquad\qquad\qquad\qquad\qquad\qquad\qquad\qquad\square$

Theorem 2 shows that: strong duality holds between the primal (6) and (the negative of) the dual (8); there is a unique classifier $h^{\mathrm{opt},N}$ minimizing our fairness formulation (6); there is a unique solution $\boldsymbol{\lambda}_{\zeta,N}^\star$ to the dual (5); and there is an explicit functional form of $h^{\mathrm{opt},N}$ in terms of $\boldsymbol{\lambda}_{\zeta,N}^\star$ in (7). Moreover, Theorem 2 yields a *practical* two-step procedure for solving the functional optimization in equation (6): (i) compute the dual variables $\boldsymbol{\lambda}$ by solving the strongly convex optimization in (8); (ii) tilt the base classifier by using the dual variables according to (7). This process is applied on real-world datasets using `FairProjection` (see Algorithm 1) in the next section.

The key distinctions between our formulation and Theorem 1 are that we use the empirical measure $\widehat{P}_X$ (e.g., produced using a dataset with i.i.d. samples), we have a *strongly* convex dual problem in (8) (in contrast to the convex program in (5)), and we prove strong duality in Theorem 2 (whereas an analogous strong duality is absent from the results of [AAW+20]).

**Remark 1.** In practice, Assumption 1 is not a limiting factor for Theorem 2 and `FairProjection`. This is because: we are considering here a finite-set domain so continuity is automatic; we can perturb $h^{\mathrm{base}}$ by negligible noise to push it away from the simplex boundary; and the uniform classifier is strictly feasible. Nevertheless, Assumption 1 simplifies the derivation of our theoretical results.

## 4 Fair projection and theoretical guarantees

We introduce a parallelizable algorithm, `FairProjection`, that solves (6) using $N$ i.i.d. data points. We prove that its utility converges to $D^\star$ (see (5)) in the population limit and establish both sample-complexity and convergence rate guarantees. Applying `FairProjection` to the group-fairness intervention problem in (1) yields the optimal parameters in (7) for post-processing (i.e., tilting) the output of a multi-class classifier in order to satisfy target fairness constraints.

The `FairProjection` algorithm uses ADMM [BPC+11] to solve the convex program in (8). Recall that it suffices to optimize (8) for computing (6) as proved in Theorem 2. Algorithm 1 presents the steps of `FairProjection`, and its detailed derivation is given in Appendix A.1.2. A salient feature of `FairProjection` is its *parallelizability*. Each step that is done for $i$ varying over $[N]$ can be executed for each $i$ separately and in parallel. In particular, this applies to the most computationally intensive step, the $\boldsymbol{v}_i$-update step. We discuss next how the $\boldsymbol{v}_i$-update step is carried out.

**Inner iterations.** One approach to carry out the inner iteration in Algorithm 1 that updates $\boldsymbol{v}_i$ is to study the vanishing of the gradient of $\boldsymbol{v} \mapsto D_f^{\mathrm{conj}}(\boldsymbol{v}, \boldsymbol{p}_i) + \xi\|\boldsymbol{v}\|_2^2 + \boldsymbol{a}_i^T \boldsymbol{v}$ (where $\xi = (\rho + \zeta)/2$ and $\boldsymbol{a}_i \in \mathbb{R}^C$ is some vector). In the KL-divergence case, $D_{\mathrm{KL}}^{\mathrm{conj}}$ is given by a log-sum-exp function, so its gradient is given by a softmax function, and equating the gradient to zero becomes a fixed-point equation. We give an iterative routine to solve this fixed point equation in Appendix A.1.3.1, whose proof of convergence is discussed in the same section. Beyond the KL-divergence case, setting the gradient to zero does not seem to be an analytically tractable problem. Nevertheless, we may reduce the vector minimization in Line 6 of Algorithm 1 to a tractable 1-dimensional root-finding problem, as the following result aids in showing.

**Lemma 1.** *For $p \in \Delta_C^+$, $a \in \mathbb{R}^C$, and $\xi > 0$, if $f$ satisfies Assumption 1, we have that*

$$\min_{v \in \mathbb{R}^C} D_f^{\text{conj}}(v, p) + \xi \|v\|_2^2 + a^T v = -\sup_{\theta \in \mathbb{R}} -\theta + \sum_{c \in [C]} \min_{q_c \geq 0} p_c f\left(\frac{q_c}{p_c}\right) + \frac{(a_c + q_c)^2}{4\xi} + \theta q_c. \quad (9)$$

*Proof.* See Appendix A.1.3.2. $\quad\square$

We note that the $v_i$-update steps for both KL and CE (provided in detail in Appendix A.1.3.3) give, as a byproduct, the implicitly defined function $\gamma(x; \lambda)$ (see the statements of Theorems 1–2).

**Convergence guarantees.** Our proposed algorithm, FairProjection, enjoys the following convergence guarantees. The output after the $t$-th iteration $\lambda_{\zeta,N}^{(t)}$ converges exponentially fast to $\lambda_{\zeta,N}^\star$ (see (8)).

**Theorem 3.** *Suppose Assumption 1 holds, and that the matrix $(G(X_i))_{i \in N} \in \mathbb{R}^{K \times NC}$ has full row-rank. Let $\lambda_{\zeta,N}^{(t)}$ and $h^{(t)}$ be the $t$-th iteration outputs of FairProjection for the KL-divergence case. Then, we have the exponential decay of errors $\|\lambda_{\zeta,N}^{(t)} - \lambda_{\zeta,N}^\star\|_2 = e^{-\Omega(t)}$ and $h^{(t)}(x) = h^{\text{opt},N}(x) \cdot \left(1 \pm e^{-\Omega(t)}\right)$ uniformly in $x \in \mathbb{X}$ as $t \to \infty$.*

*Proof.* See Appendix A.1.5. $\quad\square$

**Remark 2.** The full-rank assumption on the matrix $(G(X_i))_{i \in N} \in \mathbb{R}^{K \times NC}$ can be ensured by adding negligible noise to it. Further, although Theorem 3 is shown for the KL-divergence, the proof directly extends to general $f$-divergences satisfying Assumption 1 (see Appendix A.1.6 for further discussions). Finally, we show in Theorem 3 in Appendix A.1.7 that carrying $t = \Omega(\log N)$ iterations of FairProjection, with regularizer $\zeta = \Theta(N^{-1/2})$, yields a parameter $\lambda_{\zeta,N}^{(t)}$ that works well for the *population* problem for information projection (5); this makes FairProjection have a computational runtime of $O(N \log N)$.

**Benefit of parallelization.** The parallelizability of FairProjection provides significant speedup. In Appendix A.2.2, we provide an ablation study comparing the speedup due to parallelization. For the ENEM dataset (discussed next section), parallelization yields a 15-fold reduction in runtime. In addition to the parallel advantage of FairProjection, its inherent mathematical approach is more advantageous than gradient-based solutions. When numerically solving the dual problem (8) (or any close variant) via gradient methods, the gradient of $D_f^{\text{conj}}$ (the convex conjugate of an $f$-divergence) must be computed. However, this gradient is tractable in only a very limited number of relevant instances of $f$-divergences. FairProjection tackles this intractability through having its subroutines be informed by Lemma 1 and the discussion preceding it.

---

**Algorithm 1 : FairProjection for solving (8).**

---

1: **Input:** divergence $f$, predictions $\{p_i \triangleq h^{\text{base}}(X_i)\}_{i \in [N]}$, constraints $\{G_i \triangleq G(X_i)\}_{i \in [N]}$, regularizer $\zeta$, ADMM penalty $\rho$, and initializers $\lambda$ and $(w_i)_{i \in [N]}$.

2: **Output:** $h_c^{\text{opt},N}(x) \triangleq h_c^{\text{base}}(x) \cdot \phi(\gamma(x; \lambda) + v_c(x; \lambda))$.

3: $Q \leftarrow \frac{\zeta}{2} I + \frac{\rho}{2N} \sum_{i \in [N]} G_i G_i^T$

4: **for** $t = 1, 2, \cdots, t'$ **do**

5: $\quad a_i \leftarrow w_i + \rho G_i^T \lambda$ $\hfill i \in [N]$

6: $\quad v_i \leftarrow \underset{v \in \mathbb{R}^C}{\arg\min} \, D_f^{\text{conj}}(v, p_i) + \frac{\rho + \zeta}{2} \|v\|_2^2 + a_i^T v$ $\hfill i \in [N]$

7: $\quad q \leftarrow \frac{1}{N} \sum_{i \in [N]} G_i \cdot (w_i + v_i)$

8: $\quad \lambda \leftarrow \underset{\ell \in \mathbb{R}_+^K}{\arg\min} \, \ell^T Q \ell + q^T \ell$

9: $\quad w_i \leftarrow w_i + \rho \cdot \left(v_i + G_i^T \lambda\right)$ $\hfill i \in [N]$

10: **end for**

---

# 5 Numerical benchmarks

We present empirical results and show that `FairProjection` has competitive performance both in terms of runtime and fairness-accuracy trade-off curves compared to benchmarks—most notably the reductions approach in [ABD⁺18], which requires retraining. Extensive additional benchmarks and experiment details are reported in Appendix A.2.

**Setup.** We consider three base classifiers (`Base`): gradient boosting (GBM), logistic regression (LR), and random forest (RF), implemented by Scikit-learn [PVG⁺11]. For `FairProjection` (the constrained optimization in (6)), we use cross-entropy (`FairProjection-CE`) and KL-divergence (`FairProjection-KL`) as the loss function[8]. We consider two fairness constraints: mean equalized odds (MEO) and statistical parity (SP) (cf. Table 2). Particularly, to measure multi-class performance, we extend the definition of MEO as

$$\mathsf{MEO} = \max_{i \in \mathcal{Y}} \max_{s_1, s_2 \in \mathcal{S}} (|\mathsf{TPR}_i(s_1) - \mathsf{TPR}_i(s_2)| + |\mathsf{FPR}_i(s_1) - \mathsf{FPR}_i(s_2)|)/2 \qquad (10)$$

where $\mathsf{TPR}_i(s) = P(\widehat{Y} = i | Y = i, S = s)$, and $\mathsf{FPR}_i(s) = P(\widehat{Y} = i | Y \neq i, S = s)$. The definition of multi-class statistical parity is provided in Appendix A.2.4.2. All values reported in this section are from the test set with 70/30 train-test split. When benchmarking against methods tailored to binary classification, we restrict our results to both binary $Y$ and $S$ since, unlike `FairProjection`, competing methods cannot necessarily handle multi-class predictions and multiple groups.

**Datasets.** We evaluate `FairProjection` and all benchmarks on four datasets. We use two datasets in the education domain: the high-school longitudinal study (HSLS) dataset [IPH⁺11, JWC22] and a novel dataset ENEM [INE20] (details in Appendix A.2.1). The ENEM dataset contains Brazilian college entrance exam scores along with student demographic information and socio-economic questionnaire answers (e.g., if they own a computer). After pre-processing, the dataset contains ∼1.4 million samples with 139 features. Race was used as the group attribute $S$, and Humanities exam score is used as the label $Y$. The score can be quantized into an arbitrary number of classes. For binary experiments, we quantize $Y$ into two classes, and for multi-class, we quantize it to 5 classes. The race feature $S$ has 5 categories, but we binarize it into White and Asian ($S = 1$) and others ($S = 0$). We call the entire ENEM dataset ENEM-1.4M. We also created smaller versions of the dataset with 50k samples: ENEM-50k-2C (binary classes) and ENEM-50k-5C (5 classes).[9] For completeness, we report results on UCI Adult [Lic13] and COMPAS [ALMK16].

**Benchmarks.** For binary classification experiments, we compare our method with five existing fair learning algorithms: `Reduction` [ABD⁺18], reject-option classifier [KKZ12, `Rejection`], equalized-odds [HPS16, `EqOdds`], calibrated equalized-odds [PRW⁺17, `CalEqOdds`], and leveraging equal opportunity [CDH⁺19, `LevEqOpp`].[10] The choice of benchmarks is based on the availability of reproducible codes. For the first four baselines, we use IBM AIF360 library [BDH⁺18]. For `Reduction` and `Rejection`, we vary the tolerance to achieve different operation points on the fairness-accuracy trade-off curves. As `EqOdds`, `CalEqOdds` and `LevEqOpp` only allow hard equality constraint on equalized odds, they each produce a single point on the plot (see Fig. 1). We include the group attribute as a feature in the training set following the same benchmark procedure described in [ABD⁺18, WRC21] for a consistent comparison. For multi-class classification experiments, we did not find methods that can be easily compared against `FairProjection` and use the multi-class extensions of mean equalized odds and statistical parity. For the sake of completeness, we modified the codes of adversarial debiasing [ZLM18, `Adversarial`], and compare our method against it. Note that `Reduction` [ABD⁺18] and `Adversarial` [ZLM18] are in-processing methods, and the rest of the benchmark algorithms are post-processing methods like `FairProjection`. Additional comparisons to [KCT20] are given in Appendix A.2.4.1.

There are four methods in Table 1 we did not include the experiments: FACT [KCT20], Identifying [JN20], FST [WRC21], and Overlapping [YCK20], as explained in Appendix A.2.1.3.

---

[8]We focus on `FairProjection-CE` and random forest here; results for `FairProjection-KL` and other models are in Appendix A.2.

[9]A datasheet (see [GMV⁺21]) for ENEM is given in Appendix A.3.

[10]https://github.com/lucaoneto/NIPS2019_Fairness.

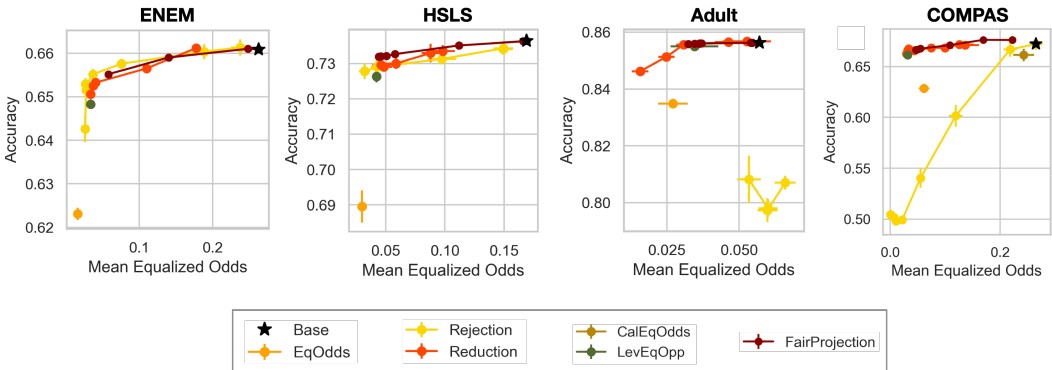

**Figure 1:** Fairness-accuracy trade-off comparisons between `FairProjection` and five baselines on ENEM-50k-2C, HSLS, Adult and COMPAS datasets. For all methods, we used random forest as a base classifier. Note that `EqOdds`, `CalEqOdds`, and `LevEqOpp` only produce a single accuracy-fairness trade-off point, whereas the rest of the methods are capable of producing the accuracy-fairness trade-off curves by varying the fairness budget $\alpha$ for the group fairness criteria listed in Table 2 — a smaller $\alpha$ corresponds to a lefter point on the accuracy-fairness trade-off curve.

**Binary classification results.** We compare `FairProjection` with benchmarks tailored to binary classification in terms of the MEO-accuracy trade-off on the ENEM-50k-2C, HSLS, Adult, and COMPAS datasets in Fig. 1. Each point is obtained by averaging 10 runs with different train-test splits. `FairProjection-CE` curves were obtained by varying $\alpha$ values (cf. Table 2). When $\alpha = 1.0$, the outputs of `FairProjection-CE` are equivalent to the base classifier RF.

We observe that `FairProjection-CE` and `Reduction` have the overall best and most consistent performances. On ENEM-50k-2C and HSLS datasets, although `EqOdds` achieves the best fairness, that fairness comes at the cost of $4\%$ accuracy drop (additively). The other four methods, on the other hand, produce comparatively good fairness with an accuracy loss of $< 1\%$. In particular, `FairProjection-CE` has the smallest accuracy drop whilst improving MEO from 0.17 to 0.04 on HSLS. `CalEqOdds` requires strict calibration requirements and yields inconsistent performance when these requirements are not met. On ENEM-50k-2C and HSLS, `LevEqOpp` achieves comparable MEO with a slight accuracy drop, and on COMPAS, `LevEqOpp` performs equally well as `FairProjection-CE` and `Reduction`. Note that with high fairness constraints (i.e., small tolerance), the accuracy of `Rejection` deteriorates.

**Multi-Class results.** We illustrate how `FairProjection` performs on multi-class prediction using HSLS and ENEM-50k-5C. For HSLS, we divided student math performance into quartiles and generated four classes. In Figure 2, we plot fairness-accuracy trade-off of `FairProjection-CE` with logistic regression and adversarial debiasing [ZLM18, `Adversarial`]. As their base classifiers are different (`Adversarial` is a GAN-based method), we plot accuracy difference compared to the base classifier instead of plotting the absolute value of accuracy[11]. `FairProjection` reduces MEO significantly with very small loss in accuracy. While `Adversarial` is also able to reduce MEO with negligible accuracy drop, it does not reduce the MEO as much as `FairProjection`. We show more extensive results with multi-group and multi-class ($|\mathcal{Y}| = 5, = |\mathcal{S}| = 5$) in Appendix A.2.4.2.

**Runtime comparisons.** To demonstrate the scalability of `FairProjection`, in Table 3, we record the runtime of `FairProjection-CE` and `-KL` with the five benchmarks on ENEM-1.4M-2C, which is the biggest dataset we have. These experiments were run on a machine with AMD Ryzen 2990WX 64-thread 32-Core CPU and NVIDIA TITAN Xp 12-GB GPU. For consistency, we used the same fairness metric (MEO, $\alpha = 0.01$), base classifier (GBM), and train/test split, and each number is the average of 2 repeated experiments. `EqOdds`, `LevEqOpp`, and `CalEqOdds` are faster than `FairProjection` since they are optimized to produce one trade-off point (cf. Fig. 1). Compared to baselines that produce full fairness-accuracy trade-off curves (i.e., `Reduction` and `Rejection`), `FairProjection` has the fastest runtime. Also, the non-parallel implementation of `FairProjection-KL` takes 25.3 mins—parallelization attains $15\times$ speedup (detailed results in Appendix A.2.2). We further compare

---

[11]Base accuracy for `FairProjection` = 0.336, `Adversarial` = 0.307. Random guessing accuracy = 0.2.

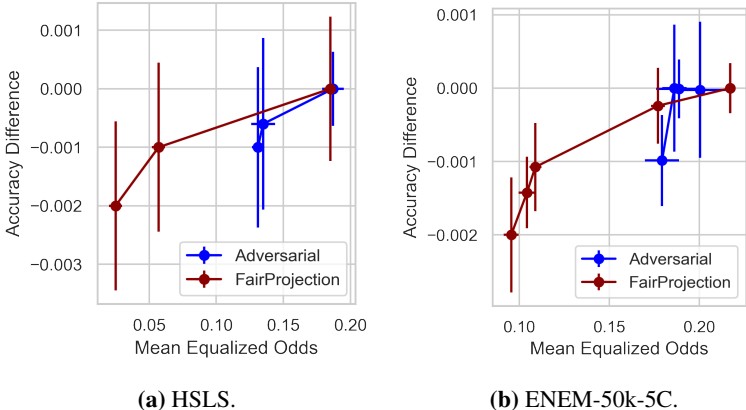

| (a) HSLS. | (b) ENEM-50k-5C. |

**Figure 2:** Fairness-accuracy trade-off for multi-class prediction on HSLS and ENEM-50k-5C. FairProjection is `FairProjection-CE` with LR base classifier.

| Method | *Reduction* [ABD+18] | **Rejection** [KKZ12] | EqOdds [HPS16] | LevEqOpp [CDH+19] | CalEqOdds [PRW+17] | *FairProjection (ours)* CE | KL |
|---|---|---|---|---|---|---|---|
| **Runtime** | 223.6 | 16.9 | 5.9 | 7.9 | 5.3 | 11.3 | 11.6 |

**Table 3:** Execution time of `FairProjection` on the ENEM-1.4M-2C compared with five baseline methods (time shown in minutes). Methods in **bold** are capable of producing a fairness-accuracy trade-off curve. Methods that are *italicized* have a uniformly superior performance. The time reported here for `FairProjection` includes the time to fit the base classifiers. If base classifiers are given, the runtime of e.g. `FairProjection-KL` is 1.63 mins. The runtimes are consistent with small standard deviations across repeated experiments.

the runtime results for the binary HSLS, which is the second biggest dataset, with the baselines that produce full fairness-accuracy trade-off curves. The runtimes for `Reduction`, `Rejection` and `FairProjection-CE` are $81.1$ sec, $9.73$ sec and $4.50$ sec respectively—again, `FairProjection` has the fastest runtime. For a theoretical comparison between the runtime of `FairProjection` and `Reduction`, see Appendix A.2.3.

## 6   Final remarks and limitations

We introduce a theoretically-grounded and versatile fairness intervention method, `FairProjection`, and showcase its favorable performance in extensive experiments. We encourage the reader to peruse our theoretical result in Appendix A.1 and extensive additional numerical benchmarks in Appendix A.2. `FairProjection` is able to correct bias for multigroup/multiclass datasets, and it enjoys a fast runtime thanks to its parallelizability. We also evaluate our method on the ENEM dataset (see Appendix A.3 for a detailed description of the dataset). Our benchmarks are a step forward in moving away from the overused COMPAS and UCI Adult datasets.

We only consider group-fairness, and it would be interesting to try to incorporate other fairness notions (e.g., individual fairness [DHP+12]) into our formulation. We assume that $h^{\text{base}}$ is a pre-trained accurate (and potentially unfair) classifier; one future research direction is understanding how the accuracy of $h^{\text{base}}$ influences the performance of the projected classifier. Finally, the performance of `FairProjection` is inherently constrained by data availability. Performance may degrade with intersectional increases of the number of groups, the number of labels, and the number of fairness constraints.

## Acknowledgement

We thank the anonymous referees for their careful critique, which helped improve the quality of the paper considerably. This material is based upon work supported by the National Science Foundation under grants CAREER 1845852, IIS 1926925, FAI 2040880, CIF 1900750, an HDSI Bias^2 award, a gift from Oracle Research, and Meta Ph.D. Fellowship.

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
