# Appendix
# Beyond Adult and COMPAS:
# Fair Multi-Class Prediction via Information Projection

**Wael Alghamdi**[1*]**, Hsiang Hsu**[1*]**, Haewon Jeong**[1*]**,**
**Hao Wang**[1]**, P. Winston Michalak**[1]**, Shahab Asoodeh**[2]**, Flavio P. Calmon**[1]
[1]John A. Paulson School of Engineering and Applied Sciences, Harvard University
[2]Department of Computing and Software, McMaster University

This appendix is divided into three parts: Appendix A.1: Proofs of theoretical results; Appendix A.2: More details on the experimental setup, additional quantitative experiments, and more qualitative comparisons with related work; and Appendix A.3: A datasheet for ENEM (2020) dataset.

## A.1   Proofs of theoretical results

The theoretical details of our work are included in this appendix. We prove the strong duality stated in Theorem 2 in Appendix A.1.1. Algorithm 1 is derived in Appendix A.1.2. The inner iterations of Algorithm 1 are further developed in Appendices A.1.3–A.1.4. The convergence rate result in Theorem 3 is proved in Appendix A.1.5, and an extension of it (to general $f$-divergences) is discussed in Appendix A.1.6. The performance of `FairProjection` for the population problem (5) is stated in Theorem 3 in A.1.7 and proved there too. Explicit formulas for the $G$ matrix induced by the fairness metrics in Table 2 are given in Appendices A.1.8 and A.1.9.

### A.1.1   Proof of Theorem 2: strong duality

We use the following minimax theorem, which is a generalization of Sion's minimax theorem.

**Theorem 1** ([ET99b], Chapter VI, Prop. 2.2]). *Let $V$ and $Z$ be two reflexive Banach spaces, and fix two convex, closed, and non-empty subsets $\mathcal{A} \subset V$ and $\mathcal{B} \subset Z$. Let $L : \mathcal{A} \times \mathcal{B} \to \mathbb{R}$ be a function such that for each $u \in \mathcal{A}$ the function $p \mapsto L(u, p)$ is concave and upper semicontinuous, and for each $p \in \mathcal{B}$ the function $u \mapsto L(u, p)$ is convex and lower semicontinuous. Suppose that there exist points $u_0 \in \mathcal{A}$ and $p_0 \in \mathcal{B}$ such that $\lim_{p \in \mathcal{B}, \|p\| \to \infty} L(u_0, p) = -\infty$ and $\lim_{u \in \mathcal{A}, \|u\| \to \infty} L(u, p_0) = \infty$. Then, $L$ has at least one saddle-point $(\overline{u}, \overline{p})$, and*

$$L(\overline{u}, \overline{p}) = \min_{u \in \mathcal{A}} \sup_{p \in \mathcal{B}} L(u, p) = \max_{p \in \mathcal{B}} \inf_{u \in \mathcal{A}} L(u, p). \tag{A.1}$$

*In particular, in (A.1), there exists a minimizer in $\mathcal{A}$ of the outer minimization, and a maximizer in $\mathcal{B}$ of the outer maximization.*

Denote $\boldsymbol{h}_i \triangleq \boldsymbol{h}(X_i)$, $\boldsymbol{p}_i \triangleq \boldsymbol{h}^{\mathrm{base}}(X_i)$, $\boldsymbol{a}_i \triangleq \boldsymbol{a}(X_i)$, and $\boldsymbol{G}_i \triangleq \boldsymbol{G}(X_i)$, and let the matrix $\boldsymbol{\mathcal{G}}_N \triangleq \left( \boldsymbol{G}_1/\sqrt{N}, \cdots, \boldsymbol{G}_N/\sqrt{N}, \boldsymbol{I}_K \right) \in \mathbb{R}^{K \times (NC+K)}$ be as in the theorem statement. We may rewrite the

---

*Equal contribution. Correspondence to: Wael Alghamdi and Flavio P. Calmon (`alghamdi@g.harvard.edu` and `flavio@seas.harvard.edu`).

optimization (6) as

$$\begin{array}{ll}
\underset{(\boldsymbol{h}_i, \boldsymbol{a}_i, \boldsymbol{b}) \in \boldsymbol{\Delta}_C \times \mathbb{R}^C \times \mathbb{R}^K, i \in [N]}{\text{minimize}} & \frac{1}{N} \sum_{i \in [N]} D_f\left(\boldsymbol{h}_i \| \boldsymbol{p}_i\right) + \tau_1 \cdot \left(\|\boldsymbol{a}_i\|_2^2 + \|\boldsymbol{b}\|_2^2\right) \\
\text{subject to} & \frac{1}{N} \sum_{i \in [N]} \boldsymbol{G}_i \boldsymbol{h}_i + \tau_2 \cdot \left(\boldsymbol{G}_i \boldsymbol{a}_i - \boldsymbol{b}\right) \le \boldsymbol{0}.
\end{array} \tag{A.2}$$

We define $f$ at 0 by the right limit $f(0) \triangleq f(0+)$. Assume for now that $f(0+) < \infty$, and we will explain at the end of this proof how to treat the case $f(0+) = \infty$. For the optimization problem (A.2), the Lagrangian $L : \boldsymbol{\Delta}_C^N \times \mathbb{R}^{NC} \times \mathbb{R}^K \times \mathbb{R}_+^K \to \mathbb{R}$ is given by

$$\begin{aligned}
L\left((\boldsymbol{h}_i)_{i \in [N]}, (\boldsymbol{a}_i)_{i \in [N]}, \boldsymbol{b}, \boldsymbol{\lambda}\right) \triangleq & \frac{1}{N} \sum_{i \in [N]} D_f\left(\boldsymbol{h}_i \| \boldsymbol{p}_i\right) + \tau_1 \left(\|\boldsymbol{a}_i\|_2^2 + \|\boldsymbol{b}\|_2^2\right) \\
& + \boldsymbol{\lambda}^T \left(\boldsymbol{G}_i \boldsymbol{h}_i + \tau_2 \left(\boldsymbol{G}_i \boldsymbol{a}_i - \boldsymbol{b}\right)\right).
\end{aligned} \tag{A.3}$$

With $\boldsymbol{v}(x; \boldsymbol{\lambda}) \triangleq -\boldsymbol{G}(x)^T \boldsymbol{\lambda}$ as in the theorem statement, and denoting $\boldsymbol{v}_i \triangleq \boldsymbol{v}(X_i; \boldsymbol{\lambda}) = -\boldsymbol{G}_i^T \boldsymbol{\lambda}$, we may rewrite the Lagrangian as

$$\begin{aligned}
L\left((\boldsymbol{h}_i)_{i \in [N]}, (\boldsymbol{a}_i)_{i \in [N]}, \boldsymbol{b}, \boldsymbol{\lambda}\right) = & \frac{1}{N} \sum_{i \in [N]} D_f\left(\boldsymbol{h}_i \| \boldsymbol{p}_i\right) - \boldsymbol{v}_i^T \boldsymbol{h}_i + \tau_1 \|\boldsymbol{a}_i\|_2^2 - \tau_2 \boldsymbol{v}_i^T \boldsymbol{a}_i \\
& + \tau_1 \|\boldsymbol{b}\|_2^2 - \tau_2 \boldsymbol{\lambda}^T \boldsymbol{b}.
\end{aligned} \tag{A.4}$$

The optimization problem (A.2) can be written as

$$\inf_{(\boldsymbol{h}_i, \boldsymbol{a}_i, \boldsymbol{b}) \in \boldsymbol{\Delta}_C \times \mathbb{R}^C \times \mathbb{R}^K, i \in [N]} \sup_{\boldsymbol{\lambda} \in \mathbb{R}_+^K} L\left((\boldsymbol{h}_i)_{i \in [N]}, (\boldsymbol{a}_i)_{i \in [N]}, \boldsymbol{b}, \boldsymbol{\lambda}\right). \tag{A.5}$$

We check that the Lagrangian $L$ satisfies the conditions in Theorem 1. First, any Euclidean space $\mathbb{R}^M$ (for $M \in \mathbb{N}$) is a reflexive Banach space since it is finite-dimensional. In addition, the convex nonempty sets $\boldsymbol{\Delta}_C^N \times \mathbb{R}^{NC} \times \mathbb{R}^K$ and $\mathbb{R}_+^K$ are closed in their respective ambient Euclidean spaces. By continuity and convexity of $f$, and linearity of $L$ in $\boldsymbol{\lambda}$, we have that $L$ satisfies all the convexity, concavity, and semicontinuity conditions in Theorem 1. Further, fixing any $\boldsymbol{h}_i \in \boldsymbol{\Delta}_C, i \in [N]$, and letting $\boldsymbol{a}_i = \boldsymbol{0}, i \in [N]$, and $\boldsymbol{b} = \frac{1}{\tau_2}\left(\boldsymbol{1} + \frac{1}{N} \sum_{i \in [N]} \boldsymbol{G}_i \boldsymbol{h}_i\right)$, we would get that

$$L\left((\boldsymbol{h}_i)_{i \in [N]}, (\boldsymbol{a}_i)_{i \in [N]}, \boldsymbol{b}, \boldsymbol{\lambda}\right) = -\boldsymbol{\lambda}^T \boldsymbol{1} + \frac{1}{N} \sum_{i \in [N]} D_f\left(\boldsymbol{h}_i \| \boldsymbol{p}_i\right) + \tau_1 \|\boldsymbol{b}\|_2^2 \to -\infty \quad \text{as} \quad \|\boldsymbol{\lambda}\|_2 \to \infty. \tag{A.6}$$

In addition, choosing $\boldsymbol{\lambda} = \boldsymbol{0}$, we have the Lagrangian

$$L\left((\boldsymbol{h}_i)_{i \in [N]}, (\boldsymbol{a}_i)_{i \in [N]}, \boldsymbol{b}, \boldsymbol{\lambda}\right) = \frac{1}{N} \sum_{i \in [N]} D_f\left(\boldsymbol{h}_i \| \boldsymbol{p}_i\right) + \tau_1 \|\boldsymbol{a}_i\|_2^2 + \tau_1 \|\boldsymbol{b}\|_2^2 \to \infty \tag{A.7}$$

as $\|\boldsymbol{b}\|_2 + \sum_{i \in [N]} \|\boldsymbol{h}_i\|_2 + \|\boldsymbol{a}_i\|_2 \to \infty$. Thus, we may apply the minimax result in Theorem 1 to obtain the existence of a saddle-point of $L$ and that

$$\begin{aligned}
\min_{(\boldsymbol{h}_i, \boldsymbol{a}_i, \boldsymbol{b}) \in \boldsymbol{\Delta}_C \times \mathbb{R}^C \times \mathbb{R}^K, i \in [N]} & \sup_{\boldsymbol{\lambda} \in \mathbb{R}_+^K} L\left((\boldsymbol{h}_i)_{i \in [N]}, (\boldsymbol{a}_i)_{i \in [N]}, \boldsymbol{b}, \boldsymbol{\lambda}\right) \\
& = \max_{\boldsymbol{\lambda} \in \mathbb{R}_+^K} \inf_{(\boldsymbol{h}_i, \boldsymbol{a}_i, \boldsymbol{b}) \in \boldsymbol{\Delta}_C \times \mathbb{R}^C \times \mathbb{R}^K, i \in [N]} L\left((\boldsymbol{h}_i)_{i \in [N]}, (\boldsymbol{a}_i)_{i \in [N]}, \boldsymbol{b}, \boldsymbol{\lambda}\right).
\end{aligned} \tag{A.8}$$

In particular, there exists a minimizer $(\boldsymbol{h}_i^{\text{opt},N}, \boldsymbol{a}_i^{\text{opt},N}, \boldsymbol{b}^{\text{opt},N}) \in \boldsymbol{\Delta}_C \times \mathbb{R}^C \times \mathbb{R}^K, i \in [N]$, of the outer minimization in the left-hand side in (A.8), and a maximizer $\boldsymbol{\lambda}^\star \in \mathbb{R}_+^K$ of the outer maximization in the right-hand side of (A.8). By strict convexity of the objective function in (A.2) (and convexity of the feasibility set), we obtain that the minimizer $(\boldsymbol{h}_i^{\text{opt},N}, \boldsymbol{a}_i^{\text{opt},N}, \boldsymbol{b}^{\text{opt},N}) \in \boldsymbol{\Delta}_C \times \mathbb{R}^C \times \mathbb{R}^K, i \in [N]$, is unique. We show next that the optimizer $\boldsymbol{\lambda}^\star$ is unique too, which we will denote by $\boldsymbol{\lambda}_{\zeta,N}^\star$ as in the theorem statement. We also show that, for each fixed $\boldsymbol{\lambda} \in \mathbb{R}_+^K$, there is a unique minimizer

$(h_i^{\boldsymbol{\lambda}}, a_i^{\boldsymbol{\lambda}}, b^{\boldsymbol{\lambda}}) \in \boldsymbol{\Delta}_C \times \mathbb{R}^C \times \mathbb{R}^K, i \in [N]$, of the *inner* minimization in the right-hand side of (A.8); by strict convexity of $f$, this would imply that $h_i^{\mathrm{opt},N} = h_i^{\boldsymbol{\lambda}_{\zeta,N}^{\star}}$.

Now, fix $\boldsymbol{\lambda} \in \mathbb{R}_+^K$, and consider the inner minimization in (A.8). We have that

$$\inf_{(h_i, a_i, b) \in \boldsymbol{\Delta}_C \times \mathbb{R}^C \times \mathbb{R}^K, i \in [N]} L\left((h_i)_{i \in [N]}, (a_i)_{i \in [N]}, b, \boldsymbol{\lambda}\right)$$

$$= \inf_{(h_i, a_i, b) \in \boldsymbol{\Delta}_C \times \mathbb{R}^C \times \mathbb{R}^K, i \in [N]} \frac{1}{N} \sum_{i \in [N]} D_f\left(h_i \| p_i\right) - v_i^T h_i + \tau_1 \|a_i\|_2^2 - \tau_2 v_i^T a_i + \tau_1 \|b\|_2^2 - \tau_2 \boldsymbol{\lambda}^T b \tag{A.9}$$

$$= \frac{1}{N} \sum_{i \in [N]} \inf_{h_i \in \boldsymbol{\Delta}_C} D_f(h_i \| p_i) - v_i^T h_i + \inf_{a_i \in \mathbb{R}^C} \tau_1 \|a_i\|_2^2 - \tau_2 v_i^T a_i + \inf_{b \in \mathbb{R}^K} \tau_1 \|b\|_2^2 - \tau_2 \boldsymbol{\lambda}^T b \tag{A.10}$$

$$= \frac{1}{N} \sum_{i \in [N]} -D_f^{\mathrm{conj}}(v_i, p_i) - \frac{1}{2} \zeta \|v_i\|_2^2 - \frac{1}{2} \zeta \|\boldsymbol{\lambda}\|_2^2 \tag{A.11}$$

$$= -\frac{\zeta}{2} \left\| \boldsymbol{\mathcal{G}}_N^T \boldsymbol{\lambda} \right\|_2^2 - \frac{1}{N} \sum_{i \in [N]} D_f^{\mathrm{conj}}(v_i, p_i) \tag{A.12}$$

where $\zeta \triangleq \tau_2^2/(2\tau_1)$. Here, the minimizers are $a_i^{\boldsymbol{\lambda}} \triangleq \frac{\tau_2}{2\tau_1} v_i$ and $b_i^{\boldsymbol{\lambda}} \triangleq \frac{\tau_2}{2\tau_1} \boldsymbol{\lambda}$, and $h_i^{\boldsymbol{\lambda}}$ is the unique probability vector in $\boldsymbol{\Delta}_C$ for which $D_f^{\mathrm{conj}}(v_i, p_i) = D_f(h_i^{\boldsymbol{\lambda}} \| p_i) - v_i^T h_i^{\boldsymbol{\lambda}}$; the existence and uniqueness of $h_i^{\boldsymbol{\lambda}}$ is guaranteed since $q \mapsto D_f(q \| p_i) - v_i^T q$ is lower semicontinuous and strictly convex, and $\boldsymbol{\Delta}_C$ is compact. Rewriting it in the form (A.12), the function

$$\boldsymbol{\lambda} \mapsto \inf_{(h_i, a_i, b) \in \boldsymbol{\Delta}_C \times \mathbb{R}^C \times \mathbb{R}^K, i \in [N]} L\left((h_i)_{i \in [N]}, (a_i)_{i \in [N]}, b, \boldsymbol{\lambda}\right) \tag{A.13}$$

can be seen to be strictly concave. Indeed, the function $\boldsymbol{\lambda} \mapsto \left\| \boldsymbol{\mathcal{G}}_N^T \boldsymbol{\lambda} \right\|_2^2$ is strictly convex. Also, each function $\boldsymbol{\lambda} \mapsto D_f^{\mathrm{conj}}(v_i, p_i)$ is convex as it is a pointwise supremum of linear functions: recalling that $v_i = -G_i^T \boldsymbol{\lambda}$, we have the formula

$$D_f^{\mathrm{conj}}(v_i, p_i) = \sup_{q \in \boldsymbol{\Delta}_C} -q^T G_i^T \boldsymbol{\lambda} - D_f(q \| p_i). \tag{A.14}$$

Hence, the outer maximizer $\boldsymbol{\lambda}^{\star}$ in (A.8) is indeed unique, which we denote by $\boldsymbol{\lambda}_{\zeta,N}^{\star}$. Note that $\boldsymbol{\lambda}_{\zeta,N}^{\star}$ is the unique solution to the *minimization* (8), i.e.,

$$\boldsymbol{\lambda}_{\zeta,N}^{\star} = \underset{\boldsymbol{\lambda} \in \mathbb{R}_+^K}{\mathrm{argmin}} \frac{1}{N} \sum_{i \in [N]} D_f^{\mathrm{conj}}(v_i, p_i) + \frac{\zeta}{2} \left\| \boldsymbol{\mathcal{G}}_N^T \boldsymbol{\lambda} \right\|_2^2, \tag{A.15}$$

as stated by the theorem.

Since $h^{\mathrm{opt},N} = h^{\boldsymbol{\lambda}_{\zeta,N}^{\star}}$, the following formula for $h^{\boldsymbol{\lambda}}$ (for a general $\boldsymbol{\lambda} \in \mathbb{R}_+^K$) yields the desired functional form (7) for $h^{\mathrm{opt},N}$ in terms of $\boldsymbol{\lambda}_{\zeta,N}^{\star}$.

**Lemma 1** ([AAW+20a, Lemma 4]). *Let $f : [0, \infty) \to \mathbb{R} \cup \{\infty\}$ be a strictly convex function that is continuously differentiable over $(0, \infty)$ and satisfying $f(0) = f(0+)$, $f(1) = 0$, and $f'(0+) = -\infty$. Let $\phi$ denote the inverse of $f'$. Fix $p \in \boldsymbol{\Delta}_C^+$ and $v \in \mathbb{R}^C$. Then, the unique minimizer of $q \mapsto D_f(q \| p) - v^T q$ over $q \in \boldsymbol{\Delta}_C$ is given by $q_c^{\star} = p_c \cdot \phi(\gamma + v_c)$, $c \in [C]$, where $\gamma \in \mathbb{R}$ is the unique number satisfying $\mathbb{E}_{c \sim p}[\phi(\gamma + v_c)] = 1$.*

From Lemma 1, and using $v(x; \boldsymbol{\lambda}_{\zeta,N}^{\star}) = -G(x)^T \boldsymbol{\lambda}_{\zeta,N}^{\star}$ and $\phi = (f')^{-1}$, we get that there exists a uniquely defined function $\gamma : \mathbb{X} \times \mathbb{R}^K \to \mathbb{R}$ for which

$$\mathbb{E}_{c \sim h^{\mathrm{base}}(x)}\left[\phi\left(\gamma(x; \boldsymbol{\lambda}_{\zeta,N}^{\star}) + v_c(x; \boldsymbol{\lambda}_{\zeta,N}^{\star})\right)\right] = 1 \tag{A.16}$$

for every $x \in \mathbb{X}$. For this $\gamma$, we know from Lemma 1 that

$$h_c^{\boldsymbol{\lambda}_{\zeta,N}^{\star}}(x) = h_c^{\mathrm{base}}(x) \cdot \phi\left(\gamma(x; \boldsymbol{\lambda}_{\zeta,N}^{\star}) + v_c(x; \boldsymbol{\lambda}_{\zeta,N}^{\star})\right) \tag{A.17}$$

for every $c \in [C]$ and $x \in \mathbb{X}$. Since $\boldsymbol{h}^{\mathrm{opt},N} = \boldsymbol{h}^{\boldsymbol{\lambda}^\star_{\zeta,N}}$, we obtain formula (7) for $\boldsymbol{h}^{\mathrm{opt},N}$ in terms of $\boldsymbol{\lambda}^\star_{\zeta,N}$, and the proof of Theorem 2 is complete in the case $f(0+) < \infty$.

Finally, we note how the case $f(0+) = \infty$ is treated, so assume $f(0) = f(0+) = \infty$. The only difference in this case is that the Lagrangian $L$ might attain the value $\infty$, whereas we need it to be $\mathbb{R}$-valued to apply the minimax result in Theorem 1. Nevertheless, the only way $L$ can be infinite is if some classifier $\boldsymbol{h}_i$ has an entry equal to 0, in which case the objective function in (6) (or (A.2)) will also be infinite, so such a classifier can be thrown out without affecting the optimization problem. More precisely, we still have strict convexity and lower semicontinuity of the objective function in (A.2). Thus, there is a unique minimizer $\boldsymbol{h}^{\mathrm{opt},N}$ of (A.2). For this optimizer, there must be an $\varepsilon_1 > 0$ such that $\boldsymbol{h}^{\mathrm{opt},N}(x) \geq \varepsilon_1 \mathbf{1}$ for *every* $x \in \mathbb{X}$. Thus, the optimization problem (A.2) remains unchanged if $\boldsymbol{\Delta}_C$ is restricted to classifiers bounded away from 0 by $\varepsilon_1$. Moreover, by the same reasoning, the optimization problem (A.14) for finding $D_f^{\mathrm{conj}}$ also remains unchanged if $\boldsymbol{\Delta}_C$ is replaced by the set of classifiers bounded away from 0 by some $\varepsilon_2 > 0$ that is *independent* of the $X_i$. Hence, choosing $\varepsilon = \min(\varepsilon_1, \varepsilon_2) > 0$, and replacing $\boldsymbol{\Delta}_C$ by $\widetilde{\boldsymbol{\Delta}}_C \triangleq \{\boldsymbol{q} \in \boldsymbol{\Delta}_C \, ; \, \boldsymbol{q} \geq \varepsilon \mathbf{1}\}$ in the above proof, we attain the same results for the case $f(0+) = \infty$.

**Remark 1.** In addition to our fairness problem formulation (6) being different from that in [AAW+20a], we note that our proof techniques are distinct. Indeed, the proofs in [AAW+20a] develop several techniques since they are based only on Sion's minimax theorem, precisely because a generalized minimax result such as Theorem 1 is inapplicable in the setup of [AAW+20a]. The reason behind this inapplicability is that the ambient Banach space $\mathcal{C}(\mathcal{X}, \mathbb{R}^C)$ is *not* reflexive when $\mathcal{X}$ is infinite, e.g., when $\mathcal{X} = \mathbb{R}^d$ as is assumed in [AAW+20a], whereas it is reflexive in our case as we consider a finite set of samples $\mathbb{X} \subset \mathcal{X}$.

### A.1.2   Algorithm 1: derivation of the ADMM iterations

ADMM is applicable to problems taking the form

$$\begin{aligned} \underset{(\boldsymbol{V},\boldsymbol{\lambda}) \in \mathbb{R}^V \times \mathbb{R}^K}{\text{minimize}} \quad & F(\boldsymbol{V}) + \psi(\boldsymbol{\lambda}) \\ \text{subject to} \quad & \boldsymbol{A}\boldsymbol{V} + \boldsymbol{B}\boldsymbol{\lambda} = \boldsymbol{m}, \end{aligned} \tag{A.18}$$

where $F : \mathbb{R}^V \to \mathbb{R} \cup \{\infty\}$ and $\psi : \mathbb{R}^K \to \mathbb{R} \cup \{\infty\}$ are closed proper convex functions, and $\boldsymbol{A} \in \mathbb{R}^{U \times V}, \boldsymbol{B} \in \mathbb{R}^{U \times K}$, and $\boldsymbol{m} \in \mathbb{R}^U$ are fixed.

We rewrite the convex problem (8) into the ADMM form (A.18) as follows. With the samples $X_1, \cdots, X_N \overset{\mathrm{i.i.d.}}{\sim} P_X$ fixed, we denote the following fixed vectors and matrices: for each $i \in [N]$, set

$$\boldsymbol{p}_i \triangleq \boldsymbol{h}^{\mathrm{base}}(X_i) \in \boldsymbol{\Delta}_C^+ = \{\boldsymbol{q} \in \boldsymbol{\Delta}_C \, ; \, \boldsymbol{q} > \boldsymbol{0}\}, \tag{A.19}$$

$$\boldsymbol{G}_i \triangleq \boldsymbol{G}(X_i) \in \mathbb{R}^{K \times C}. \tag{A.20}$$

We introduce a variable $\boldsymbol{V} \triangleq (\boldsymbol{v}_i)_{i \in [N]} \in \mathbb{R}^{NC}$ (with components $\boldsymbol{v}_i \in \mathbb{R}^C$), and consider the objective functions

$$F(\boldsymbol{V}) \triangleq \frac{1}{N} \sum_{i \in [N]} D_f^{\mathrm{conj}}(\boldsymbol{v}_i, \boldsymbol{p}_i) + \frac{\zeta}{2} \|\boldsymbol{V}\|_2^2, \tag{A.21}$$

$$\psi(\boldsymbol{\lambda}) \triangleq \mathbb{I}_{\mathbb{R}_+^K}(\boldsymbol{\lambda}) + \frac{\zeta}{2} \|\boldsymbol{\lambda}\|_2^2. \tag{A.22}$$

Then, setting[2]

$$\boldsymbol{A} = \frac{1}{\sqrt{N}} \boldsymbol{I}_{NC}, \quad \boldsymbol{B} = \frac{1}{\sqrt{N}} (\boldsymbol{G}_i)_{i \in [N]}^T, \text{ and } \boldsymbol{m} = \boldsymbol{0}_{NC}, \tag{A.23}$$

our finite-sample problem (8) takes the ADMM form (A.18).

In addition, this reparametrization allows us to parallelize the ADMM iterations, which we briefly review next. One starts with forming the augmented Lagrangian for problem (A.18), $L_\rho : \mathbb{R}^V \times$

---

[2]The prefactor $1/\sqrt{N}$ is unnecessary since $\boldsymbol{m} = \boldsymbol{0}$, but we introduce it to simplify the ensuing expressions.

$\mathbb{R}^K \times \mathbb{R}^U \to \mathbb{R} \cup \{\infty\}$, where $\rho > 0$ is a fixed *penalty parameter* and $\boldsymbol{U} \in \mathbb{R}^U$ denotes a *dual variable*, by

$$L_\rho(\boldsymbol{V}, \boldsymbol{\lambda}, \boldsymbol{U}) \triangleq F(\boldsymbol{V}) + \psi(\boldsymbol{\lambda}) + \boldsymbol{U}^T (\boldsymbol{A}\boldsymbol{V} + \boldsymbol{B}\boldsymbol{\lambda} - \boldsymbol{m}) + \frac{\rho}{2} \|\boldsymbol{A}\boldsymbol{V} + \boldsymbol{B}\boldsymbol{\lambda} - \boldsymbol{m}\|_2^2. \qquad \text{(A.24)}$$

The ADMM iterations then repeatedly update the triplet after the $t$-th iteration $(\boldsymbol{V}^{(t)}, \boldsymbol{\lambda}^{(t)}, \boldsymbol{U}^{(t)})$ into a triplet $(\boldsymbol{V}^{(t+1)}, \boldsymbol{\lambda}^{(t+1)}, \boldsymbol{U}^{(t+1)})$ that is given by

$$\boldsymbol{V}^{(t+1)} \in \operatorname*{argmin}_{\boldsymbol{V} \in \mathbb{R}^V} L_\rho(\boldsymbol{V}, \boldsymbol{\lambda}^{(t)}, \boldsymbol{U}^{(t)}), \qquad \text{(A.25)}$$

$$\boldsymbol{\lambda}^{(t+1)} \in \operatorname*{argmin}_{\boldsymbol{\lambda} \in \mathbb{R}^V} L_\rho(\boldsymbol{V}^{(t+1)}, \boldsymbol{\lambda}, \boldsymbol{U}^{(t)}), \qquad \text{(A.26)}$$

$$\boldsymbol{U}^{(t+1)} = \boldsymbol{U}^{(t)} + \rho \cdot \left( \boldsymbol{A}\boldsymbol{V}^{(t+1)} + \boldsymbol{B}\boldsymbol{\lambda}^{(t+1)} \right). \qquad \text{(A.27)}$$

We next instantiate the ADMM iterations to our problem, and we note that we will consider the scaled dual variable $\boldsymbol{W} = \sqrt{N}\boldsymbol{U}$.

In our case, the augmented Lagrangian splits into non-interacting components along the $\boldsymbol{v}_i$. This splitting allows parallelizability of the $\boldsymbol{V}$-update step, which is the most computationally intensive step. Consider a conforming decomposition $\boldsymbol{U} = (\boldsymbol{u}_i)_{i \in [N]}$ for $\boldsymbol{u}_i \in \mathbb{R}^C$, and let $\boldsymbol{W} = \sqrt{N}\boldsymbol{U}$. With some algebra, one can show that the ADMM iterations for the ADMM problem specified by (A.21)–(A.23) are expressible by[3]

$$\boldsymbol{v}_i^{(t+1)} = \operatorname*{argmin}_{\boldsymbol{v} \in \mathbb{R}^C} D_f^{\text{conj}}(\boldsymbol{v}, \boldsymbol{p}_i) + \mathcal{R}_i^{(t)}(\boldsymbol{v}), \qquad i \in [N], \qquad \text{(A.28)}$$

$$\boldsymbol{\lambda}^{(t+1)} = \operatorname*{argmin}_{\boldsymbol{\lambda} \in \mathbb{R}_+^K} \boldsymbol{\lambda}^T \boldsymbol{Q} \boldsymbol{\lambda} + \boldsymbol{q}^{(t)T} \boldsymbol{\lambda}, \qquad \text{(A.29)}$$

$$\boldsymbol{w}_i^{(t+1)} = \boldsymbol{w}_i^{(t)} + \rho \cdot \left( \boldsymbol{v}_i^{(t+1)} + \boldsymbol{G}_i^T \boldsymbol{\lambda}^{(t+1)} \right), \qquad i \in [N], \qquad \text{(A.30)}$$

where $\mathcal{R}_i^{(t)} : \mathbb{R}^C \to \mathbb{R}$ is the quadratic form

$$\mathcal{R}_i^{(t)}(\boldsymbol{v}) \triangleq \frac{\rho + \zeta}{2} \|\boldsymbol{v}\|_2^2 + \left( \boldsymbol{w}_i^{(t)} + \rho \boldsymbol{G}_i^T \boldsymbol{\lambda}^{(t)} \right)^T \boldsymbol{v}, \qquad \text{(A.31)}$$

and the fixed matrix $\boldsymbol{Q} \in \mathbb{R}^{K \times K}$ and vectors $\boldsymbol{q}^{(t)} \in \mathbb{R}^K$ are given by

$$\boldsymbol{Q} \triangleq \frac{\zeta}{2} \boldsymbol{I}_K + \frac{\rho}{2N} \sum_{i \in [N]} \boldsymbol{G}_i \boldsymbol{G}_i^T, \qquad \text{(A.32)}$$

$$\boldsymbol{q}^{(t)} \triangleq \frac{1}{N} \sum_{i \in [N]} \boldsymbol{G}_i \cdot \left( \boldsymbol{w}_i^{(t)} + \boldsymbol{v}_i^{(t+1)} \right). \qquad \text{(A.33)}$$

Note that both the first (A.28) and last (A.30) steps can be carried out for each sample $i \in [N]$ in parallel.

### A.1.3 The inner iterations: minimizing the convex conjugate of $f$-divergence

Only updating the primal-variable $\boldsymbol{v}_i$ in Algorithm 1, i.e., solving

$$\min_{\boldsymbol{v} \in \mathbb{R}^C} D_f^{\text{conj}}(\boldsymbol{v}, \boldsymbol{p}) + \xi \|\boldsymbol{v}\|_2^2 + \boldsymbol{a}^T \boldsymbol{v} \qquad \text{(A.34)}$$

for fixed $(\boldsymbol{p}, \xi, \boldsymbol{a}) \in \boldsymbol{\Delta}_C^+ \times (0, \infty) \times \mathbb{R}^C$, is a nonstandard task. We propose in this section two approaches to execute this step, which aim at re-expressing the required minimization as either a fixed-point or a root-finding problem. In more detail, if one has access to an explicit formula for the

---

[3]Note also that in these specific ADMM iterations, unlike in the general ADMM iterations, we write "$= \operatorname{argmin}$" as opposed to "$\in \operatorname{argmin}$" since strict convexity and coercivity guarantee that a unique minimizer exists (see [CST17] for a case where $\operatorname{argmin}$ is empty). Also, we write here $\boldsymbol{q}^{(t)T}$ instead of $\left( \boldsymbol{q}^{(t)} \right)^T$ for readability.

gradient of $D_f^{\mathrm{conj}}$, then one can transform (A.34) into a fixed-point equation. This case applies for the KL-divergence, for which $\nabla D_{\mathsf{KL}}^{\mathrm{conj}}$ is the softmax function (Appendix A.1.3.1). Furthermore, for the convergence of the fixed-point iterations, we derive an improved Lipschitz constant for the softmax function in Appendix A.1.4. On the other hand, if one does not have a tractable formula for $\nabla D_f^{\mathrm{conj}}$, we propose the reduction provided in Lemma 1, whose proof is provided in Appendix A.1.3.2. We specialize the reduction provided by Lemma 1 to the cross-entropy case in Appendix A.1.3.3. Finally, we include in Appendix A.1.3.4 a general formula for $\nabla D_f^{\mathrm{conj}}$ that can be used for the $\boldsymbol{v}_i$-update step for a general $f$-divergence, and we also utilize it in Appendices A.1.5–A.1.7 to prove the convergence rate of Algorithm 1 stated in Theorems 3–3.

### A.1.3.1   Primal update for KL-divergence

Consider the case when the $f$-divergence of choice is the KL-divergence, i.e., $f(t) = t \log t$. Then, the convex conjugate $D_f^{\mathrm{conj}}$ is given by the log-sum-exp function [DV75], namely, for $(\boldsymbol{p}, \boldsymbol{v}) \in \boldsymbol{\Delta}_C^+ \times \mathbb{R}^C$ we have

$$D_f^{\mathrm{conj}}(\boldsymbol{v}, \boldsymbol{p}) = \log \sum_{c \in [C]} p_c e^{v_c}. \tag{A.35}$$

Thus, the first step in a given ADMM iteration, as in (A.28) (see also the beginning of the for-loop in Algorithm 1), amounts to solving

$$\min_{\boldsymbol{v} \in \mathbb{R}^C} \ \log \sum_{c \in [C]} p_c e^{v_c} + \xi \|\boldsymbol{v}\|_2^2 + \boldsymbol{a}^T \boldsymbol{v} \tag{A.36}$$

for $\xi \triangleq \frac{\rho + \zeta}{2} > 0$ and some fixed vectors $(\boldsymbol{p}, \boldsymbol{a}) \in \boldsymbol{\Delta}_C^+ \times \mathbb{R}^C$; see (A.19), (A.28) and (A.31) for explicit expressions. The problem (A.36) is strictly convex. Further, we may recast this problem, via introducing the variable $\boldsymbol{z} \in \mathbb{R}^C$ by $z_c \triangleq v_c + \log p_c$, as

$$\min_{\boldsymbol{z} \in \mathbb{R}^C} \ \log \sum_{c \in [C]} e^{z_c} + \xi \|\boldsymbol{z}\|_2^2 + \boldsymbol{b}^T \boldsymbol{z}, \tag{A.37}$$

where $b_c = a_c - 2\xi \log p_c$ is fixed. To solve this latter problem, it suffices to find a zero of the gradient, which is given by

$$\nabla_{\boldsymbol{z}} \left( \log \sum_{c \in [C]} e^{z_c} + \xi \|\boldsymbol{z}\|_2^2 + \boldsymbol{b}^T \boldsymbol{z} \right) = \sigma(\boldsymbol{z}) + 2\xi \boldsymbol{z} + \boldsymbol{b} \tag{A.38}$$

where $\sigma : \mathbb{R}^C \to \boldsymbol{\Delta}_C^+$ denotes the softmax function $\sigma(\boldsymbol{z}) \triangleq \left( \frac{e^{z_{c'}}}{\sum_{c \in [C]} e^{z_c}} \right)_{c' \in [C]}$. Thus, we arrive at the fixed-point problem $\theta(\boldsymbol{z}) = \boldsymbol{z}$ for the function

$$\theta(\boldsymbol{z}) \triangleq -\frac{1}{2\xi} \left( \sigma(\boldsymbol{z}) + \boldsymbol{b} \right). \tag{A.39}$$

We solve $\theta(\boldsymbol{z}) = \boldsymbol{z}$ using a fixed-point-iteration method, i.e., with some initial $\boldsymbol{z}_0$, we iteratively compute the compositions $\theta^{(m)}(\boldsymbol{z}_0)$ for $m \in \mathbb{N}$. This procedure is summarized in Algorithm A.1.

The exponentially-fast convergence of Algorithm A.1 is guaranteed in view of Lipschitzness of $\theta$ as defined in (A.39). Indeed, it is known that the softmax function is 1-Lipschitz (see, e.g., [GP17, Prop. 4]); we improve this Lipschitz constant to $1/2$ in Appendix A.1.4. This improvement yields a better guarantee on the convergence speed of `FairProjection`. Indeed, as a lower value of the ADMM penalty $\rho$ correlates with a faster convergence, lowering the Lipschitz constant of the softmax function allows us to speed up `FairProjection` by choosing $\rho > \frac{1}{2} - \zeta$ instead of $\rho > 1 - \zeta$.

**Algorithm A.1 :** $\underset{\boldsymbol{v} \in \mathbb{R}^C}{\operatorname{argmin}} D_{\mathsf{KL}}^{\mathsf{conj}}(\boldsymbol{v}, \boldsymbol{p}) + \xi \|\boldsymbol{v}\|_2^2 + \boldsymbol{a}^T \boldsymbol{v}$

---

**Input:** $\xi > 0, \boldsymbol{p} \in \boldsymbol{\Delta}_C^+, \boldsymbol{a}, \boldsymbol{v} \in \mathbb{R}^C.$

$z_c \leftarrow v_c + \log p_c$ $\hfill c \in [C]$

$b_c \leftarrow a_c - 2\xi \log p_c$ $\hfill c \in [C]$

**repeat**

$\quad \boldsymbol{z} \leftarrow -\frac{1}{2\xi}\left(\sigma(\boldsymbol{z}) + \boldsymbol{b}\right)$

**until** convergence

**Output:** $v_c \triangleq z_c - \log p_c$ $\hfill c \in [C]$

---

### A.1.3.2 Proof of Lemma 1: primal update for general $f$-divergences

The lemma follows by the following sequence of steps:

$$\min_{\boldsymbol{v} \in \mathbb{R}^C} D_f^{\mathsf{conj}}(\boldsymbol{v}, \boldsymbol{p}) + \xi \|\boldsymbol{v}\|_2^2 + \boldsymbol{a}^T \boldsymbol{v} \stackrel{\text{(I)}}{=} \min_{\boldsymbol{v} \in \mathbb{R}^C} \max_{\boldsymbol{q} \in \boldsymbol{\Delta}_C} \boldsymbol{q}^T \boldsymbol{v} - D_f(\boldsymbol{q} \,\|\, \boldsymbol{p}) + \boldsymbol{a}^T \boldsymbol{v} + \xi \|\boldsymbol{v}\|_2^2 \tag{A.40}$$

$$\stackrel{\text{(II)}}{=} \max_{\boldsymbol{q} \in \boldsymbol{\Delta}_C} \min_{\boldsymbol{v} \in \mathbb{R}^C} \boldsymbol{q}^T \boldsymbol{v} - D_f(\boldsymbol{q} \,\|\, \boldsymbol{p}) + \boldsymbol{a}^T \boldsymbol{v} + \xi \|\boldsymbol{v}\|_2^2 \tag{A.41}$$

$$\stackrel{\text{(III)}}{=} \max_{\boldsymbol{q} \in \boldsymbol{\Delta}_C} -D_f(\boldsymbol{q} \,\|\, \boldsymbol{p}) - \frac{1}{4\xi}\|\boldsymbol{a} + \boldsymbol{q}\|_2^2 \tag{A.42}$$

$$= -\min_{\boldsymbol{q} \in \boldsymbol{\Delta}_C} D_f(\boldsymbol{q} \,\|\, \boldsymbol{p}) + \frac{1}{4\xi}\|\boldsymbol{a} + \boldsymbol{q}\|_2^2 \tag{A.43}$$

$$= -\min_{\boldsymbol{q} \in \mathbb{R}_+^C} \sup_{\theta \in \mathbb{R}} D_f(\boldsymbol{q} \,\|\, \boldsymbol{p}) + \frac{1}{4\xi}\|\boldsymbol{a} + \boldsymbol{q}\|_2^2 + \theta \cdot \left(\mathbf{1}^T \boldsymbol{q} - 1\right) \tag{A.44}$$

$$\stackrel{\text{(IV)}}{=} -\sup_{\theta \in \mathbb{R}} \min_{\boldsymbol{q} \in \mathbb{R}_+^C} D_f(\boldsymbol{q} \,\|\, \boldsymbol{p}) + \frac{1}{4\xi}\|\boldsymbol{a} + \boldsymbol{q}\|_2^2 + \theta \cdot \left(\mathbf{1}^T \boldsymbol{q} - 1\right) \tag{A.45}$$

$$\stackrel{\text{(V)}}{=} -\sup_{\theta \in \mathbb{R}} -\theta + \sum_{c \in [C]} \min_{q_c \geq 0} p_c f\left(\frac{q_c}{p_c}\right) + \frac{1}{4\xi}(a_c + q_c)^2 + \theta q_c, \tag{A.46}$$

where (I) holds by definition of $D_f^{\mathsf{conj}}$ (see (3)), (II) by Sion's minimax theorem, (III) since the inner minimization occurs at $\boldsymbol{v} = -\frac{1}{2\xi}(\boldsymbol{q} + \boldsymbol{a})$, (IV) by generalized minimax theorems [see, e.g., Chapter VI, Proposition 2.2 in ET99a] (restated as Theorem 1 herein for convenience), and (V) by separability.

### A.1.3.3 Primal update for cross-entropy

In the cross-entropy (CE) case, i.e., $f(t) = -\log t$, instead of using an explicit formula for $D_f^{\mathsf{conj}}$ (which would yield unwieldy expressions), we utilize the reduction shown in Lemma 1. Thus, we have the equality

$$\min_{\boldsymbol{v} \in \mathbb{R}^C} D_f^{\mathsf{conj}}(\boldsymbol{v}, \boldsymbol{p}) + \xi \|\boldsymbol{v}\|_2^2 + \boldsymbol{a}^T \boldsymbol{v} = -\sup_{\theta \in \mathbb{R}} -\theta + \sum_{c \in [C]} \min_{q_c \geq 0} p_c f\left(\frac{q_c}{p_c}\right) + \frac{1}{4\xi}(a_c + q_c)^2 + \theta q_c. \tag{A.47}$$

As per (A.47), we focus next on solving the inner single-variable minimization

$$\min_{q \geq 0} -p \log q + \frac{1}{4\xi}(a + q)^2 + \theta q. \tag{A.48}$$

It is easily seen that the solution to this minimization is the unique point making the objective's derivative vanish, i.e., it is $q^\star \in (0, \infty)$ for which

$$-\frac{p}{q^\star} + \frac{q^\star}{2\xi} + \theta + \frac{a}{2\xi} = 0. \tag{A.49}$$

**Algorithm A.2 :** $\underset{\boldsymbol{v} \in \mathbb{R}^C}{\operatorname{argmin}} \; D_{\mathsf{CE}}^{\mathrm{conj}}(\boldsymbol{v}, \boldsymbol{p}) + \xi \|\boldsymbol{v}\|_2^2 + \boldsymbol{a}^T \boldsymbol{v}$

---

**Input:** $\xi > 0$, $z \in \mathbb{R}$, $\boldsymbol{p} \in \boldsymbol{\Delta}_C^+$, $\boldsymbol{a} \in \mathbb{R}^C$.

**repeat**

$$g(z) \leftarrow -1 + \sum_{c \in [C]} \sqrt{\left(z + \frac{a_c}{2}\right)^2 + 2p_c\xi} - \left(z + \frac{a_c}{2}\right)$$

$$g'(z) \leftarrow -C + \sum_{c \in [C]} \frac{2z + a_c}{\sqrt{\left(z + \frac{a_c}{2}\right)^2 + 2p_c\xi}}$$

$$z \leftarrow z - \frac{g(z)}{g'(z)}$$

**until** convergence

**Output:** $v_c \triangleq \dfrac{1}{2\xi}\left(z - \dfrac{a_c}{2} - \sqrt{\left(z + \dfrac{a_c}{2}\right)^2 + 2p_c\xi}\right)$

---

This is easily solvable as a quadratic, yielding

$$q^\star = \sqrt{\left(\theta\xi + \frac{a}{2}\right)^2 + 2p\xi} - \left(\theta\xi + \frac{a}{2}\right). \tag{A.50}$$

Therefore, solving (A.47) amounts to finding the constant $\theta \in \mathbb{R}$ that yields a probability vector $\boldsymbol{q} \in \boldsymbol{\Delta}_C$, where

$$q_c \triangleq \sqrt{\left(\theta\xi + \frac{a_c}{2}\right)^2 + 2p_c\xi} - \left(\theta\xi + \frac{a_c}{2}\right). \tag{A.51}$$

Consider the function

$$g(z) \triangleq -1 + \sum_{c \in [C]} \sqrt{\left(z + \frac{a_c}{2}\right)^2 + 2p_c\xi} - \left(z + \frac{a_c}{2}\right), \tag{A.52}$$

so we simply are looking for a root of $g$ (then set $\theta = z/\xi$ and $\boldsymbol{v} = -\frac{1}{2\xi}(\boldsymbol{q} + \boldsymbol{a})$). This can be efficiently accomplished via Newton's method. Namely, we compute

$$g'(z) = -C + \sum_{c \in [C]} \frac{2z + a_c}{\sqrt{\left(z + \frac{a_c}{2}\right)^2 + 2p_c\xi}}, \tag{A.53}$$

then, starting from $z^{(0)}$, we form the sequence

$$z^{(t+1)} \triangleq z^{(t)} - \frac{g\left(z^{(t)}\right)}{g'\left(z^{(t)}\right)}. \tag{A.54}$$

This procedure is summarized in Algorithm A.2.

### A.1.3.4   On the gradient of the convex conjugate of $f$-divergence

The following general result on the differentiability of $D_f^{\mathrm{conj}}$ can be used to carry out the $\boldsymbol{v}_i$-update step for a general $f$-divergence, and it will also be useful in Appendices A.1.5–A.1.7 for proving the convergence rate of Algorithm 1 as stated in Theorems 3–3.

**Lemma 2.** *Suppose* $f : (0, \infty) \to \mathbb{R}$ *is strictly convex. For any fixed* $\boldsymbol{p} \in \boldsymbol{\Delta}_C^+$, *the function* $\boldsymbol{v} \mapsto D_f^{\mathrm{conj}}(\boldsymbol{v}, \boldsymbol{p})$ *is differentiable, and its gradient is given by*

$$\nabla_{\boldsymbol{v}} D_f^{\mathrm{conj}}(\boldsymbol{v}, \boldsymbol{p}) = \boldsymbol{q}_f^{\mathrm{conj}}(\boldsymbol{v}, \boldsymbol{p}) \in \boldsymbol{\Delta}_C, \tag{A.55}$$

*where*

$$\boldsymbol{q}_f^{\mathrm{conj}}(\boldsymbol{v}, \boldsymbol{p}) \triangleq \underset{\boldsymbol{q} \in \boldsymbol{\Delta}_C}{\operatorname{argmin}} \; D_f(\boldsymbol{q} \,\|\, \boldsymbol{p}) - \boldsymbol{v}^T \boldsymbol{q}. \tag{A.56}$$

*Proof.* From [Roc09, Proposition 11.3], since $\boldsymbol{q} \mapsto D_f(\boldsymbol{q} \,\|\, \boldsymbol{p})$ is a lower semicontinuous proper convex function, the subgradient of its convex conjugate $\boldsymbol{v} \mapsto D_f^{\mathrm{conj}}(\boldsymbol{v}, \boldsymbol{p})$ is given by

$$\partial_{\boldsymbol{v}} D_f^{\mathrm{conj}}(\boldsymbol{v}, \boldsymbol{p}) = \operatorname*{argmin}_{\boldsymbol{q} \in \boldsymbol{\Delta}_C} D_f(\boldsymbol{q} \,\|\, \boldsymbol{p}) - \boldsymbol{v}^T \boldsymbol{q}. \tag{A.57}$$

Recall also that a function is differentiable at a point if and only if its subgradient there consists of a singleton [BFG87]. Thus, it only remains to show that the right-hand side in (A.57) is a singleton. For this, we note that $\boldsymbol{q} \mapsto D_f(\boldsymbol{q} \,\|\, \boldsymbol{p}) - \boldsymbol{v}^T \boldsymbol{q}$ is lower semicontinuous and strictly convex, and $\boldsymbol{\Delta}_C$ is compact. $\qquad\square$

### A.1.4 ½-Lipschitzness of the Softmax Function

As stated in Section 4 and Appendix A.1.3.1, the convergence speed of the inner iteration (the $\boldsymbol{v}_i$ update step) of `FairProjection` can be guaranteed to be faster if the Lipschitz constant of the softmax function is lowered from 1 (which is proved in [GP17, Prop. 4]). By Lipschitzness here, we mean $\ell_2$-norm Lipschitzness. We prove the following proposition in this appendix.

**Proposition 1.** *For any $n \in \mathbb{N}$, the softmax function $\sigma(\boldsymbol{z}) \triangleq \left( \frac{e^{z_j}}{\sum_{i=1}^n e^{z_i}} \right)_{j \in [n]}$ is $\frac{1}{2}$-Lipschitz.*

We will need the following result.

**Lemma 3** (Theorem 2.1.6 in [Nes04]). *A twice continuously differentiable function $f : \mathbb{R}^n \to \mathbb{R}$ is convex and has an L-Lipschitz continuous gradient if and only if its Hessian is positive semidefinite with maximal eigenvalue at most L.*

Since the softmax function is the gradient of the log-sum-exp function, and since the spectral norm is upper bounded by the Frobenius norm, it suffices to upper bound the Frobenius norm of the Jacobian of $\sigma$ by $1/2$. Suppose that $\sigma$ is operating on $n$ symbols. Consider the sum of powers functions $s_k(\boldsymbol{x}) \triangleq \sum_{i \in [n]} x_i^k$ for $\boldsymbol{x} \in \mathbb{R}^n$. For any $\boldsymbol{v} \in \mathbb{R}^n$, denoting $\boldsymbol{x} = \sigma(\boldsymbol{v})$, the square of the Frobenius norm of the Jacobian of $\sigma$ at $\boldsymbol{v}$ is given by

$$w(\boldsymbol{x}) \triangleq s_2(\boldsymbol{x})^2 + s_2(\boldsymbol{x}) - 2s_3(\boldsymbol{x}). \tag{A.58}$$

We show that $w(\boldsymbol{x}) \leq \frac{1}{4}$ for any $n \in \mathbb{N}$ and $\boldsymbol{x} \in \boldsymbol{\Delta}_n$.

The approach we take is via reduction to the case $n \leq 3$, which one can directly verify. Namely, assuming, without loss of generality, that $x_1 \leq x_2 \leq \cdots \leq x_n$, we show that if $x_1 + x_2 \leq 1/2$ then $w(\boldsymbol{y}) \geq w(\boldsymbol{x})$ where $\boldsymbol{y} \in \boldsymbol{\Delta}_{n-1}$ is given by $\boldsymbol{y} = (x_1 + x_2, x_3, \cdots, x_n)$. Note that if $n \geq 4$ then we must have $x_1 + x_2 \leq 1/2$, because $x_1 + x_2 \leq x_3 + x_4$ and $x_1 + x_2 + x_3 + x_4 \leq 1$. Thus, we will have reduced the problem from an $n \geq 4$ to $n - 1$, which iteratively reduces the problem to $n \leq 3$. Fix $n \geq 4$.

Denote $\boldsymbol{z} = (x_3, \cdots, x_n)$. A direct computation yields that

$$w(\boldsymbol{y}) - w(\boldsymbol{x}) = 2x_1 x_2 \cdot (2s_2(\boldsymbol{z}) + g(x_1, x_2)) \tag{A.59}$$

with the quadratic

$$g(a, b) \triangleq 2a^2 + 2b^2 + 2ab - 3a - 3b + 1. \tag{A.60}$$

By assumption, $x_i \geq \max(x_1, x_2)$ for each $i \geq 3$, so $2s_2(\boldsymbol{z}) \geq (n-2)x_1^2 + (n-2)x_2^2 \geq x_1^2 + x_2^2$. Then,

$$w(\boldsymbol{y}) - w(\boldsymbol{x}) \geq 2x_1 x_2 \cdot h(x_1, x_2) \tag{A.61}$$

with

$$h(a, b) \triangleq 3a^2 + 3b^2 + 2ab - 3a - 3b + 1. \tag{A.62}$$

Now, we show that $h$ is nonnegative for every $a, b \geq 0$ with $a + b \leq 1/2$. With $c = a + b$, we may write

$$h(a, b) = 3c^2 - (3 + 4a)c + 4a^2 + 1. \tag{A.63}$$

This quadratic in $c$ has its vertex at $c_{\min} = (3 + 4a)/6$. As $a \geq 0$, $c_{\min} \geq 1/2$. As $a + b \leq 1/2$, we see that the minimum of $h$ is attained for $c = 1/2$. Substituting $b = 1/2 - a$, we obtain

$$h(a, b) = \left( 2a - \frac{1}{2} \right)^2, \tag{A.64}$$

which is nonnegative, as desired.

### A.1.5 Convergence rate of Algorithm 1: proof of Theorem 3

We recall a general result on the R-linear convergence rate for ADMM, which corresponds to case 1 in scenario 1 in [DY16]; see Tables 1 and 2 therein. Recall that a sequence $\{z^{(t)}\}_{t\in\mathbb{N}}$ is said to converge R-linearly to $z^\star$ if there is a constant $\eta \in (0,1)$ and a sequence $\{\beta^{(t)}\}_{t\in\mathbb{N}}$ such that $\|z^{(t)} - z^\star\| \le \beta^{(t)}$ and $\sup_t \left(\beta^{(t+1)}/\beta^{(t)}\right) \le \eta$. In particular, one has exponentially small errors:

$$\|z^{(t)} - z^\star\| \le \beta^{(0)} \cdot \eta^t. \tag{A.65}$$

The following theorem is used in our proof of Theorem 3.

**Theorem 2** ([DY16]). *Suppose that problem* (A.18) *has a saddle point, $F$ is strongly convex and differentiable with Lipschitz-continuous gradient, $\boldsymbol{A}$ has full row-rank, and $\boldsymbol{B}$ has full column-rank. Then, the ADMM iterations* (A.25)–(A.27) *converge R-linearly to a global optimizer.*

In Appendix A.1.2, we show that the dual (8) of our fairness optimization problem (6) can be written in the ADMM general form (A.18) with the choices

$$F(\boldsymbol{V}) = \frac{1}{N} \sum_{i\in[N]} D_f^{\text{conj}}(\boldsymbol{v}_i, \boldsymbol{p}_i) + \frac{\zeta}{2} \|\boldsymbol{V}\|_2^2 \tag{A.66}$$

and

$$\boldsymbol{A} = \frac{1}{\sqrt{N}}\boldsymbol{I}_{NC}, \quad \boldsymbol{B} = \frac{1}{\sqrt{N}}(\boldsymbol{G}_i)_{i\in[N]}^T. \tag{A.67}$$

Recall from Theorem 2 (see also the proof in Appendix A.1.1) that our problem (8) has a saddle point. Further, the function $F : \mathbb{R}^{NC} \to \mathbb{R}$ is $\zeta$-strongly convex and differentiable. Indeed, each $\boldsymbol{v} \mapsto D_f^{\text{conj}}(\boldsymbol{v}, \boldsymbol{p}_i)$ is convex, and the term $\frac{\zeta}{2}\|\boldsymbol{V}\|_2^2$ is $\zeta$-strongly convex, so $F$ is $\zeta$-strongly convex too. In addition, by the formula for $\nabla D_f^{\text{conj}}$ in Lemma 2, the gradient of $F$ is

$$\nabla F(\boldsymbol{V}) = \frac{1}{N}\boldsymbol{q}_f^{\text{conj}}(\boldsymbol{V}) + \zeta\boldsymbol{V}, \tag{A.68}$$

where

$$\boldsymbol{q}_f^{\text{conj}}(\boldsymbol{V}) \triangleq \left(\boldsymbol{q}_f^{\text{conj}}(\boldsymbol{v}_i, \boldsymbol{p}_i)\right)_{i\in[N]}, \tag{A.69}$$

with $\boldsymbol{q}_f^{\text{conj}}(\boldsymbol{v}_i)$ as defined in (A.56).

In the KL-divergence case, i.e., $f(t) = t\log t$, the gradient of $D_f^{\text{conj}}$ is given by the softmax function (see Appendix A.1.3.1)

$$\boldsymbol{q}_f^{\text{conj}}(\boldsymbol{v}, \boldsymbol{p}) = \sigma(\boldsymbol{v} + \log \boldsymbol{p}) = \left(\frac{p_c e^{v_c}}{\sum_{c'\in[C]} p_{c'} e^{v_{c'}}}\right)_{c\in[C]}. \tag{A.70}$$

Therefore, we have that

$$\nabla F(\boldsymbol{V}) = \frac{1}{N}\left(\sigma(\boldsymbol{v}_i + \log \boldsymbol{p}_i)\right)_{i\in[N]} + \zeta\boldsymbol{V}. \tag{A.71}$$

By Proposition 1, the softmax function $\sigma$ is $\frac{1}{2}$-Lipschitz. Hence, $\nabla F$ is $\left(\frac{1}{2N} + \zeta\right)$-Lipschitz.

Therefore, the general ADMM convergence rate in Theorem 2 yields that there is a constant $r > 0$ such that

$$\left\|\boldsymbol{\lambda}_{\zeta,N}^{(t)} - \boldsymbol{\lambda}_{\zeta,N}^\star\right\|_2 \le \beta \cdot e^{-rt} \tag{A.72}$$

where $\beta \triangleq \left\|\boldsymbol{\lambda}_{\zeta,N}^{(0)} - \boldsymbol{\lambda}_{\zeta,N}^\star\right\|_2$. (Although Theorem 2 guarantees exponentially-fast convergence of $\boldsymbol{\lambda}_{\zeta,N}^{(t)}$ to *a* global optimizer, recall that $\boldsymbol{\lambda}_{\zeta,N}^\star$ is the *unique* optimizer of (8), as Theorem 2 shows.)

Finally, it remains to bound the distance between $\boldsymbol{h}^{\text{opt},N}$ and the output classifier $\boldsymbol{h}^{(t)}$ after the $t$-th iteration of Algorithm 1. Note that $\phi(u) = (f')^{-1}(u) = e^{u-1}$, so $\gamma$ may be obtained explicitly, and equation (7) becomes

$$h_{c'}^{\text{opt},N}(x) = \frac{h_{c'}^{\text{base}}(x) \cdot e^{v_{c'}(x;\boldsymbol{\lambda}_{\zeta,N}^\star)}}{\sum_{c\in[C]} h_c^{\text{base}}(x) \cdot e^{v_c(x;\boldsymbol{\lambda}_{\zeta,N}^\star)}}. \tag{A.73}$$

Thus, using $\boldsymbol{\lambda}^{(t)} \triangleq \boldsymbol{\lambda}_{\zeta,N}^{(t)}$ in place of $\boldsymbol{\lambda}_{\zeta,N}^{\star}$, we obtain that the $t$-th classifier obtained by Algorithm 1 is

$$h_{c'}^{(t)}(x) = \frac{h_{c'}^{\text{base}}(x) \cdot e^{v_{c'}(x;\boldsymbol{\lambda}^{(t)})}}{\sum_{c\in[C]} h_c^{\text{base}}(x) \cdot e^{v_c(x;\boldsymbol{\lambda}^{(t)})}}. \tag{A.74}$$

Therefore, we have the ratios

$$\frac{h_{c'}^{(t)}(x)}{h_{c'}^{\text{opt},N}(x)} = \frac{\sum_{c\in[C]} h_c^{\text{base}}(x) e^{v_c(x;\boldsymbol{\lambda}_{\zeta,N}^{\star})}}{\sum_{c\in[C]} h_c^{\text{base}}(x) e^{v_c(x;\boldsymbol{\lambda}^{(t)})}} \cdot \exp\left(v_{c'}(x;\boldsymbol{\lambda}^{(t)}) - v_{c'}(x;\boldsymbol{\lambda}_{\zeta,N}^{\star})\right). \tag{A.75}$$

By definition of $\boldsymbol{v}$, $\boldsymbol{v}(x;\boldsymbol{\lambda}) = -\boldsymbol{G}(x)^T \boldsymbol{\lambda}$. Thus, we obtain from (A.72) and boundedness of $\boldsymbol{G}$ that

$$\left\| \boldsymbol{v}(x;\boldsymbol{\lambda}^{(t)}) - \boldsymbol{v}(x;\boldsymbol{\lambda}_{\zeta,N}^{\star}) \right\|_{\infty} = e^{-\Omega(t)}, \tag{A.76}$$

where the implicit constant is independent of $x$. Applying (A.76) in (A.75), and noting that $e^{\pm e^{-\Omega(t)}} = 1 \pm e^{-\Omega(t)}$ as $t \to \infty$, we conclude that

$$\left| \frac{h_{c'}^{(t)}(x)}{h_{c'}^{\text{opt},N}(x)} - 1 \right| = e^{-\Omega(t)}, \quad c' \in [C], \tag{A.77}$$

uniformly in $x$. We may rewrite (A.77) as

$$\boldsymbol{h}^{(t)}(x) = \boldsymbol{h}^{\text{opt},N}(x) \cdot \left(1 \pm e^{-\Omega(t)}\right), \tag{A.78}$$

which is the desired convergence rate in the theorem statement, and the proof is complete.

### A.1.6 Extension of Theorem 3

Though Theorem 3 is shown for the KL-divergence, the proof directly extends to general $f$-divergences satisfying Assumption 1. In fact, Lipschitz continuity of the gradient of $D_{\text{KL}}^{\text{conj}}$ is the only specific property that we apply to derive the KL-divergence case. For a general $f$-divergence, Lipschitz continuity of $\nabla D_f^{\text{conj}}$ may be derived as follows. Combining Lemmas 1–2 reveals the formula $\nabla_{\boldsymbol{v}} D_f^{\text{conj}}(\boldsymbol{v},\boldsymbol{p}) = (p_c \cdot \phi(\gamma(\boldsymbol{v}) + v_c))_{c\in[C]}$, where $\phi = (f')^{-1}$ and $\gamma(\boldsymbol{v})$ is uniquely defined by $\mathbb{E}_{c\sim\boldsymbol{p}}[\phi(\gamma(\boldsymbol{v}) + v_c)] = 1$, with $\boldsymbol{p} \in \Delta_C^+$ fixed. Since $\phi' = 1/(f'' \circ \phi)$, we have that $\phi$ is locally Lipschitz. From the proof of Theorem 5 in [AAW+20a], we know that $\boldsymbol{v} \mapsto \gamma(\boldsymbol{v})$ is locally Lipschitz. Thus, $\boldsymbol{v} \mapsto \nabla_{\boldsymbol{v}} D_f^{\text{conj}}(\boldsymbol{v},\boldsymbol{p})$ is locally Lipschitz. Further, $\boldsymbol{\lambda} \mapsto \nabla_{\boldsymbol{v}} D_f^{\text{conj}}(\boldsymbol{v}(x;\boldsymbol{\lambda}),\boldsymbol{p})$ is then also locally Lipschitz. Note that we may restrict $\boldsymbol{\lambda}$ *a priori* to be within some finite ball (see Lemma 4). Thus, if, e.g., $X$ is compactly-supported, we would obtain the desired Lipschitzness properties of the gradient of $D_f^{\text{conj}}$, and the proof of Theorem 3 carries through for $D_f$ in place of $D_{\text{KL}}$.

### A.1.7 Convergence rate to the population problem

The following result shows, roughly, that the parameter $\boldsymbol{\lambda}_{N^{-1/2},N}^{(\log N)}$ obtainable from `FairProjection` performs well for the *population* problem for information projection (5).

**Theorem 3.** *Suppose Assumption 1 holds, let $\mathcal{X} = \mathbb{R}^d$, and consider the KL-divergence case. Then, choosing $\zeta = \Theta\left(N^{-1/2}\right)$ and $t = \Omega(\log N)$ we obtain for any $\delta \in (0,1)$ that (see (5))*

$$\Pr\left\{ \mathbb{E}_X\left[ D_{\text{KL}}^{\text{conj}}\left(\boldsymbol{v}\left(X;\boldsymbol{\lambda}_{\zeta,N}^{(t)}\right), \boldsymbol{h}^{\text{base}}(X)\right)\right] > D^{\star} + O\left(\frac{1}{\sqrt{N}}\right) \right\} \le \delta. \tag{A.79}$$

The proof is divided in this appendix into several lemmas. We note first that, in the course of the proof of Theorems 1 and 2 in [AAW+20b], it was shown that at least one minimizer $\boldsymbol{\lambda}^{\star}$ of (5) exists. Further, any such minimizer satisfies the following bound. Denote the constraint function by $\boldsymbol{\mu}(\boldsymbol{h}) \triangleq \mathbb{E}_{P_X}[\boldsymbol{G}\boldsymbol{h}]$. Throughout this proof, we set $\mathcal{X} \triangleq \mathbb{R}^d$.

**Lemma 4.** *Suppose Assumption 1 holds, and fix a strictly feasible classifier $\boldsymbol{h} \in \mathcal{H}$, i.e., $\boldsymbol{\mu}(\boldsymbol{h}) < \boldsymbol{0}$. Every minimizer $\boldsymbol{\lambda}^{\star} \in \mathbb{R}_+^K$ of (5) must satisfy the inequality*

$$\|\boldsymbol{\lambda}^{\star}\|_1 \le \lambda_{\max} \triangleq \frac{D_f\left(\boldsymbol{h} \,\|\, \boldsymbol{h}^{\text{base}} \mid P_X\right)}{\min_{k\in[K]} -\mu_k(\boldsymbol{h})}. \tag{A.80}$$

We note that for the fairness metrics specified in Table 2, one valid choice of a strictly feasible $\boldsymbol{h}$ (i.e., one for which $\boldsymbol{\mu}(\boldsymbol{h}) < \boldsymbol{0}$) is the uniform classifier $\boldsymbol{h}(x) \equiv \frac{1}{C}\boldsymbol{1}$. In any case, we have that $\lambda_{\max} < \infty$ since both $\boldsymbol{h}$ and $\boldsymbol{h}^{\mathrm{base}}$ are assumed to belong to $\mathcal{H}$ and $f$ is continuous over $(0, \infty)$; e.g., one bound on $\lambda_{\max}$ is $\lambda_{\max} \leq \max_{m \leq t \leq M} f(t)/\min_{k \in [K]} -\mu_k(\boldsymbol{h})$ where $m = \inf_{c,x} h_c(x)$ and $M = 1/\inf_{x,c} h_c^{\mathrm{base}}(x)$. We will also need the following constants for the convergence analysis:

$$g_{\mathrm{mean}} \triangleq \mathbb{E}\left[\|\boldsymbol{G}(X)\|_2^2\right], \tag{A.81}$$

$$g_{\mathrm{max}} \triangleq \sup_{x \in \mathcal{X}} \|\boldsymbol{G}(x)\|_2^2. \tag{A.82}$$

Clearly, $g_{\mathrm{mean}} \leq g_{\mathrm{max}}$. By the boundedness of $\boldsymbol{G}$ in the second item in Assumption 1, $g_{\mathrm{max}}$ is finite.

**Remark 2.** Although the results in this paper are stated to hold under Assumption 1, we note that those conditions do not essentially impose any restriction on carrying our `FairProjection` algorithm. Indeed, we focus in this paper on the CE and KL cases, for which $f$ satisfies the imposed conditions. We also note that only boundedness of $\boldsymbol{G}$ is required for Theorem 2, which is true for the fairness metrics in Table 2 in non-degenerate cases (e.g., no empty groups). The condition on $\boldsymbol{h}^{\mathrm{base}}$ being bounded away from zero can be made to hold by perturbing it if necessary with negligible noise. The condition on $\boldsymbol{h}^{\mathrm{base}}$ being continuous is automatically satisfied if its domain is a finite set (as is the case for Theorem 2). Finally, the strict feasibility condition is verified by the uniform classifier.

Now, consider a form of $\ell_2$ regularization of (5):

$$\min_{\boldsymbol{\lambda} \in \mathbb{R}_+^K} \mathbb{E}\left[D_f^{\mathrm{conj}}\left(\boldsymbol{v}(X; \boldsymbol{\lambda}), \boldsymbol{h}^{\mathrm{base}}(X)\right) + \frac{\zeta}{2}\left\|\widetilde{\boldsymbol{G}}(X)^T\boldsymbol{\lambda}\right\|_2^2\right] \tag{A.83}$$

where $\widetilde{\boldsymbol{G}}(x) \triangleq (\boldsymbol{G}(x), \boldsymbol{I}_K) \in \mathbb{R}^{K \times (K+C)}$. We show now that there is a unique minimizer $\boldsymbol{\lambda}_\zeta^\star$ of (A.83).

**Lemma 5.** *Under Assumption 1, there exists a unique minimizer $\boldsymbol{\lambda}_\zeta^\star$ of the regularized problem* (A.83).

*Proof.* Denote the function $A : \mathbb{R}_+^K \to \mathbb{R}$ by

$$A(\boldsymbol{\lambda}) \triangleq \mathbb{E}\left[D_f^{\mathrm{conj}}\left(\boldsymbol{v}(X; \boldsymbol{\lambda}), \boldsymbol{h}^{\mathrm{base}}(X)\right) + \frac{\zeta}{2}\left\|\widetilde{\boldsymbol{G}}(X)^T\boldsymbol{\lambda}\right\|_2^2\right]. \tag{A.84}$$

That the range of $A$ falls within $\mathbb{R}$ follows by Assumption 1, since then the function $x \mapsto D_f^{\mathrm{conj}}(\boldsymbol{v}(x; \boldsymbol{\lambda}), \boldsymbol{h}^{\mathrm{base}}(x))$ is $P_X$-integrable. We will show that $A$ is lower semicontinuous and $\zeta$-strongly convex.

By Lemma 2, $\boldsymbol{v} \mapsto D_f^{\mathrm{conj}}(\boldsymbol{v}, \boldsymbol{p})$ is differentiable for any fixed $\boldsymbol{p} \in \boldsymbol{\Delta}_C^+$, implying that it is also continuous. Thus, $\boldsymbol{\lambda} \mapsto D_f^{\mathrm{conj}}(\boldsymbol{v}(x; \boldsymbol{\lambda}), \boldsymbol{h}^{\mathrm{base}}(x))$ is continuous for each $x \in \mathcal{X}$. Hence, by Fatou's lemma and boundedness of $\boldsymbol{G}$, $A$ is lower semicontinuous.

Next, to show strong convexity, we note that $\boldsymbol{\lambda} \mapsto D_f^{\mathrm{conj}}(\boldsymbol{v}(x; \boldsymbol{\lambda}), \boldsymbol{h}^{\mathrm{base}}(x))$ is convex for each $x \in \mathcal{X}$. Indeed, this function is the supremum of affine functions. Further, the regularization term is $\zeta$-strongly convex, as its Hessian is given by

$$\zeta \cdot \left(\mathbb{E}\left[\widetilde{\boldsymbol{G}}(X)\widetilde{\boldsymbol{G}}(X)^T\right] + \boldsymbol{I}\right), \tag{A.85}$$

which is positive definite with minimal eigenvalue at least $\zeta$.

Now, for each fixed $\theta > 0$, consider the compact set $\Lambda_\theta \triangleq \{\boldsymbol{\lambda} \in \mathbb{R}_+^K ; \|\boldsymbol{\lambda}\|_2^2 \leq \theta\}$. By what we have shown thus far, there is a unique minimizer $\boldsymbol{\lambda}_\theta$ of $A$ over $\Lambda_\theta$. By strong convexity, if $A$ has a global minimizer then it is unique. We will show that $\boldsymbol{\lambda}_\theta$ is a global minimizer of $A$, where $\theta = 2(A(\boldsymbol{0}) - D^\star)/\zeta$. Suppose that $\boldsymbol{0}$ is not a global minimzer. Fix $\boldsymbol{\lambda} \in \mathbb{R}_+^K$ such that $A(\boldsymbol{0}) > A(\boldsymbol{\lambda})$. Then,

$$A(\boldsymbol{0}) > A(\boldsymbol{\lambda}) \geq D^\star + \frac{\zeta}{2}\left(\mathbb{E}\left[\|\boldsymbol{G}(X)^T\boldsymbol{\lambda}\|_2^2\right] + \|\boldsymbol{\lambda}\|_2^2\right) \geq D^\star + \frac{\zeta}{2}\|\boldsymbol{\lambda}\|_2^2. \tag{A.86}$$

Thus, $\|\boldsymbol{\lambda}\|_2^2 < \theta$. This implies that $\boldsymbol{\lambda}_\theta$ is a global minimizer of $A$, hence it is the unique global minimizer of $A$. The proof of the lemma is thus complete. $\qquad\square$

The following bound shows that $\boldsymbol{\lambda}_\zeta^\star$ is within $O(\zeta)$ of achieving $D^\star$ (see (5)).

**Lemma 6.** *Suppose Assumption 1 holds, fix $\zeta \geq 0$, and denote the unique solution and the optimal objective value of (A.83) by $\boldsymbol{\lambda}_\zeta^\star$ and $D_\zeta^\star$, respectively. We have the bounds*

$$\mathbb{E}\left[D_f^{\mathrm{conj}}\left(\boldsymbol{v}(X;\boldsymbol{\lambda}_\zeta^\star),\boldsymbol{h}^{\mathrm{base}}(X)\right)\right] \leq D_\zeta^\star \leq D^\star + \theta_{\mathrm{reg}}\cdot\zeta, \tag{A.87}$$

*where we define the constant $\theta_{\mathrm{reg}} \triangleq \lambda_{\max}^2\cdot(1+g_{\mathrm{mean}})/2$.*

*Proof.* The first bound is trivial. Using Lemma 4, we may fix a $\boldsymbol{\lambda}^\star \in \mathbb{R}_+^K$ with $\|\boldsymbol{\lambda}^\star\|_1 \leq \lambda_{\max}$ such that $\boldsymbol{\lambda}^\star$ achieves $D^\star$. By definition of $D_\zeta^\star$,

$$D_\zeta^\star \leq \mathbb{E}\left[D_f^{\mathrm{conj}}\left(\boldsymbol{v}(X;\boldsymbol{\lambda}^\star),\boldsymbol{h}^{\mathrm{base}}(X)\right) + \frac{\zeta}{2}\left\|\widetilde{\boldsymbol{G}}(X)^T\boldsymbol{\lambda}^\star\right\|_2^2\right] \leq D^\star + \theta_{\mathrm{reg}}\cdot\zeta,$$

where the last inequality follows since for the 2-matrix norm, $\|\boldsymbol{M}\boldsymbol{\lambda}\|_2 \leq \|\boldsymbol{M}\|_2\|\boldsymbol{\lambda}\|_2$ and $\|\boldsymbol{M}^T\|_2 = \|\boldsymbol{M}\|_2$. $\qquad\square$

Next, we derive a sample-complexity bound for the finite-sample problem (8) via generalizing the proofs of Theorem 3 in [AAW+20b] and Theorem 13.2 in [HR19].

**Lemma 7.** *Suppose Assumption 1 holds, and let $\lambda_{\max}$ and $g_{\max}$ be as defined in Lemma 4 and equation (A.82). For any $\delta \in (0,1)$, with $\boldsymbol{\lambda}_{\zeta,N}^\star$ denoting the unique solution to (8), it holds with probability at least $1-\delta$ that*

$$\mathbb{E}_X\left[D_f^{\mathrm{conj}}\left(\boldsymbol{v}(X;\boldsymbol{\lambda}_{\zeta,N}^\star),\boldsymbol{h}^{\mathrm{base}}(X)\right)\right] \leq D_\zeta^\star + \frac{2g_{\max}\cdot(1+\zeta\cdot\lambda_{\max})^2}{\delta\zeta N}. \tag{A.88}$$

*Proof.* Let $\Lambda \triangleq \{\boldsymbol{\lambda} \in \mathbb{R}_+^K \;;\; \|\boldsymbol{\lambda}\|_1 \leq \lambda_{\max}\}$, and consider the function $\ell : \Lambda \times \mathcal{X} \to \mathbb{R}$ defined by

$$\ell(\boldsymbol{\lambda},x) \triangleq D_f^{\mathrm{conj}}\left(\boldsymbol{v}(x;\boldsymbol{\lambda}),\boldsymbol{h}^{\mathrm{base}}(x)\right) + \frac{\zeta}{2}\left\|\widetilde{\boldsymbol{G}}(x)^T\boldsymbol{\lambda}\right\|_2^2. \tag{A.89}$$

Note that the regularized problem (A.83) can be written as

$$D_\zeta^\star \triangleq \min_{\boldsymbol{\lambda}\in\mathbb{R}_+^K}\ \mathbb{E}\left[\ell(\boldsymbol{\lambda},X)\right], \tag{A.90}$$

and the finite-sample version of it (8) can also be written as

$$D_{\zeta,N}^\star \triangleq \min_{\boldsymbol{\lambda}\in\mathbb{R}_+^K}\ \frac{1}{N}\sum_{i\in[N]}\ell(\boldsymbol{\lambda},X_i). \tag{A.91}$$

We show first that, for each fixed $x \in \mathcal{X}$, the function $\boldsymbol{\lambda} \mapsto \ell(\boldsymbol{\lambda},x)$ is $\zeta$-strongly convex over $\Lambda$. The gradient of the regularization term is $\zeta\widetilde{\boldsymbol{G}}(x)^T\boldsymbol{\lambda}$, and its Hessian is given by

$$\nabla_{\boldsymbol{\lambda}}^2\ \frac{\zeta}{2}\left\|\widetilde{\boldsymbol{G}}(x)^T\boldsymbol{\lambda}\right\|_2^2 = \zeta\boldsymbol{G}(x)\boldsymbol{G}(x)^T + \zeta\boldsymbol{I}_K. \tag{A.92}$$

Further, the function $\boldsymbol{\lambda} \mapsto D_f^{\mathrm{conj}}(\boldsymbol{v}(x;\boldsymbol{\lambda}),\boldsymbol{h}^{\mathrm{base}}(x))$ is convex as it is a pointwise supremum of linear functions. Indeed, for any $\boldsymbol{p} \in \boldsymbol{\Delta}_C$, recalling that $\boldsymbol{v}(x;\boldsymbol{\lambda}) = -\boldsymbol{G}(x)^T\boldsymbol{\lambda}$, we have the formula

$$D_f^{\mathrm{conj}}(\boldsymbol{v}(x;\boldsymbol{\lambda}),\boldsymbol{p}) = \sup_{\boldsymbol{q}\in\boldsymbol{\Delta}_C} -\boldsymbol{q}^T\boldsymbol{G}(x)^T\boldsymbol{\lambda} - D_f(\boldsymbol{q}\,\|\,\boldsymbol{p}). \tag{A.93}$$

Next, we show Lipschitzness of $\boldsymbol{\lambda} \mapsto \ell(\boldsymbol{\lambda},x)$. For any fixed $\boldsymbol{v} \in \mathbb{R}^C$ and $\boldsymbol{p} \in \boldsymbol{\Delta}_C^+$, we have the gradient (see Lemma 2)

$$\nabla_{\boldsymbol{v}}D_f^{\mathrm{conj}}(\boldsymbol{v},\boldsymbol{p}) = \boldsymbol{q}^{\mathrm{conj}}(\boldsymbol{v}) \in \boldsymbol{\Delta}_C, \tag{A.94}$$

where

$$\boldsymbol{q}^{\mathrm{conj}}(\boldsymbol{v}) \triangleq \operatorname*{argmin}_{\boldsymbol{q}\in\boldsymbol{\Delta}_C} D_f(\boldsymbol{q}\,\|\,\boldsymbol{p}) - \boldsymbol{v}^T\boldsymbol{q}. \tag{A.95}$$

Thus, we have the gradient

$$\nabla_{\boldsymbol{\lambda}} D_f^{\mathrm{conj}}\left(\boldsymbol{v}(x;\boldsymbol{\lambda}), \boldsymbol{h}^{\mathrm{base}}(x)\right) = -\boldsymbol{G}(x)\boldsymbol{q}^{\mathrm{conj}}\left(\boldsymbol{v}(x;\boldsymbol{\lambda})\right). \tag{A.96}$$

Hence, the gradient of $\boldsymbol{\lambda} \mapsto \ell(\boldsymbol{\lambda}, x)$ is

$$\nabla_{\boldsymbol{\lambda}}\ell(\boldsymbol{\lambda}, x) = -\boldsymbol{G}(x)\boldsymbol{q}^{\mathrm{conj}}\left(\boldsymbol{v}(x;\boldsymbol{\lambda})\right) + \zeta\widetilde{\boldsymbol{G}}(x)^T\boldsymbol{\lambda}, \tag{A.97}$$

which therefore satisfies the bound

$$\|\nabla_{\boldsymbol{\lambda}}\ell(\boldsymbol{\lambda}, x)\|_2 \le \|\boldsymbol{G}(x)\|_2\left(1 + \zeta \cdot \lambda_{\max}\right). \tag{A.98}$$

Therefore, each $\boldsymbol{\lambda} \mapsto \ell(\boldsymbol{\lambda}, x)$ is $A$-Lipschitz with

$$A = (1 + \zeta \cdot \lambda_{\max}) \cdot \sup_{x \in \mathcal{X}} \|\boldsymbol{G}(x)\|_2. \tag{A.99}$$

Thus, by Theorem 13.1 in [HR19], with probability $1 - \delta$ we have the bound

$$\mathbb{E}_X\left[\ell\left(\boldsymbol{\lambda}_{\zeta,N}^{\star}, X\right)\right] \le D_{\zeta}^{\star} + \frac{2A^2}{\delta\zeta N}. \tag{A.100}$$

With probability one, we have the bound

$$\mathbb{E}_X\left[D_f^{\mathrm{conj}}\left(\boldsymbol{v}\left(X;\boldsymbol{\lambda}_{\zeta,N}^{\star}\right), \boldsymbol{h}^{\mathrm{base}}(X)\right)\right] \le \mathbb{E}_X\left[\ell\left(\boldsymbol{\lambda}_{\zeta,N}^{\star}, X\right)\right]. \tag{A.101}$$

This completes the proof of the lemma. $\qquad\square$

Now, we are ready to finish the proof of Theorem 3 by specializing the above lemmas to the KL-divergence case. So, we set $f(t) = t\log t$ for the rest of the proof. By Lemmas 6–7, we have with probability $1 - \delta$

$$\mathbb{E}_X\left[D_f^{\mathrm{conj}}\left(\boldsymbol{v}(X;\boldsymbol{\lambda}_{\zeta,N}^{\star}), \boldsymbol{h}^{\mathrm{base}}(X)\right)\right] \le D^{\star} + \theta_{\mathrm{reg}} \cdot \zeta + \frac{2g_{\max} \cdot (1 + \zeta \cdot \lambda_{\max})^2}{\delta\zeta N}. \tag{A.102}$$

Thus, by Lipschitzness (Proposition 1) and (A.72)

$$\mathbb{E}_X\left[D_f^{\mathrm{conj}}\left(\boldsymbol{v}(X;\boldsymbol{\lambda}_{\zeta,N}^{(t)}), \boldsymbol{h}^{\mathrm{base}}(X)\right)\right] \le D^{\star} + \frac{1}{2}\sqrt{g_{\mathrm{mean}}}\beta e^{-rt} + \theta_{\mathrm{reg}} \cdot \zeta + \frac{2g_{\max} \cdot (1 + \zeta \cdot \lambda_{\max})^2}{\delta\zeta N}. \tag{A.103}$$

Here, we are choosing the constant $\beta$ independently of $N$ (as the optimal values of $\boldsymbol{\lambda}$ are bounded), and $r$ of order $\sqrt{\frac{\zeta}{\frac{1}{2N}+\zeta}}$ (as can be guaranteed from Corollary 3.1 and Theorem 3.4 in [DY16]).

Choose $\zeta = \Theta(N^{-1/2})$. Collecting the constants in (A.103), we obtain that

$$\mathbb{E}_X\left[D_f^{\mathrm{conj}}\left(\boldsymbol{v}(X;\boldsymbol{\lambda}_{\zeta,N}^{(t)}), \boldsymbol{h}^{\mathrm{base}}(X)\right)\right] \le D^{\star} + \frac{1}{2}\sqrt{g_{\mathrm{mean}}}\beta e^{-rt} + \frac{\ell}{\delta\sqrt{N}} \tag{A.104}$$

for some constant $\ell$ that is completely determined by $\theta_{\mathrm{reg}}$, $g_{\max}$, and $\lambda_{\max}$. This bound can be further upper bounded by $D^{\star} + O(N^{-1/2})$ by choosing $t \ge \frac{1}{2r}\log N = \Theta(\log N)$, thereby completing the proof of the theorem.

### A.1.8 Linearized multi-class group fairness criteria

We include, for completeness, how the group-fairness metrics in Table 2 linearize, i.e., written in the form:

$$\boldsymbol{\mu}(\boldsymbol{h}) \triangleq \mathbb{E}_{P_X}\left[\boldsymbol{G}(X)\boldsymbol{h}(X)\right] \le \boldsymbol{0}. \tag{A.105}$$

We assume that we have in hand a well-calibrated classifier that approximates $P_{Y,S|X}$, i.e., that predicts both group membership $S$ and the true label $Y$ from input variables $X$. This classifier can be directly marginalized into the following models:

- a label classifier $\boldsymbol{h}^{\mathrm{base}} : \mathcal{X} \to \boldsymbol{\Delta}_C$ that predicts true label from input variables,

$$\boldsymbol{h}^{\mathrm{base}}(x) \triangleq (P_{Y|X=x}(1), \cdots, P_{Y|X=x}(C)) \quad \text{for } x \in \mathcal{X}, \tag{A.106}$$

- a group membership classifier $\boldsymbol{s} : \mathcal{X} \times \mathcal{Y} \to \boldsymbol{\Delta}_A$ that uses input and output variables to predict group membership,

$$\boldsymbol{s}(x, y) \triangleq (P_{S|X,Y}(1 \mid x, y), \cdots, P_{S|X,Y}(A \mid x, y)) \quad \text{for } (x, y) \in \mathcal{X} \times \mathcal{Y}, \tag{A.107}$$

We let $\boldsymbol{e}_1, \cdots, \boldsymbol{e}_C$ denote the standard basis vectors of $\mathbb{R}^C$. We suppose that the support of the group attribute $S$ is $\mathcal{S} \triangleq [A]$.

**Statistical parity.** This fairness metric measures whether the predicted outcome $\widehat{Y}$ is independent of the group attribute $S$. For statistical parity, the $\boldsymbol{G}(x)$ matrix has rows:

$$\left((-1)^\delta \frac{\sum_{c=1}^C s_a(x,c) h_c^{\text{base}}(x)}{P_S(a)} - \left(\alpha + (-1)^\delta\right)\right) \boldsymbol{e}_{c'}.$$

There are $K = 2AC$ rows since $(\delta, a, c') \in \{0,1\} \times [A] \times [C]$. See Appendix A.1.9 for a full derivation.

**Equalized odds.** This fairness metric requires the predicted outcome $\widehat{Y}$ and the group attribute $S$ to be independent conditioned on the true label $Y$. When the classification task is binary, the equalized odds becomes the equality of false positive rate and false negative rate over all groups. For equalized odds, the $\boldsymbol{G}(x)$ matrix has rows:

$$\left((-1)^\delta \frac{s_{a'}(x,c) h_c^{\text{base}}(x)}{P_{S|Y=c}(a')} - \left(\alpha + (-1)^\delta\right) h_c^{\text{base}}(x)\right) \boldsymbol{e}_{c'}.$$

There are $K = 2AC^2$ rows.

**Overall accuracy equality.** This fairness metric requires the accuracy of the predictive model to be the same across all group groups. The $\boldsymbol{G}(x)$ matrix has rows:

$$(-1)^\delta \frac{s_a(x,\cdot) \odot \boldsymbol{h}^{\text{base}}(x)}{P_S(a)} - \left(\alpha + (-1)^\delta\right) \cdot \boldsymbol{h}^{\text{base}}(x),$$

where $\odot$ represents the element-wise product. There are $K = 2A$ rows.

### A.1.9   A detailed derivation of linearization of statistical parity

We derive the linearized formula for Statistical Parity given in Appendix A.1.8. Recall that a prediction $\widehat{Y}$ satisfies statistical parity (SP) if it is independent of the group attribute $S$, i.e., $P_{\widehat{Y}|S=a}(c') = P_{\widehat{Y}}(c')$ for every $(a, c') \in [A] \times [C]$. A relaxed notion of SP is 'approximate independence' of $\widehat{Y}$ and $S$: $|P_{\widehat{Y}|S=a}(c')/P_{\widehat{Y}}(c') - 1| \leq \alpha$ for some small $\alpha \geq 0$ and all $(a, c') \in [A] \times [C]$. Using $P_{\widehat{Y}|S} = P_{\widehat{Y},S}/P_S$ and rearranging, the above inequality is equivalent to

$$\pm P_{\widehat{Y},S}(c',a)/P_S(a) - (\alpha \pm 1)P_{\widehat{Y}}(c') \leq 0.$$

We expand via conditioning $\widehat{Y}$ on $X$, and $S$ on $(X,Y)$. Recall that $h_{c'}(x) = P_{\widehat{Y}|X=x}(c')$ and $s_a(x,c) = P_{S|X=x,Y=c}(a)$ by definition, and that we have a Markov chain $(Y,S)$–$X$–$\widehat{Y}$; hence, $P_{\widehat{Y}}(c') = \mathbb{E}[h_{c'}(X)]$ and $P_{\widehat{Y},S}(c',a) = \mathbb{E}[\sum_{c \in [C]} s_a(X,c) h_c^{\text{base}}(X) h_{c'}(X)]$. Thus, we can write approximate SP as

$$\mathbb{E}\left[\left(\pm P_S(a)^{-1} \sum_{c \in [C]} s_a(X,c) h_c^{\text{base}}(X) - (\alpha \pm 1)\right) h_{c'}(X)\right] \leq 0.$$

We denote $\boldsymbol{h}(x) = (h_1(x), h_2(x), \cdots, h_C(x))^T$, and for $(\delta, a, c') \in \{0,1\} \times [A] \times [C]$, denote

$$\boldsymbol{g}^{(\delta,a,c')}(x) := \left((-1)^\delta P_S(a)^{-1} \sum_{c \in [C]} s_a(x,c) h_c^{\text{base}}(x) - (\alpha + (-1)^\delta)\right) \boldsymbol{e}_{c'},$$

where $\{\boldsymbol{e}_{c'}\}_{c' \in [C]}$ is the standard basis of $\mathbb{R}^C$. Then, for each pair $(\delta, a, c') \in \{0,1\} \times [A] \times [C]$, we have a linear constraint $\mathbb{E}[\boldsymbol{g}^{(\delta,a,c')}(X)^T \boldsymbol{h}(X)] \leq 0$. Since there are $K = 2AC$ possible triplets $(\delta, a, c')$, we convert the SP constraint into $K$ linear constraints.

## A.2 Additional experiments and more details on the experimental setup

### A.2.1 Numerical Benchmark Details

#### A.2.1.1 Datasets

The HSLS dataset is collected from 23,000+ participants across 944 high schools in the USA, and it includes thousands of features such as student demographic information, school information, and students' academic performance across several years. We preprocessed the dataset (e.g., dropping rows with a significant number of missing entries, performing k-NN imputation, normalization), and the number of samples reduced to 14,509.

The ENEM dataset, collected from the 2020 Brazilian high school national exam and made available by the Brazilian Government [INE20], is comprised of student demographic information, socio-economic questionnaire answers (e.g., parents education level, if they own a computer) and exam scores. We preprocess the dataset by removing missing values, repeated exam takers, and students taking the exam before graduation ("treineiros") and obtain ∼1.4 million samples with 138 features.

#### A.2.1.2 Hyperparameters

For logistic regression and gradient boosting, we use the default parameters given by Scikit-learn. For random forest, we set the number of trees and the minimum number of samples per leaf to 10. For all classifiers, we fixed the random state to 42. When running `FairProjection` (cf. Algorithm 1), we set the hyperparameters $\zeta = 1/\sqrt{N}$ (see Theorem 3) and $\rho = 2$ (see Appendix A.1.3.1), where $N$ is the number of samples.

#### A.2.1.3 Benchmark Methods

For binary classification, we compare with six different benchmark methods:

- EqOdds [HPS16]: We use AIF360 implementation of EqOddsPostprocessing and we use 50% of the test set as a validation set, i.e., 70% training set, 15% validation set, 15% test set.
- CalEqOdds [PRW+17]: We use AIF360 implementation of CalibratedEqOddsPostprocessing and we use 50% of the test set as a validation set, i.e., 70% training set, 15% validation set, 15% test set.
- Reduction [ABD+18]: We use AIF360 implementation of ExponentiatedGradientReduction, and we use 10 different epsilon values as follows: $[0.001, 0.005, 0.01, 0.02, 0.05, 0.1, 0.2, 0.5, 1, 2]$. We used EqualizedOdds constraint for MEO experiments and DemographicParity for statistical parity experiments.
- Rejection [KKZ12]: We use AIF360 implementation of RejectOptionClassification. We use the default parameters except metric_ub and metric_lb, namely, low_class_thresh = 0.01, high_class_thresh = 0.99, num_class_thresh = 100, num_ROC_margin = 50. We set the values metric_ub = $\epsilon$ and metric_lb = $-\epsilon$ to obtain trade-off curves. Epsilon values we used are: $[0.001, 0.005, 0.01, 0.02, 0.05, 0.1, 0.2, 0.5, 1, 2]$.
- LevEqOpp [CDH+19]: We used the code provided in the Github repo, originally programmed in R. We converted it into Python, and verified that the Python version achieved similar accuracy/fairness performance to their R version on UCI Adult dataset. We follow the same hyperparameters setup in [CDH+19].

The following four methods, despite being mentioned in Table 1, are not included in the experiments:

- FACT [KCT20]: We used the code provided on the Github repo. We did not include the results in the main text as we found that:
  - (i) This method is not directly comparable because they find post-processing parameters on the entire test set and apply them on the test set. This is different from all other methods we are comparing including our method, which use training set or a separate validation set to fit the post-processing mechanism. For this reason, FACT often has a point that lies above all other curves on the accuracy-fairness plot. However, this is not a fair comparison. We include the results of FACT in the COMPAS plots for the sake of demonstration.

(ii) We found the results produced by this method inconsistent. Partial reason is due to the problem of finding mixing rates—probability of flipping $\widehat{Y} = 1$ to $0$ (i.e., $P(\widetilde{Y} = 0|\widehat{Y} = 1)$) and vice-versa—which have to be between 0 and 1. But there are cases where these values lie outside $[0, 1]$, which leads to erroneous and inconsistent results.

For the results we present in the COMPAS plots, we used 20 epsilon values from 1 to $10^{-4}$, equidistant in log space. We used 10 different train/test splits as we do in all other experiments. If certain splits does not produce a feasible solution, we drop those results. If none of the 10 splits produce a feasible solution, we drop the epsilon value. At the end, we had 19 epsilon values.

- Identifying [JN20]: Their optimization formulation is a special case of our formulation when $f$-divergence is KL divergence, but their algorithm requires retraining a classifier multiple times to solve the optimization problem, which results in a much slower runtime compared to ours (see Lines 1037–1046 in Appendix B.4). Nevertheless, we will add experiments for binary classification using [JN20] in the final version.

- FST [WRC20, WRC21]: Codes are not available publicly.

- Overlapping [YCK20]: We did not include this method for binary classification experiments as it reduces to the Reductions [ABD+18] approach for the binary class, binary protected group case. We could not benchmark for multi-class experiments with the code available online as it was assuming binary class (even though multiple protected groups).

For multi-class comparison, we compare with Adversarial [ZLM18]. In theory, the adversarial debiasing method is applicable to multi-class labels and groups, but its AIF360 implementation works only for binary labels and binary groups. We adapted their implementation to work on multi-class labels by changing the last layer of the classifier model from one-neuron sigmoid activation to multi-neuron soft-max activation. We varied adversary_loss_weight to obtain a trade-off curve, values taken from $[0.001, 0.01, 0.1, 0.2, 0.35, 0.5, 0.75]$. For all other parameters, we used the default values: num_epochs $= 50$, batch_size $= 128$, classifier_num_hidden_units $= 200$.

There are some methods that are relevant to our work but we could not benchmark in our experiments due to the lack of publicly available codes, including [WRC21], [MW18], [JSW22].

### A.2.2 Additional experiments on runtime of FairProjection

We preform an ablation study on the runtime to illustrate that the parallelizability of FairProjection can significantly reduce the runtime, especially when the dataset contains hundreds of thousands of samples. We report the runtime of FairProjection-KL on ENEM with 2 classes, 2 groups, and with different sizes. In Table A.1, we observe that when the number of samples exceeds 200k, parallelization leads to $10.1\times$ to $15.5\times$ speedup of the runtime.

| Method | # of Samples (in thousands) | | | | | |
|---|---|---|---|---|---|---|
| | 20 | 50 | 100 | 200 | 500 | $\sim$1400 |
| Non-Parallel | $0.37\pm0.00$ | $0.87\pm0.01$ | $1.72\pm0.01$ | $3.53\pm0.01$ | $9.09\pm0.01$ | $25.26\pm0.02$ |
| Parallel (GPU) | $0.18\pm0.00$ | $0.22\pm0.01$ | $0.25\pm0.01$ | $0.32\pm0.01$ | $0.64\pm0.01$ | $1.63\pm0.05$ |
| Speedup | $2.00\times$ | $3.92\times$ | $7.21\times$ | $10.97\times$ | $14.23\times$ | $15.46\times$ |

**Table A.1:** Execution time of parallel (on GPU) and non-parallel (on CPU) versions of the FairProjection-KL ADMM algorithm on the ENEM datasets with different sizes (time shown in minutes) with gradient boosting base classifiers.

### A.2.3 Additional Explanation on runtime comparison

The theoretical analysis below contrasts the runtimes of both FairProjection and Reduction [ABD+18], which is in line with our numerically observed comparison in Table 3. Two key factors make FairProjection faster than Reduction:

1. FairProjection needs a much lower number of iterations than Reduction does (logarithmic vs. polynomial).

2. Each iteration for FairProjection is less computationally expensive than its counterpart in Reduction. In fact, it is independent of the underlying model being projected, whereas Reduction requires retraining.

In more detail, one can obtain from [ABD+18, Theorem 3] that the Reductions approach converges in $O(N^2)$ iterations (where $N$ is the number of samples and we use the suggested $\alpha = 1/2$ in [ABD+18, Theorem 3] according to the discussion at the top of page 6 therein). Taking the runtime of each iteration into consideration, one cannot hope for a runtime faster than $O(N^4)$ for Reduction. In fact, the runtime for Reduction must be higher than $O(N^4)$, since each of its iterations performs the subroutine $\text{BEST}_h(\lambda)$, which is a 'cost-sensitive classification' problem (i.e., numerically solving for an optimal classifier), and the $O(N^4)$ estimate would hold only if this *retraining* procedure can be done in *constant time* (which might be overly optimistic). In contrast, FairProjection does not require this retraining procedure at all, runs in $O(\log N)$ iterations, has $O(N)$ runtime for each iteration, and can perform much of each iteration in a parallel way.

For the dependence of the runtime of FairProjection on the number of groups, we note that there is a *linear* dependence on the number of constraints $K$ when the number of samples $N$ is much larger than $K$ (which is the case for all datasets we consider), so one can say that the runtime is at most $\gamma K N \log N$ for an *absolute* constant $\gamma$. Note that there are $K = 2AC$ constraints for statistical parity, where $A$ is the number of sensitive groups, and $C$ is the number of classes; e.g., for the ENEM-1.4M-2C dataset that is used in Table 3, we get $K = 8$ for statistical parity. The $K$ factor in the $O(KN \log N)$ rate comes from the creation of the vector $q$ in Algorithm 1. If one does not parallelize, still one gets a runtime of $O(CKN \log N)$. Interestingly, the $v_i$-update step runtime in Algorithm 1 is $O(C)$ for a fixed $i \in [N]$ for both KL-divergence and Cross Entropy (see Appendices A.1.3 and A.1.4).

### A.2.4 Omitted Experimental Results on Accuracy-Fairness Trade-off

#### A.2.4.1 Accuracy-fairness trade-off in binary classification

We include the results of benchmark methods and Fair Projection on 4 datasets (HSLS, ENEM-50k, Adult, and COMPAS) and 3 base classifiers (Logistic regression, Random forest, and GBM) in Figures A.1-A.8. For equalized odds experiments, we have six benchmark methods (EqOdds, Rejection, Reduction, CalEqOdds, FACT, LevEqOpp). For statistical parity experiments, we have Rejection and Reduction. We plot Fair Projection with both cross entropy and KL divergence.

When a method performs significantly worse than others, we did not plot its results. We did not include Rejection in the Adult plots as it did not produce consistent and reliable results on this dataset. CalEqOdds is included only in COMPAS as its performance was significantly worse and the point was too far away from other curves in all other datasets. FACT is also included only in the COMPAS plots and the reasons for this are explained in Appendix A.2.1.3.

We observe that Fair Projection performs consistently well in all four datasets. FairProjection-CE and FairProjection-KL have similar performance (i.e., overlapping curves) in most cases. The performance of Fair Projection is often comparable with Reduction. Rejection has competitive performance in ENEM-50k and HSLS, but its performance falters in COMPAS and Adult. EqOdds produces a point with very low MEO but with a substantial loss in accuracy. LevEqOpp also yields a point with low MEO but with a much smaller accuracy drop. Even though LevEqOpp only optimizes for FNR difference between two groups, it performs surprisingly well in terms of MEO in all four datasets. However, we note that LevEqOpp can only produce a point, not a curve, and it does not enjoy the generality of Fair Projection as it is specifically designed for binary-class, binary-group predictions and minimizing Equalized Opportunity difference.

#### A.2.4.2 Accuracy-fairness trade-off in multi-class/multi-group classification

In the main text, we showed the performance of FairProjection-CE on multi-class prediction with 5 classes and 2 groups (see Figure 2). We include results under a few different multi-class settings here. First, we show results on ENEM-50k-5-5 which has 5 classes and 5 groups in Figure A.9 and A.10 . We obtain 5 groups by not binarizing the race feature. Then, we show results on binary classification with 5 groups in Figure A.11 and A.12. Finally, we include the extended version of Figure 2 that include both FairProjection-CE and FairProjection-KL in Figure A.13.

To measure multi-class performance, we extend the definition of mean equalized odds (MEO) and statistical parity as follows:

$$\text{MEO} = \max_{i \in \mathcal{Y}} \max_{s_1, s_2 \in \mathcal{S}} (|\text{TPR}_i(s_1) - \text{TPR}_i(s_2)| + |\text{FPR}_i(s_1) - \text{FPR}_i(s_2)|)/2$$

(A.108)

$$\text{Statistical Parity} = \max_{i \in \mathcal{Y}} \max_{s_1, s_2 \in \mathcal{S}} |\text{Rate}_i(s_1) - \text{Rate}_i(s_2)|$$

(A.109)

where we denote $\text{TPR}_i(s) = P(\widehat{Y} = i \mid Y = i, S = s)$, $\text{FPR}_i(s) = P(\widehat{Y} = i \mid Y \neq i, S = s)$, and $\text{Rate}_i(s) = P(\widehat{Y} = i \mid S = s)$.

In all experiments, `FairProjection` reduces MEO and statistical parity significantly (e.g., 0.22 to 0.14) with a negligible sacrifice in accuracy.

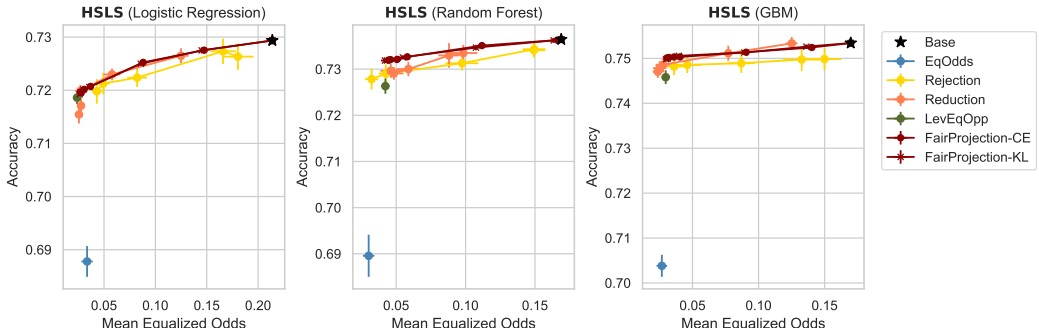

**Figure A.1:** Accuracy-fairness curves of FairProjection and benchmark methods on the HSLS dataset with 3 different models (Logistic regression, Random forest, GBM). The fairness constraint is MEO.

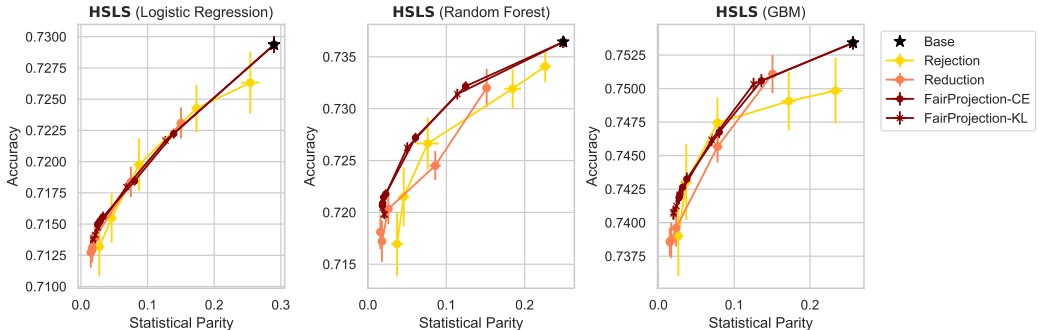

**Figure A.2:** Accuracy-fairness curves of FairProjection and benchmark methods on the HSLS dataset with 3 different models (Logistic regression, Random forest, GBM). The fairness constraint is statistical parity.

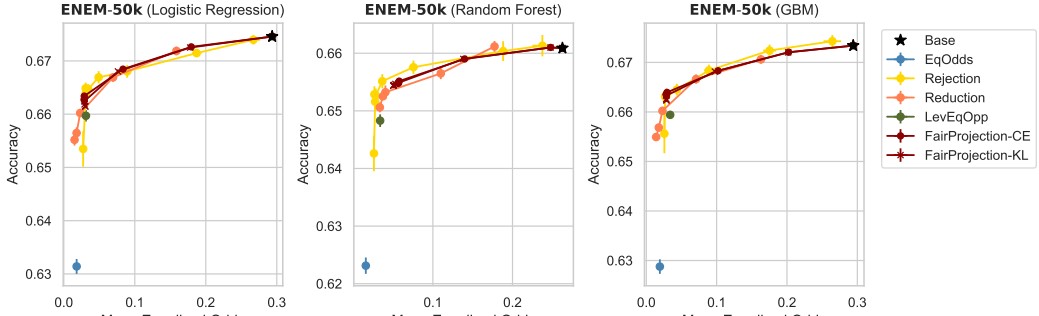

**Figure A.3:** Accuracy-fairness curves of FairProjection and benchmark methods on the ENEM-50k-2C dataset with 3 different models (Logistic regression, Random forest, GBM). The fairness constraint is MEO.

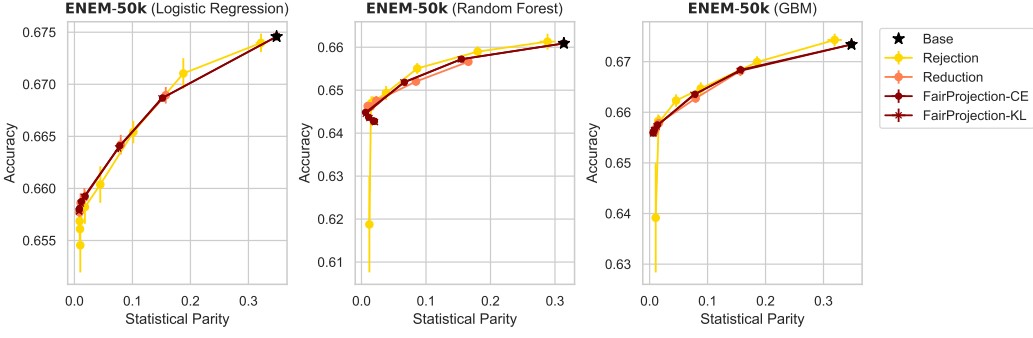

**Figure A.4:** Accuracy-fairness curves of FairProjection and benchmark methods on the ENEM-50k-2C dataset with 3 different models (Logistic regression, Random forest, GBM). The fairness constraint is statistical parity.

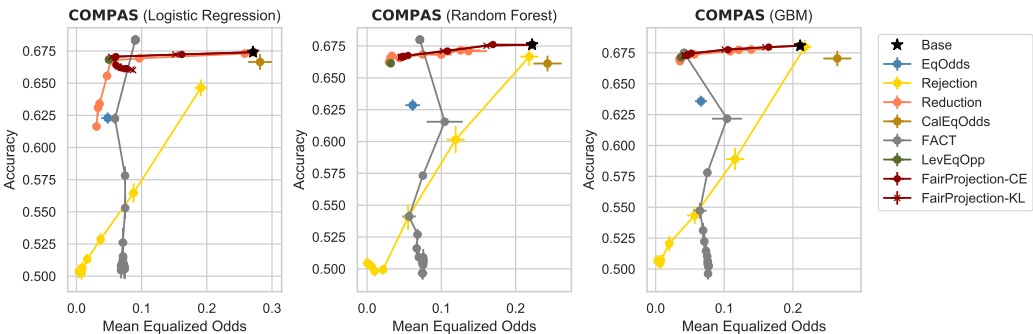

**Figure A.5:** Accuracy-fairness curves of FairProjection and benchmark methods on COMPAS with 3 different models (Logistic regression, Random forest, GBM). The fairness constraint is MEO.

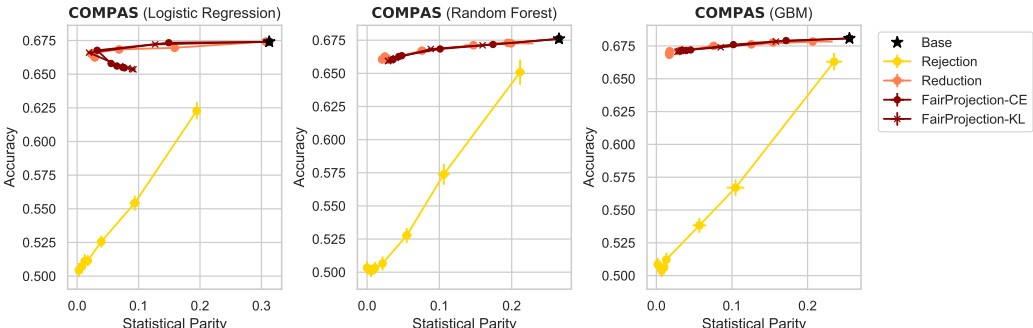

**Figure A.6:** Accuracy-fairness curves of FairProjection and benchmark methods on COMPAS with 3 different models (Logistic regression, Random forest, GBM). The fairness constraint is statistical parity.

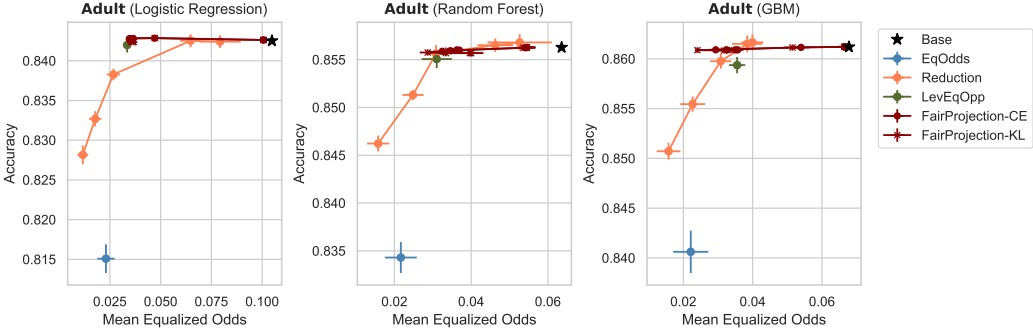

**Figure A.7:** Accuracy-fairness curves of FairProjection and benchmark methods on the Adult dataset with 3 different models (Logistic regression, Random forest, GBM). The fairness constraint is MEO.

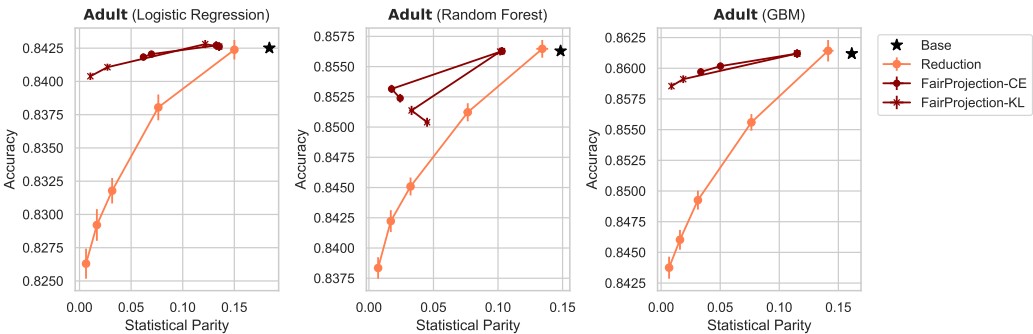

**Figure A.8:** Accuracy-fairness curves of FairProjection and benchmark methods on the Adult dataset with 3 different models (Logistic regression, Random forest, GBM). The fairness constraint is statistical parity.

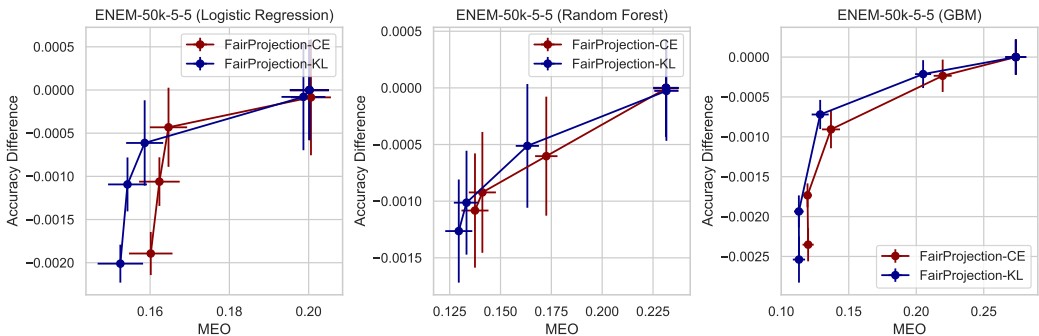

**Figure A.9:** Accuracy-fairness curves of `FairProjection-CE` and `FairProjection-KL` on ENEM-50k with with 5 labels, 5 groups and different base classifiers base classifiers. The fairness constraint is MEO.

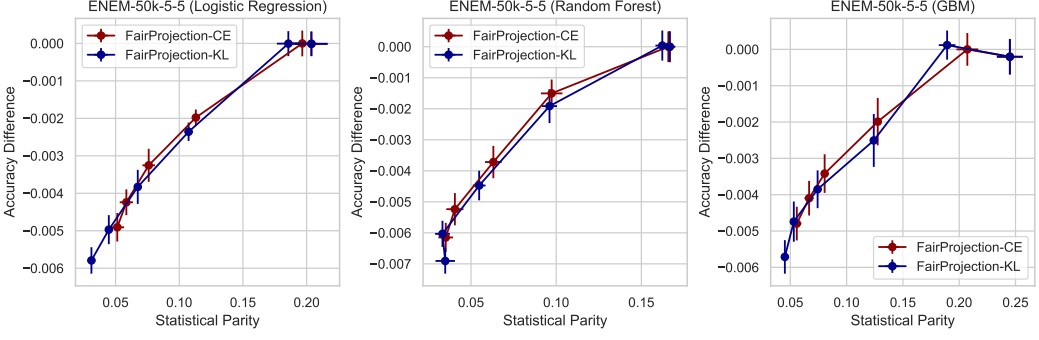

**Figure A.10:** Accuracy-fairness curves of `FairProjection-CE` and `FairProjection-KL` on ENEM-50k with with 5 labels, 5 groups and different base classifiers base classifiers. The fairness constraint is SP.

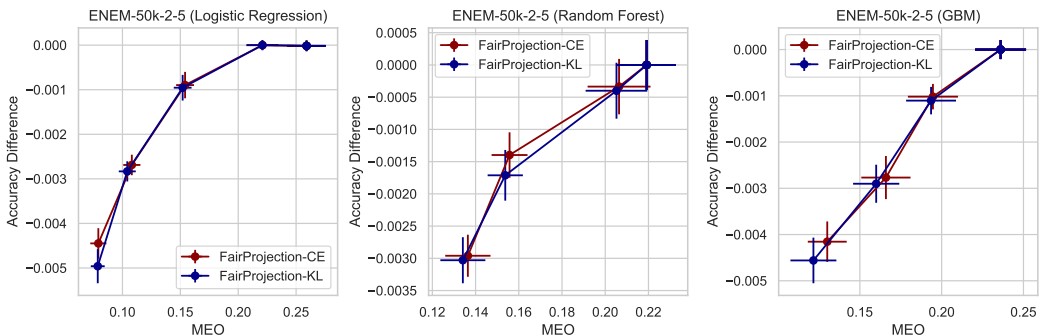

**Figure A.11:** Accuracy-fairness curves of `FairProjection-CE` and `FairProjection-KL` on ENEM-50k with with 2 labels, 5 groups and different base classifiers base classifiers. The fairness constraint is MEO.

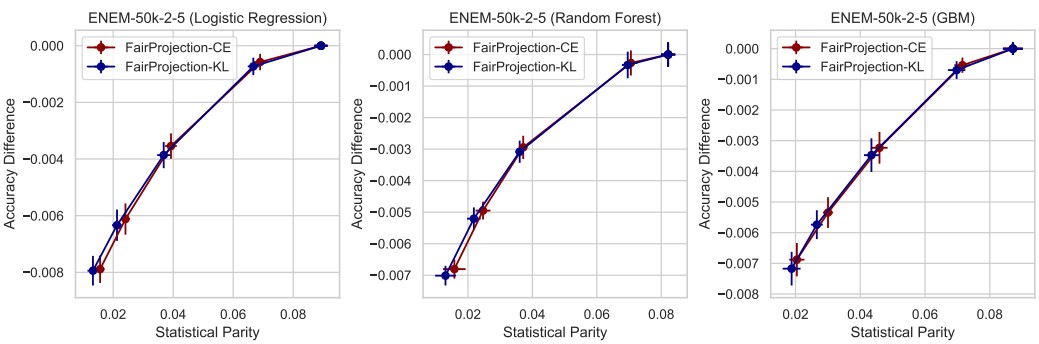

**Figure A.12:** Accuracy-fairness curves of `FairProjection-CE` and `FairProjection-KL` on ENEM-50k with with 2 labels, 5 groups and different base classifiers base classifiers. The fairness constraint is SP.

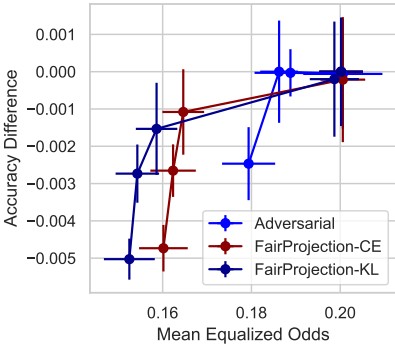

**Figure A.13:** Comparison of `FairProjection-CE` and `FairProjection-KL` with `Adversarial` on ENEM-50k-5-2, meaning 5 labels, 2 groups. The reason for the difference comparing to Fig. 2 is that we resampled 50k data points from ENEM.

| Method | Multiclass | Multigroup | **Feature** Scores | Curve | Parallel | Rate | Metric |
|---|---|---|---|---|---|---|---|
| Reductions [ABD+18] | ✗ | ✓ | ✓ | ✓ | ✗ | ✓ | SP, (M)EO |
| Reject-option [KKZ12] | ✗ | ✓ | ✗ | ✓ | ✗ | ✗ | SP, (M)EO |
| EqOdds [HPS16] | ✗ | ✓ | ✗ | ✗ | ✗ | ✓ | EO |
| LevEqOpp [CDH+19] | ✗ | ✗ | ✗ | ✗ | ✗ | ✗ | FNR |
| CalEqOdds [PRW+17] | ✗ | ✗ | ✓ | ✗ | ✗ | ✓ | MEO |
| FACT [KCT20] | ✗ | ✗ | ✗ | ✓ | ✗ | ✗ | SP, (M)EO |
| Identifying [JN20] | ✓✗ | ✓ | ✓ | ✓ | ✗ | ✗ | SP, (M)EO |
| FST [WRC20, WRC21] | ✗ | ✓ | ✓ | ✓ | ✗ | ✓ | SP, (M)EO |
| Overlapping [YCK20] | ✓ | ✓ | ✓ | ✓ | ✗ | ✗ | SP, (M)EO |
| Adversarial [ZLM18] | ✓ | ✓ | N/A | ✓ | ✓ | ✗ | SP, (M)EO |
| FairProjection (ours) | ✓ | ✓ | ✓ | ✓ | ✓ | ✓ | SP, (M)EO |

**Copy of Table 1.** Comparison between benchmark methods. **Multiclass/multigroup**: implementation takes datasets with multiclass/multigroup labels; **Scores**: processes raw outputs of probabilistic classifiers; **Curve**: outputs fairness-accuracy tradeoff curves (instead of a single point); **Parallel**: parallel implementation (e.g., on GPU) is available; **Rate**: convergence rate or sample complexity guarantee is proved; **Metric**: applicable fairness metric, with SP↔Statistical Parity, EO↔Equalized Odds, MEO↔Mean EO. Since FairProjection is a post-processing method, we focus our comparison on post-processing fairness intervention methods, except for Reductions [ABD+18], which is a representative in-processing method, and Adversarial [ZLM18], which we use to benchmark multi-class prediction. For comparing in-processing methods, see [LPB+21, Table 1].

### A.2.5  More on related work

Our method is a model-agnostic post-processing method, so we focus our comparison on such post-processing fairness intervention methods. In the above table, the only exception is Adversarial [ZLM18], which we use to benchmark multi-class prediction. Adversarial [ZLM18] is an in-processing method based on generative-adversarial network (GAN) where the adversary tries to guess the sensitive group attribute $S$ from $Y$ and $\widehat{Y}$. Even though this GAN-based approach is applicable to multi-class, multi-group prediction, it cannot be universally applied to any pre-trained classifier like our method.

EqOdds [HPS16], CalEqOdds [PRW+17] and LevEqOpp [CDH+19] are post-processing methods designed for binary prediction with binary groups. They find different decision thresholds for each group that equalize FNR and FPR of two groups. CalEqOdds [PRW+17] has an additional constraint that the post-processed classifier must be well-calibrated, and we observe in our experiments that this stringent constraint leads to a low-accuracy classifier especially when there is a big gap in the base rate between the two groups. FACT [KCT20] follows a similar approach but generalizes this to an optimization framework that can have both equalized odds and statistical parity constraints and flexible accuracy-fairness trade-off. The optimization formulation finds a desired confusion matrix, and their proposed post-processing method flips the predictions to match the desired confusion matrix. Reject-option [KKZ12] is similar in that it flips predictions near the decision threshold. In [KKZ12], instead of finding the optimal confusion matrix, it performs grid search to find the optimal margin around the decision threshold that can minimize either equalized odds or statistical parity. For these methods that center around modifying decision thresholds, it is not straightforward to extend to multi-class and multi-group as one will have to consider $\binom{|\mathcal{Y}|}{2} \cdot \binom{|\mathcal{S}|}{2}$ boundaries.

FST [WRC20, WRC21] tackles fairness intervention via minimizing cross-entropy for binary classes. Their method is inherently tailored to binary classification *and* only a cross-entropy objective function, and our FairProjection-CE reduces to FST for the case of CE and binary classification tasks. Identifying [JN20] is a method for minimizing KL-divergence for group-fairness intervention, which changes the label weights (via a convex combination) between unweighted and weighted samples, but it is not clear that this would navigate a good fairness-accuracy trade-off curve. Their method can be extended to non-binary prediction with non-binary groups by an appropriate choice of base classifier and fairness constraints, which is a non-trivial extension of the accompanying code, and we chose not to pursue this. Note that [JN20] and FairProjection solve the KL-divergence minimization in very different ways. In particular, the runtime of [WRC20, WRC21] on a 350k training dataset is

longer than 30 minutes using logistic regression as a base classifier (in comparison, the runtime of `FairProjection` for a 500k dataset is less than 1 minute). This is because they require reweighing the data and retraining a large number of times. Hence, it is inherently non-parallelizable.

### A.2.5.1 Fairness in Multi-Class Prediction

Methods that are based on optimization with a fairness regularizer often can be easily extended to multi-class prediction as it only requires a small change in the regularizer. For example, instead of using $|\text{FNR}_0(x) - \text{FNR}_1(x)|$, one can replace this with

$$\sum_{i \in \mathcal{Y}} \sum_{j \neq i \in \mathcal{Y}} |P(\widehat{Y} = j \mid Y = i, S = 0) - P(\widehat{Y} = j \mid Y = i, S = 1)|. \qquad (A.110)$$

FERM [DOBD$^+$18] mentions how their method can be extended to multi-class sensitive attribute. Similarly, we believe that their method can be used for multi-class labels as well. The reductions approach [ABD$^+$18] assumes binary labels but is has natural extension to multi-class, which is explored in [YCK20]. In-processing methods proposed in [CHS20] and [ZLM18] allow for both multi-class labels and multi-class group attributes. [ZLM18] aims to achieve the independence between the sensitive attribute $S$ and $\widehat{Y}$ or $\widehat{Y}$ given $Y$ by training an adversary who tries to figure out $\widehat{S}$. [CHS20] directly estimates the fairness loss (e.g., A.110) using kernel density estimation. They also demonstrate the empirical performance in a three-class classification using synthetic data. Another in-processing method is [AAV19] where the authors propose a way to incorporate multi-class fairness constraints into decision tree training. The preprocessing method suggested in [CKV20] is conceptually similar to our methods in that it aims to minimize the KL-divergence between the original distribution and preprcoessed distribution while satisfying fairness constraints. Their method, however, requires all feature vectors to be binary, and applies only to demographic parity or representation rate. There exist other notions of fairness, which is different from commonly-used group fairness metrics such as envy-freeness [BDNP19] or best-effort [KJW$^+$21], which can be applied to multi-class prediction tasks.

Finally, there are unpublished works [DEHH21, YX20] that could handle multi-class classification. Specifically, [DEHH21] presents a post-processing method that selects different thresholds for each group to achieve demographic parity. [YX20] formulates SVM training as a mixed-integer program and integrates fairness regularizer in the objective, which can also deal with multi-class.

## A.3   Datasheet for ENEM 2020 dataset

### Questions

The questions below are derived from [GMV$^+$21] and aim to provide context about the ENEM-2020 dataset. We highlight that we did not create the dataset nor collect the data included in it. Instead, we simply provide a link to the ENEM-2020 data at [INE20]. At the time of writing, the ENEM-2020 dataset is open and made freely available by the Brazilian Government at [INE20] under a Creative Commons Attribution-NoDerivs 3.0 Unported License [Com]. We provide the datasheet below to clarify certain aspects of the dataset (e.g., motivation, composition, etc.) since the original information is available in Portuguese at [INE20], thus limiting its access to a broader audience. The website [INE20] contains a link to download a `.zip` file which contains the ENEM-2020 data in `.csv` format and extensive accompanying documentation.

The datasheet below is **not** a substitute for the explanatory files that are downloaded together with the dataset at [INE20], and we emphatically recommend the user to familiarize themselves with associated documentation prior to usage. We also strongly recommend the user to carefully read the "Leia-Me" (readme) file `Leia_Me_Enem_2020.pdf` available in the same `.zip` folder that contains the dataset. The answers in the datasheet below are based on an English translation of information available at [INE20] and may be incomplete or inaccurate. The datasheet below is based on our own independent analysis and in no way represents or attempts to represent the opinion or official position of the Brazilian Government and its agencies.

We also note that we do not distribute the ENEM-2020 dataset directly nor host the dataset ourselves. Instead, we provide a link to download the data from a public website hosted by the Brazilian Government. The dataset may become unavailable in case the link in [INE20] becomes inaccessible.

**Motivation**

- **For what purpose was the dataset created?** According to the "Leia-me" (Read Me) file that accompanies the data, the dataset was made available to fufill the mission of the Instituto Nacional de Estudos e Pesquisas Educacionais Anísio Teixeira (INEP) of developing and disseminating data about exams and evaluations of basic education in Brazil.

- **Who created the dataset (e.g., which team, research group) and on behalf of which entity (e.g., company, institution, organization)?** The dataset was developed by INEP, which is a government agency connected to the Brazilian Ministry of Education.

- **Who funded the creation of the dataset?** The data is made freely available by the Brazilian Government.

**Composition**

- **What do the instances that comprise the dataset represent (e.g., documents, photos, people, countries)?** The instances of the dataset are information about individual students who took the Exame Nacional do Ensino Médio (ENEM). The ENEM is the capstone exam for Brazilian students who are graduating or have graduated high school.

- **How many instances are there in total (of each type, if appropriate)?** The raw data provided in at [INE20] has approximately 5.78 million entries. The processed version we use in our experiments has approximately 1.4 million entries.

- **Does the dataset contain all possible instances or is it a sample (not necessarily random) of instances from a larger set?** The data provided is the lowest level of aggregation of data collected from ENEM exam-takers made available by INEP.

- **What data does each instance consist of?** We provide a brief description of the features available in the raw public data provided at [INE20]. Upon downloading the data, a detailed description of features and their values are available (in Portuguese) in the file titled `Dicionário_Mircrodados_ENEM_2020.xsls`. The features include:

  - **Information about exam taker:** exam registration number (masked), year the exam was taken (2020), age range, sex, marriage status, race, nationality, status of high school graduation, year of high school graduation, type of high school (public, private, n/a), if they are a "treineiro" (i.e., taking the exam as practice).
  - **School data**: city and state of participant's school, school administration type (private, city, state, or federal), location (urban or rural), and school operation status.
  - **Location where exam was taken**: city and state.
  - **Data on multiple-choice questions**: The exam is divided in 4 parts (translated from Portuguese): natural sciences, human sciences, languages and codes, and mathematics. For each part there is data if the participant attended the corresponding portion of the exam, the type of exam book they received, their overall grade, answers to exam questions, and the answer sheet for the exam.
  - **Data on essay question**: if participant took the exam, grade on different evaluation criteria, and overall grade.
  - **Data on socio-economic questionnaire answers:** the data include answers to 25 socio-economic questions (e.g., number of people who live in your house, family average income, if the your house has a bathroom, etc.).

- **Is there a label or target associated with each instance?** No, there is no explicit label. In our fairness benchmarks, we use grades in various components of the exam as a predicted label.

- **Is any information missing from individual instances?** Yes, certain instances have missing values.

- **Are relationships between individual instances made explicit (e.g., users' movie ratings, social network links)?** No explicit relationships identified.

- **Are there recommended data splits (e.g., training, development/validation, testing)?** No.

- **Are there any errors, sources of noise, or redundancies in the dataset?** The data contains missing values and, according to INEP, was collected from individual exam takers. The information is self-reported and collected at the time of the exam.

- **Is the dataset self-contained, or does it link to or otherwise rely on external resources (e.g., websites, tweets, other datasets)?** Self-contained.

- **Does the dataset contain data that might be considered confidential (e.g., data that is protected by legal privilege or by doctor–patient confidentiality, data that includes the content of individuals' non-public communications)?** According to the *Leia-me* (readme) file (in Portuguese) that accompanies the dataset and our own inspection, the dataset does not contain any feature that allows direct identification of exam takers such as name, email, ID number, birth date, address, etc. The exam registration number has been substituted by a sequentially generated mask. INEP states that the released data is aligned with the Brazilian *Lei Geral de Proteção dos Dados* (LGPD, General Law for Data Protection). We emphatically recommend the user to view the Readme file prior to usage.

- **Does the dataset contain data that, if viewed directly, might be offensive, insulting, threatening, or might otherwise cause anxiety?** The official terminology used by the Brazilian Government to denote race can be viewed as offensive. Specifically, the term used to describe the race of exam takers of Asian heritage is "Amarela," which is the Portuguese word for the color yellow. Moreover, the term "Pardo," which roughly translates to brown, is used to denote individuals of multiple or mixed ethnicity. This outdated and inappropriate terminology is still in official use by the Brazilian Government, including in its population census. The dataset itself includes integers to denote race, which are mapped to specific categories through the variable dictionary.

- **Does the dataset relate to people?** Yes.

- **Does the dataset identify any subpopulations (e.g., by age, gender)?** Yes. Information about age, sex, and race are included in the dataset.

- **Is it possible to identify individuals (i.e., one or more natural persons), either directly or indirectly (i.e., in combination with other data) from the dataset?** The *Leia-me* (readme) file notes that the individual exam-takers cannot be directly identified from the data. However, in the same file, INEP recognizes that the Brazilian data protection law (LGPD) does not clearly define what constitutes a reasonable effort of de-identification. Thus, INEP adopted a cautious approach: this dataset is a simplified/abbreviated version of the ENEM micro-data compared to prior releases and aims to remove any features that may allow identification of the exam-taker.

- **Does the dataset contain data that might be considered sensitive in any way (e.g., data that reveals racial or ethnic origins, sexual orientations, religious beliefs, political opinions or union memberships, or locations; financial or health data; biometric or genetic data; forms of government identification, such as social security numbers; criminal history)?** The data includes race information and socio-economic questionnaire answers.

**Collection Process**

Since we did not produce the data, we cannot speak directly about the collection process. Our understanding is that the data contains self-reported answers from exam-takers of the ENEM collected at the time of the exam. The exam was applied on 17 and 24 of January 2021 (delayed due to COVID). The data was aggregated and made publicly available by INEP at [INE20]. After consulting the IRB office at our institution, no specific IRB was required to use this data since it is anonymized and publicly available.

**Preprocessing/cleaning/labeling**

- **Was any preprocessing/cleaning/labeling of the data done (e.g., discretization or bucketing, tokenization, part-of-speech tagging, SIFT feature extraction, removal of instances, processing of missing values)?** Some mild pre-processing was done on the data to ensure anonymity, as indicated in the "Leia-me" file. This includes aggregating participant ages, masking exam registration numbers, and removing additional information that could allow de-anonymization.

- **Was the "raw" data saved in addition to the preprocessed/cleaned/labeled data (e.g., to support unanticipated future uses)?** The raw data is not publicly available.

**Uses**

- **Has the dataset been used for any tasks already?** We have used this dataset to benchmark fairness interventions in ML in the present paper. ENEM microdata has also been widely used in studies ranging from public policy in Brazil to item response theory in high school exams.

- **Are there tasks for which the dataset should not be used?** INEP does not clearly define tasks that should not be used on this dataset. However, no attempt should be made to de-anonymize the data.

**Distribution and Maintenance**

The ENEM-2020 dataset is open and made freely available by the Brazilian Government at [INE20] under a Creative Commons Attribution-NoDerivs 3.0 Unported License [Com] at the time of writing. The dataset may become unavailable in case the link in [INE20] becomes inaccessible.