# OpenReview forum: "Beyond Adult and COMPAS: Fair Multi-Class Prediction via Information Projection"
_NeurIPS.cc/2022/Conference — NeurIPS 2022 Accept_

### Official Review · Reviewer_3JtH · 2022-07-09

**Rating:** 8
**Confidence:** 4
**Soundness:** 3 good
**Presentation:** 4 excellent
**Contribution:** 4 excellent

**Summary:**

The authors of this paper present an approach to tabular fairness problems beyond sinlge-class classification problems. The other main contribution of this work is the use of a new tabular dataset with over one million samples on Brazil’s national high school exam. The technical approach the authors take is within the post-processing line of research in the fairness domain where they innovate with an approach to account for multiple protected groups and which is scalable to larger datasets.


**Questions:**

Question: How would the authors believe their work would accommodate notions of intersectionality? Could the authors speak to the increasing work and critical analysis of algorithmic fairness through the lens of intersectionality (https://facctconference.org/static/pdfs_2022/facct22-28.pdf, https://facctconference.org/static/pdfs_2022/facct22-49.pdf, https://facctconference.org/static/pdfs_2022/facct22-122.pdf)

Suggestion: I strongly suggest the authors tone down the language of “introducing” the dataset. According to the appendix, the authors were not involved in the data collection process and they merely link to the data. Suggesting that these authors “introduced” these data is irresponsible. A more honest phrasing would be that the authors “use” the existing dataset.


**Limitations:**

The authors have adequately addressed the limitations and potential negative societal impact of their work.

**Strengths And Weaknesses:**

Strengths:
1. The innovation in the approach and solution for this paper are very high. I believe that this work will have moderate impact in the field and will spur further research on both the dataset and the parallelizable, multi-class approach.
2. The use of the new dataset is much appreciated both for its size and need in this community which for too long has relied on Adult and COMPAS to prove new methods. The experimentation section adequately benchmarks available algorithms against these data as well.
3. The paper includes an extensive theoretical section and convincing experimental results as well.

Weaknesses:
1. The authors could compare to more approaches which are not post-processing style fairness algorithms. However, I don’t believe this is necessary for this paper’s acceptance as I find it to be out of scope of the acceptability of this paper.

---

> ### Author Response · Authors · 2022-08-02
> **Thank you for your thoughtful comments!**
>
> We thank the Reviewer for the thoughtful review and for appreciating the novelty, contribution, and positive prospects of the work!
>
> ---
>
> * **Q1:** ''**The authors could compare to more approaches which are not post-processing style fairness algorithms. However, I don’t believe this is necessary for this paper’s acceptance as I find it to be out of scope of the acceptability of this paper.**''
>
>     --- We thank the Reviewer for appreciating our numerical results. We note that we include comparisons against in-processing methods, such as Reductions [ABD+18] and Adversarial [ZLM18].
>
>
> ---
>
> * **Q2:** ''**How would the authors believe their work would accommodate notions of intersectionality? Could the authors speak to the increasing work and critical analysis of algorithmic fairness through the lens of intersectionality**''
>
>     --- We thank the Reviewer for raising this point. Intersectionality is a critical aspect of fair ML! We will highlight the following comment in Section A.7 of the SM. We note that the matrix $\boldsymbol{G}$ does not require working under a no-intersectionality assumption. Indeed, in our formulation, intersectional groups can be factored into the design of the constraints of the optimization problem by modifying the constraint matrix $\boldsymbol{G}$. Namely, one can consider additional (more refined) groups, thereby increasing the number of rows in $\boldsymbol{G}$ (denoted $K$ in the paper; please see Appendix A.7 for how the scale of $K$ would increase proportionally with respect to an increase in the number of protected groups, denoted $A$). Finally, as we mentioned in Section 6, performance may degrade with intersectional groups.
>
>
> ---
>
> * **Suggestion:** 'using' ENEM rather than 'introducing' ENEM.
>
>    --- We thank the Reviewer for the suggestion and will replace “introduce an open dataset” with “use an open dataset” throughout the paper. We had used the term "introduce" since, to the best of our knowledge, this is the first time that the ENEM dataset has been used for benchmarking fair ML methods, but we fully agree that this choice of wording is too strong (and potentially misleading) and will change it. Thanks!

---

### Official Review · Reviewer_RebA · 2022-07-10

**Rating:** 7
**Confidence:** 3
**Soundness:** 3 good
**Presentation:** 3 good
**Contribution:** 3 good

**Summary:**

The authors tackle the problem of achieving probabilistic group fairness through post-processing. They propose a method based on information projections -- projecting an unfair classifier into the set of fair classifiers, which corresponds to a simple tilting. They propose an algorithm to solve this optimization problem, and derive sample complexity and convergence guarantees. The authors show that their method achieves competitive performance-fairness tradeoffs relative to baseline methods, while having a faster runtime, on both binary and multiclass tasks.

**Questions:**

Please address points 1-6 from above.

**Limitations:**

The authors adequately address most of the limitations in Section 6. I would recommend adding a sentence about how group fairness may not be desirable in the real world because it often leads to reduced utility for all groups [1, 2].

[1] Corbett-Davies, Sam, and Sharad Goel. "The measure and mismeasure of fairness: A critical review of fair machine learning." arXiv preprint arXiv:1808.00023 (2018).

[2] Hu, Lily, and Yiling Chen. "Fair classification and social welfare." Proceedings of the 2020 Conference on Fairness, Accountability, and Transparency. 2020.

**Strengths And Weaknesses:**

Strengths:
The idea of achieving fair classification through information projection is novel to the best of my knowledge. The proposed method is intuitive, and the authors prove many important theoretical properties. It also achieves decent empirical results with low runtime. The paper is rigorous and well-written.

Weaknesses:
1. The method proposed is targeted towards achieving the probabilistic definitions of group fairness, whereas most other work in the algorithmic fairness literature deal with the thresholded (binary) case. The authors should briefly comment on how the two are related -- are there cases where one implies the other?
2. The authors should comment briefly about how they selected their baselines. Why were some methods from Table 1 not used in the experiments, and why were [CJG+19] and others omitted from Table 1?
3. Theorems 2 and 3 from the paper seem to be very similar to Theorem 3 from [AAW+20]. The authors should clarify how they differ.
4. Any performance improvements of the proposed method over [ABD+18] is quite marginal (Fig 1), though there is a significant runtime improvement. How do the convergence or sample complexity guarantees of this method compare with FairProjection? Further, does the convergence rate of FairProjection depend on the number of groups?
5. In Table 3, I would recommend adding in standard deviations, as well as showing these results for the other datasets as well.
6. The multiclass results (Figure 2) are not particularly convincing as there is only one dataset and one baseline method. Are there any simple baselines or additional datasets that can be included?
7. The authors should reference [1] in their related work, and potentially benchmark against this method.
8. I would suggest being more specific in the title about the proposed method.

[1] Kim, Michael P., Amirata Ghorbani, and James Zou. "Multiaccuracy: Black-box post-processing for fairness in classification." Proceedings of the 2019 AAAI/ACM Conference on AI, Ethics, and Society. 2019.

---

> ### Author Response · Authors · 2022-08-02
> **Thank you for your thoughtful comments!**
>
> We thank the Reviewer for the thoughtful review and for appreciating the merits of the work!
>
> ---
>
> * **Q1:** ''**The method proposed is targeted towards achieving the probabilistic definitions of group fairness, whereas most other work in the algorithmic fairness literature deal with the thresholded (binary) case. The authors should briefly comment on how the two are related -- are there cases where one implies the other?**''
>
>    --- To clarify, we consider the probabilistic definition of group fairness since it is more general. This probabilistic definition of group fairness has been used in several other works in the literature [e.g., ABD+18, MW18, CHKV19, WRC20, AAW+20]. When the classifier is a deterministic model (e.g., thresholded case), the probabilistic definition will become the standard definition of group fairness.
>
> ---
>
> * **Q2:** ''**The authors should comment briefly about how they selected their baselines. Why were some methods from Table 1 not used in the experiments, and why were [CJG+19] and others omitted from Table 1?**''
>
>     --- This is a good point. There are four methods in the table we did not include the experiments: FACT [KCT20], Identifying [JN20], FST [WRC20, WRC21], Overlapping [YCK 20].
>     * FACT [KCT20]: We have explained in the appendix why we didn't include the experimental results on FACT (Lines 939--952). We will add a brief explanation in the main text in the final version.
>     * Identifying [JN20]: Their optimization formulation is a special case of our formulation when $f$-divergence is KL divergence, but their algorithm requires retraining a classifier multiple times to solve the optimization problem, which results in a much slower runtime compared to ours (see Lines 1037--1046 in Appendix B.4). Nevertheless, we will add experiments for binary classification using [JN20] in the final version.
>     * FST [WRC20, WRC21]: Codes are not available publicly.
>     * Overlapping [YCK20]: We did not include this method for binary classification experiments as it reduces to the Reductions [ABD+18] approach for the binary class, binary protected group case. We could not benchmark for multi-class experiments with the code available online as it was assuming binary class (even though multiple protected groups).
>
>     Finally, there are related works mentioned in the text but not included in the table ([CJG+19], [ZVRG17], [MW18], [CDPF+17]). For completeness, we will add these to the table in the final version.
>
>
> ---
>
> * **Q3:** ''**Theorems 2 and 3 from the paper seem to be very similar to Theorem 3 from [AAW+20]. The authors should clarify how they differ.**''
>
>     --- We thank the Reviewer for raising this point. While the structures of the statements of our Theorems 2 and 4 are similar to their counterparts in [AAW+20], we highlight the contrast below.
>
>     Our framework has an additional $\ell_2$-regularization term. This regularization allows us to derive a strongly convex dual problem. This strongly convex dual problem, in turn, allows us to derive *convergence guarantees* and ensures the *uniqueness* of the optimal solution. Note that strong convexity is missing in [AAW+20] (see also the discussion in Lines 173--178).
>
>     Our results cannot be derived as corollaries from those in [AAW+20], as our $\ell_2$-regularized $f$-divergence primal problem (eq. (6)) does not conform to the setup in [AAW+20]. In addition, the proof techniques for our theorems deviate from those in [AAW+20]; for instance, we use a more advanced minimax theorem in Appendix A.1 to derive our Theorem 2, which is inapplicable in the setup of [AAW+20] (we highlight this point in more detail in Remark 3 in Appendix A.1).
>
>     Finally, we introduce a new scalable algorithm to solve fair projection and establish its convergence guarantees in Theorem 3. In contrast, [AAW+20] does not provide any algorithm for implementing information projection, let alone convergence guarantees.
>
>
> ---
>
> **(continued in next comment ...)**

---

> > ### Author Response · Authors · 2022-08-02
> > **Thank you for your thoughtful comments! (continued)**
> >
> > * **Q4:** ''**Any performance improvements of the proposed method over [ABD+18] is quite marginal (Fig 1), though there is a significant runtime improvement. How do the convergence or sample complexity guarantees of this method compare with FairProjection? Further, does the convergence rate of FairProjection depend on the number of groups?**''
> >
> >    --- These are great points! We compare the runtimes here, and further below we describe the dependence on the number of groups. The theoretical analysis below contrasting the runtimes of both FairProjection and [ABD+18], which is in line with our numerically observed comparison in Table 3 in the submitted manuscript, will be added as a remark to the SM. Two key factors make FairProjection faster than [ABD+18]:
> >
> >     **a)** FairProjection needs a much lower number of iterations than [ABD+18] does (logarithmic vs. polynomial), and
> >
> >     **b)** Each iteration for FairProjection is less computationally expensive than its counterpart in [ABD+18]. In fact, it is independent of the underlying model being projected, whereas [ABD+18] requires retraining.
> >
> >     In more detail, one can obtain from [Theorem 3, ABD+18] that the Reductions approach converges in $O(N^2)$ iterations (where $N$ is the number of samples and we use the suggested $\alpha=1/2$ in [Theorem 3, ABD+18] according to the discussion at the top of page 6 therein). Taking the runtime of each iteration into consideration, one cannot hope for a runtime faster than $O(N^4)$ for [ABD+18]. In fact, the runtime for [ABD+18] must be higher than $O(N^4)$, since each of its iterations performs the subroutine $\text{BEST}_{\boldsymbol{h}}(\boldsymbol{\lambda})$, which is a 'cost-sensitive classification' problem (i.e., numerically solving for an optimal classifier), and the $O(N^4)$ estimate would hold only if this *retraining* procedure can be done in *constant time* (which might be overly optimistic). In contrast, FairProjection does not require this retraining procedure at all, runs in $O(\log N)$ iterations, has $O(N)$ runtime for each iteration, and can perform much of each iteration in a parallel way. This theoretical comparison is in line with the numerical comparison presented in Table 3 in our paper, and we will add this discussion to the SM of the final version.
> >
> >     --- For the dependence of the runtime of FairProjection on the number of groups, we note that there is a *linear* dependence on the number of constraints $K$ when the number of samples $N$ is much larger than $K$ (which is the case for all datasets we consider), so one can say that the runtime is at most $\gamma K N \log N$ for an *absolute* constant $\gamma$. Note that, as in Lines 119--123, there are $K=2AC$ constraints for statistical parity, where $A$ is the number of sensitive groups, and $C$ is the number of classes; e.g., for the ENEM-1.4M-2C dataset that is used in Table 3, we get $K=8$ for statistical parity. The $K$ factor in the $O(KN\log N)$ rate comes from the creation of the vector $\boldsymbol{q}$ in Algorithm 1. If one does not parallelize, still one gets a runtime of $O(CKN\\log N)$. Interestingly, the $\boldsymbol{v}_i$-update step runtime in Algorithm 1 is $O(C)$ for a fixed $i\in [N]$ for both KL-divergence and Cross Entropy (see Appendices A.3 and A.4).
> >
> > ---
> >
> > * **Q5:** ''**In Table 3, I would recommend adding in standard deviations, as well as showing these results for the other datasets as well.**''
> >
> >     --- Thank you for the suggestion! We will add the runtimes for all of the other datasets in the final version. The reason why we chose to show ENEM-1.4M is that our goal was to demonstrate the scalability of the proposed FairProjection algorithm, and thus we used the biggest dataset we have. The following table shows the runtime results for the (binary) HSLS dataset, which is the second biggest dataset of the four datasets we used:
> >
> >   |    **Method**   | **Reduction** [ABD+18] | **Rejection** [KKZ12] | **EqOdds** [HPS16] | **LevEqOpp** [CDH+19] | **CalEqOdds** [PRW+17] | **FairProjection** (ours) |
> >
> >   | **Runtime** (s) |       81.1 ± 11.7      |      9.73 ± 0.15      |    2.28 ± 0.005    |   2.00 ± 0.00    |      2.29 ± 0.006      |        8.23 ± 0.59        |
> >
> > ---
> >
> > ** (continued below ...) **

---

> > > ### Author Response · Authors · 2022-08-02
> > > **Thank you for your thoughtful comments! (continued)**
> > >
> > > * **Q6:** ''**The multiclass results (Figure 2) are not particularly convincing as there is only one dataset and one baseline method. Are there any simple baselines or additional datasets that can be included?**''
> > >
> > >     --- Thank you for raising this concern. We have searched long and hard for baseline methods for multi-class prediction. Unfortunately, we did not find many methods that: 1) can be easily compared against FairProjection, and 2) use the same notions of fairness (e.g., mean equalized odds, statistical parity). We have included a literature review on multi-class fair prediction in Appendix B.4.1. None of the works we found has provided any empirical result of multi-class prediction on real-world datasets for multi-class. We modified the code for the Adversarial approach [ZLM18] to make it work for multi-class. We hope to see more authors update their code to work for general multi-class.
> > >
> > >
> > >    --- Following the suggestions by you and Reviewer 5rHq, we have conducted additional multi-class experiments on the HSLS dataset. We divided student math performance into quartiles and generated four classes. We show the results in the table below and we will include a figure in the final version of the paper. We note that if we plot the points below, we obtain a figure that looks very similar to the multi-class ENEM results shown in the paper and SM where Adversarial can slightly reduce MEO, but cannot reduce MEO as substantially as FairProjection.
> > >
> > >
> > > |  | **Achievable (Accuracy Difference, MEO)** |
> > > |:---:|:---:|
> > > | **FairProjection (CE)** | (0, 0.185), (-0.001, 0.057), (-0.002, 0.025) |
> > > | **Adversarial** | (0, 0.187), (-0.0006, 0.135), (-0.001, 0.131) |
> > >
> > >
> > >
> > > --- Looking ahead, as a community, it is imperative to establish multi-class fairness datasets that could be benchmarked and also publish codes that could be applied for multi-class prediction tasks. We believe that our paper makes a significant contribution towards this.
> > >
> > > ---
> > >
> > > - **Suggestion:** ''**The authors should reference [1] in their related work, and potentially benchmark against this method.**''
> > >
> > >     --- Thank you for bringing [1] to our attention, and we will include it in the related work. We find that the goal of [1] is very different from ours. They propose a new fairness notion called "multiaccuracy," which is different from the parity-based fairness notions we focus on in our paper. Its aim is to ensure high accuracy for all subgroups even when the group information is not given in the data, and improve the performance of the underperforming subgroups. Nevertheless, the multiaccuracy constraints can be incorporated into our optimization framework. Utilizing the FairProjection algorithm for multiaccuracy would be an interesting future direction. We will add this discussion in the final version of the paper.
> > >
> > >     --- [1] Kim, Michael P., Amirata Ghorbani, and James Zou. "Multiaccuracy: Black-box post-processing for fairness in classification." Proceedings of the 2019 AAAI/ACM Conference on AI, Ethics, and Society. 2019.
> > >
> > >
> > > ---
> > >
> > > * **Suggestion:** ''**I would suggest being more specific in the title about the proposed method.**''
> > >
> > >    --- We thank the Reviewer for the suggestion, and will probably change the title to: ''Beyond Adult and COMPAS: Fairness in Multi-Class Prediction via Information Projection.''
> > >
> > > ---
> > >
> > > * **Suggestion:** ''**The authors adequately address most of the limitations in Section 6. I would recommend adding a sentence about how group fairness may not be desirable in the real world because it often leads to reduced utility for all groups [1, 2].**''
> > >
> > >    --- We thank the Reviewer for raising this concern. We agree that the pros and cons of group fairness are highly debated nowadays and the choice of fairness metrics (either group fairness or individual fairness or other notions) should rely on the application of interest. We will include the missing references and the limitation of group fairness in Section 6.

---

> > > > ### Comment · Reviewer_RebA · 2022-08-05
> > > > **Thanks for the response!**
> > > >
> > > > I thank the authors for the detailed response. All of my concerns have been addressed.

---

> > > > > ### Author Response · Authors · 2022-08-05
> > > > > **Glad that all your concerns have been addressed!**
> > > > >
> > > > > Thank you for your positive feedback, we are glad that we addressed all of your concerns. We would sincerely appreciate it if you would kindly consider increasing your score. Thanks, and have a great weekend!

---

### Official Review · Reviewer_4NMe · 2022-07-11

**Rating:** 8
**Confidence:** 3
**Soundness:** 4 excellent
**Presentation:** 3 good
**Contribution:** 4 excellent

**Summary:**

This paper introduces a procedure for improving the fairness of models in the multi-class classification setting using a technique called information projection. In practice, this technique involves reweighting the outputs of a pre-trained model. The authors provide a theoretical basis for the technique and empirically evaluate on multiple multi-class benchmark datasets. Additionally, they introduce a new, larger scale dataset of Brazilian high school assessments (1M examples) that is to be released with the paper.

**Questions:**

The questions I list here are ones that I think, if the authors were to clarify in the text, would strengthen the paper overall.

What kinds of fairness constraints are not expressible in the form shown on line 119? That is to say, is the work limited to the criteria in table 2, and if not, what are the limits?

In figure 1, there are multiple points for each technique? It seems from line 280-281 that this is because of varying alpha. If so, please clarify in the caption of the figure. Additionally, it would be nice to indicate which direction changing alpha corresponds to moving on the curves.


**Limitations:**

The key limitation is the expressibility of the fairness constraint as an inequality. It would be good to get more clarity on which fairness metrics satisfy this. Additionally, the technique is limited to datasets where group membership is known. It would be interesting to know from the authors if there are any worst-case guarantees that can be made if the groups are only partially known, but this is more for interest rather than a necessary component of the paper.

**Strengths And Weaknesses:**

Originality: The techniques introduced are an application of the information projection technique for the purposes of improving fairness of multi-class prediction systems. This work is original in multiple senses. First, it goes beyond much of the literature just by considering the multi-class setting as opposed to the typical binary setting. It introduces a new reweighting technique that is competitive with other existing techniques while generally scaling better for large datasets. In addition to the novelty of the technique itself, I found that the focus on a more complex setting and the addition of practical considerations like the runtime of the technique distinguish this work from others. Additionally, it introduces and releases a completely new benchmark dataset for the community.

Quality: While I did not verify all of the proofs in the work, the presentation is high quality and claims are rigorously justified throughout.

Clarity: In general, the presentation was very clear. It was easy to get lost in the math in the middle of the paper, and I suggest that the authors move some of this material into the supplemental material. In particular, some of the introductory material on pages 3 and 4 on information projection could be moved, as well as the details of the theoretical guarantees of FairProjection on page 7. This will make the paper more easily digestible, and they are not necessary for understanding the empirical results shown later.

Significance: The results shown demonstrate improvement over existing techniques, both in the scope of their applicability (some of those compared to are only applicable in the binary case) and runtime constraints. With the release of code and data, it does seem that this work would be very usable by practitioners who work in the multi-class classification setting. Additionally, the newly released dataset should allow many new papers to benchmark against a tougher problem than currently exists in the literature.

---

> ### Author Response · Authors · 2022-08-02
> **Thank you for your thoughtful comments!**
>
> We thank the Reviewer for the thoughtful review and for appreciating the novelty, complexity, significance, and positive prospects of the work!
>
> ---
>
> * **Q1:** ''**What kinds of fairness constraints are not expressible in the form shown on line 119? That is to say, is the work limited to the criteria in table 2, and if not, what are the limits?**''
>
>    --- Although we only provide three fairness measures in Table 2, there exist many standard group fairness measures that can be written in a linear form. Some examples include false positive/negative rate difference, mean score parity, and conditional statistical parity (see, e.g., [WRC20] for definitions). Our framework can also be applied to convex combinations of these group-fairness measures. Finally, we remark that there exist some fairness metrics, such as calibration error, that cannot be written in linear form. More generally, metrics that require conditioning on the classifier's output (such as calibration) cannot be expressed as a linear constraint. We will clarify this point in our last section.
>
> ---
>
> * **Q2:** ''**In figure 1, there are multiple points for each technique? It seems from line 280-281 that this is because of varying alpha. If so, please clarify in the caption of the figure. Additionally, it would be nice to indicate which direction changing alpha corresponds to moving on the curves.**''
>
>    --- The techniques *EqOdds* [HPS16], *CalEqOdds* [PRW+17], and *LevEqOpp* [CDH+19] only produce a single accuracy-fairness trade-off point, whereas the rest of the methods are capable of producing the accuracy-fairness trade-off *curves* by varying the fairness budget $\alpha$ for the group fairness criteria listed in Table 2.  We will update the curves in Figure 1 with a right arrow indicating the direction of increasing $\alpha$, and improve the caption by mentioning the choices of $\alpha$ in Table 2.
>
>
> ---
>
> * **Suggestion:** On moving some of the math to the SM.
>
>    --- We thank the Reviewer for the suggestion on the organization of this paper. We will move superfluous technical details to the appendix and streamline the presentation of the first pages.
>
> ---
>
> * **Suggestion:** ''**The technique is limited to datasets where group membership is known. It would be interesting to know from the authors if there are any worst-case guarantees that can be made if the groups are only partially known, but this is more for interest rather than a necessary component of the paper.**''
>
>    --- This is a great question! We agree that missing group membership is a critical problem in practice. We remark that our framework only requires a classifier that infers the group membership from the other features. It is possible to train such a classifier by using auxiliary data even if the primary dataset does not have complete information about group membership. This point was originally included in Lines 123--126 and further highlighted in Section A.7 of the SM, and we will elaborate on this aspect in the main text.

---

> > ### Comment · Reviewer_4NMe · 2022-08-03
> > **Thank you!**
> >
> > Thank you for your thoughtful responses to the review. The explanations you provided make sense, and I appreciate the changes you suggest making to the text. I think they will clarify the questions I asked.

---

> > > ### Author Response · Authors · 2022-08-09
> > > **Thank you!**
> > >
> > > Thank you for your positive feedback, we are glad that we addressed all of your questions. Thanks again, and have a great rest of your day!

---

### Official Review · Reviewer_5rHq · 2022-07-11

**Rating:** 7
**Confidence:** 3
**Soundness:** 3 good
**Presentation:** 3 good
**Contribution:** 3 good

**Summary:**

This paper proposes a post-processing method to generate fair multi-class classification models based on the information projection technique which formulates fairness as a linear projection-based constraint. Theoretical guarantees on the uniqueness and approximation quality of the optimal solution are derived, and experimental results on real-world data sets demonstrate effectiveness of the proposed method in both binary and multi-class classification tasks.



**Questions:**

1. Please clarify the definition of multi-class fairness.

2. Please clarify how the formulated linear projection-based constraint achieves that definition.

3. Please try to enrich the experimental results for the multi-class case.

I will raise my score if the above questions are adequately addressed.

**Limitations:**

Yes.

**Strengths And Weaknesses:**

Strength
- Fair multi-class classification is a timely and useful topic.
- The proposed method looks novel and has theoretical guarantees.
- The paper is well-written in general.

Weakness
- It is not explained how "multi-class fairness" is defined and how the formulated linear projection-based constraint achieves that.
- A large portion of the experimental results focus on binary classification instead of multi-class classification.


Detailed Comments

Overall it looks like an acceptable paper. The idea of using information projection to model fairness looks novel and interesting, and theoretical guarantees are derived.

My main complaint is on the missing explanations of two things: (1) the definition of multi-class fairness, (2) how the formulated linear projection-based constraint achieved that (it is not explained in the supplementary material either). Please clarify them.

Experimental results look a bit weak, because a large part of them focuses on the binary case and only Figure 2 is for multi-class classification. (Same pattern in the supplementary material.) I suggest authors expand experimental results on the multi-class case to support the main contribution of this paper.

Also, there are two existing fair multi-class classifiers in Table 1. Why only one of them is being compared in experiment?

Finally, it might be helpful to discuss the fairness of h_base. My understanding is Theorem 4 suggests the trained model can well approximate the most accurate model h_base (probably to the optimum degree). If h_base is far from fairness, would the trained model still be fair? Or, would there be any conflict between this approximation and the formulated constraint?

---

> ### Author Response · Authors · 2022-08-02
> **Thank you for your thoughtful comments!**
>
> We thank the Reviewer for the thoughtful comments and for appreciating the novelty and value of the work. We hope that the answers below address the points raised in the review.
>
> ---
>
> * **Q1--2:** ''**Please clarify the definition of multi-class fairness**,'' and ''**Please clarify how the formulated linear projection-based constraint achieves that definition**.''
>
>     --- The definition of multi-class fairness metrics is given in Table 2 and developed in detail in Section A.7 of the SM. These measures are natural multi-class generalizations of group fairness metrics designed for binary classification (e.g., statistical parity, equalized odds, and error rate imbalance) in the following sense: if we restrict the output of the classifier to be thresholded binary decisions (i.e., 0 or 1 outputs), these metrics reduce to their binary counterparts. In Section A.7, we prove that these measures can be written in a linear form. We also include their corresponding $G$ matrix in the same section. To avoid this confusion, we will change the caption of Table 2 to “**Standard multi-class group fairness criteria**” and the title of Section A.7 to **“Linearized multi-class group fairness criteria.”**
>
> ---
>
> * **Q3:** ''**Please try to enrich the experimental results for the multi-class case.’’** (**’’I suggest authors expand experimental results on the multi-class case to support the main contribution of this paper,’’** and **‘’Also, there are two existing fair multi-class classifiers in Table 1. Why only one of them is being compared in experiment?’’**).
>
>     --- We have been searching baseline methods for multi-class prediction, and have not found methods that use similar notions of fairness (e.g., MEO and SP). We included a literature review on methods for multi-class fair predictions in Appendix B.4.1, and explained the reason why there is a lack of benchmarks for the multi-class case—none of the works we found has provided any empirical result of multi-class prediction on real-world datasets for multi-class. We adapted the codes of the Adversarial approach [ZLM18] for the multi-class case, and compared it with FairProjection on the multi-class HSLS dataset (students’ math scores are quantized into four classes). This additional experiment is provided in the table below, and will be included in future revisions. In summary, Adversarial can slightly reduce MEO, but cannot reduce MEO as substantially as FairProjection.
>
>
> |  | **Achievable (Accuracy Difference, MEO)** |
> |:---:|:---:|
> | **FairProjection (CE)** | (0, 0.185), (-0.001, 0.057), (-0.002, 0.025) |
> | **Adversarial** | (0, 0.187), (-0.0006, 0.135), (-0.001, 0.131) |
>
>
>
>
> --- On Overlapping [YCK20]: We did not include this method for binary classification experiments as it reduces to the Reductions [ABD+18] approach for the binary class, binary protected group case. We could not benchmark for multi-class experiments with the code available online as it was assuming binary class (even though multiple protected groups).
>
> ---
>
> * **Suggestion:** ''**Finally, it might be helpful to discuss the fairness of $\boldsymbol{h}^{\mathrm{base}}$. My understanding is Theorem 4 suggests the trained model can well approximate the most accurate model $\boldsymbol{h}^{\mathrm{base}}$ (probably to the optimum degree). If $\boldsymbol{h}^{\mathrm{base}}$ is far from fairness, would the trained model still be fair? Or, would there be any conflict between this approximation and the formulated constraint?**''
>
>    --- Even if $\boldsymbol{h}^{\mathrm{base}}$ (or its approximation) is far from fair or inaccurate, the projected model will satisfy the desired group fairness metrics. This is guaranteed by the constraints in our optimization problem. In fact, our goal is to post-process (the potentially unfair) $\boldsymbol{h}^{\mathrm{base}}$ so that it will become fair while maintaining the closest scores as measured by the $f$-divergence of choice. The only requirement of $\boldsymbol{h}^{\mathrm{base}}$ is that it has a good accuracy since the performance of $\boldsymbol{h}^{\mathrm{base}}$ will influence the accuracy of the projected model. Please also refer to our last section (Final remarks and limitations) for further discussion.

---

> > ### Comment · Reviewer_5rHq · 2022-08-04
> > **Thank you for the responses.**
> >
> > I appreciate the authors' responses.
> >
> > My question on the multi-class fairness definition is clarified, although I suggest a more straightforward statement e.g., "We give three multi-class group fairness definitions in Table 2.". I wouldn't worry about potential critiques on the simplicity of such extension, as long as the paper can clarify how the downstream learning problem or analysis is non-trivial. Also, I suggest remove "standard" in the updated caption of Table 2, and replace "iterate over" with more straightforward statement such as the inequality holds "for all (a,c,c')".
> >
> > My question on how the linear constraint achieves fairness is not fully addressed, though. I read Supplementary A.7 during the initial review, but they are not clear to me -- not sure if they are clear to other reviewers... I assume that explanation was compressed partly because the connection is already established in prior work? In that case, for clarity, it may be better to first present the established linear constraint (for binary case only, I assume) and then explain how this paper extends it to multi-class case. For example, is the previous "G" a K-by-2 matrix where each row is associated with one class? And this paper naturally extends it to K-by-C?
> >
> > For experimental results, I share the same concern with reviewer RebA on some insufficiency of multi-class results. I see authors have promised to add a few more baselines which is great. Also, what about this baseline: we apply one-versus-one strategy to decompose the multi-class task to several binary subtasks and apply existing binary fairness method on each subtask. (Not necessary to include them, but I suppose this is the first question that strikes most people when looking into multi-class problem.)
> >
> > Finally, my question on the fairness of h_base is not fully addressed. I understand the algorithm requires h to be fair, and I read the last section that claims h_base can be potentially unfair. But these do not address my question. Allow me to be more specific this time: Assuming my understanding on Theorem 4 is correct i.e., the post-processed model "h" approaches "h_base" as training set increases (correct me if I'm wrong), then its implication seems unreasonable if h_base is unfair but h is required to be fair -- there has to be a lower bound on the distance between the two models. From the current presentation, the only term preventing "h" from approaching "h_base" arbitrarily close seems to be D* -- but from its definition in (5), I do not see how the conflict is characterized. It would be nice to have some discussion or clarification.

---

> > > ### Author Response · Authors · 2022-08-05
> > > **Thank you for the followup questions!**
> > >
> > > Thank you for your constructive follow-up! We hope our replies below address your remaining points! We also really appreciate your time and kindness in engaging with us during the discussion period.
> > >
> > > **(1) My question on the multi-class fairness definition is clarified, although I suggest a more straightforward statement e.g., "We give three multi-class group fairness definitions in Table 2.". I wouldn't worry about potential critiques on the simplicity of such extension, as long as the paper can clarify how the downstream learning problem or analysis is non-trivial. Also, I suggest remove "standard" in the updated caption of Table 2, and replace "iterate over" with more straightforward statement such as the inequality holds "for all (a,c,c')".**
> > >
> > > We are glad that your question is clarified! Thanks for the additional suggestions. We will incorporate them in the final version.
> > >
> > > ---
> > >
> > > **(2) My question on how the linear constraint achieves fairness is not fully addressed, though. I read Supplementary A.7 during the initial review, but they are not clear to me -- not sure if they are clear to other reviewers... I assume that explanation was compressed partly because the connection is already established in prior work? In that case, for clarity, it may be better to first present the established linear constraint (for binary case only, I assume) and then explain how this paper extends it to multi-class case. For example, is the previous "G" a K-by-2 matrix where each row is associated with one class? And this paper naturally extends it to K-by-C?**
> > >
> > > We appreciate the suggestion on the presentation of this section, and we will incorporate it in the final version. You are right: existing literature [see, e.g., ABD+18, MW18, CHKV19, WRC20, AAW+20a] have already proved that the binary counterpart of the fairness constraints in Table 2 can be written as linear functions (see Line 115 in our paper for details). Below (at the end of this response thread) we include a derivation for the linearization of statistical parity, and will expand Appendix A.7 to include details on linearization for all of the other fairness metrics in Table 2.
> > >
> > > ---
> > >
> > > **(3) For experimental results, I share the same concern with reviewer RebA on some insufficiency of multi-class results. I see authors have promised to add a few more baselines which is great. Also, what about this baseline: we apply one-versus-one strategy to decompose the multi-class task to several binary subtasks and apply existing binary fairness method on each subtask. (Not necessary to include them, but I suppose this is the first question that strikes most people when looking into multi-class problem.)**
> > >
> > > We thank the reviewer for the suggested baseline—great idea! The reason we did not try one-v.s.-one combined with existing binary fairness methods is that, for metrics such as (mean) equalized odds, it is not clear if the combined multi-class classifier satisfies the desired fairness constraints even if each binary classifier is fair. The technical reason behind this observation is: pairwise independence does not imply mutual independence. Thus, ensuring fairness for pairs of decisions may not imply fairness across multiple classes. In contrast, FairProjection ensures fairness across multiple classes by design.
> > >
> > > As mentioned in our response to **Q6** for Reviewer ‘RebA’ regarding how we select baselines for multi-class experiments, we have searched long and hard for baseline methods for multi-class prediction. Unfortunately, we did not find methods that: 1) can be easily compared against FairProjection (e.g., have available codes for multi-class), and 2) use the multi-class extensions of mean equalized odds and statistical parity.
> > >
> > > We have included a literature review on multi-class fair prediction in Appendix B.4.1. None of the works we found has provided any empirical result of multi-class prediction on real-world datasets for multi-class. In fact, we modified the codes for the Adversarial approach [ZLM18] ourselves to make it work for multi-class, which took a significant effort. We hope to see more authors update their code to work for general multi-class prediction, since non-binary classification is quite common in ML practice. We believe that one reason for lack of readily available fairness interventions in multi-class classification is the lack of appropriate benchmarks—which is exactly what led us to use the ENEM dataset.
> > >
> > > Please also note that we have added an additional benchmark against the Adversarial approach on the **multi-class (4 classes) HSLS dataset** (see table in our initial response).
> > >
> > > **Continued...**

---

> > > > ### Author Response · Authors · 2022-08-05
> > > > **Thank you for the followup questions! - Continued**
> > > >
> > > > **(4) Finally, my question on the fairness of h_base is not fully addressed. I understand the algorithm requires h to be fair, and I read the last section that claims h_base can be potentially unfair. But these do not address my question. Allow me to be more specific this time: Assuming my understanding on Theorem 4 is correct i.e., the post-processed model "h" approaches "h_base" as training set increases (correct me if I'm wrong), then its implication seems unreasonable if h_base is unfair but h is required to be fair -- there has to be a lower bound on the distance between the two models. From the current presentation, the only term preventing "h" from approaching "h_base" arbitrarily close seems to be $D^\*$ -- but from its definition in (5), I do not see how the conflict is characterized. It would be nice to have some discussion or clarification.**
> > > >
> > > > Thanks for following up on this point! We first recall the setup of this paper: given a pre-trained unfair classifier $h^{\mathrm{base}}$, we project this classifier onto the set of all classifiers that satisfy desired fairness constraints. This boils down to finding the "closest" classifier to  $h^{\mathrm{base}}$ that satisfies the constraints, where "closeness" is measured by an $f$-divergence of choice. The projected classifier is denoted by $h$. Thus, $h$ will be the fair classifier that is "closest" to $h^{\mathrm{base}}$, since $h$ is necessarily a member of the set of classifiers that satisfy the target fairness constraints—even when $h^{\mathrm{base}}$ is unfair!
> > > >
> > > > An imperfect (but hopefully helpful) analogy is that of projecting a vector $\mathbf{x}$ onto a a closed and convex set $\mathcal{C}$ defined by a set of constraints. If $\mathbf{x}\notin \mathcal{C}$, then its projection (let's call it $\mathbf{x}_P$) is the closest point in $\mathcal{C}$ to $\mathbf{x}$. Note that $\mathbf{x}_P\in\mathcal{C}$, thus satisfying the constraints that define $\mathcal{C}$. If $\mathbf{x}\in \mathcal{C}$, then its projection onto $\mathcal{C}$ will be itself!
> > > >
> > > > In our case, instead of projecting vectors, we are projecting classifiers, and the set is defined by fairness constraints. Consequently, the projected classifier is—in theory—fair, and $h$ will be the "closest" fair classifier (again, in an $f$-divergence sense) to $h^{\mathrm{base}}$. Similar to the vector space analogy, if $h^{\mathrm{base}}$ is unfair, there will always be a gap between $h$ and $h^{\mathrm{base}}$, captured by $D^*$ in Theorem 4. If $h^{\mathrm{base}}$ is already fair in terms of the group fairness constraints with a preset $\alpha$, then the projection $h$ will be $h^{\mathrm{base}}$ itself.
> > > >
> > > > To your question, Theorem 4 does **not** prove that $h$ approaches $h^{\mathrm{base}}$ as the training set increases. In fact, as you noticed, there is always a gap $D^*$ that prevents $h$ from approaching $h^{\mathrm{base}}$ when $h^{\mathrm{base}}$ is unfair, i.e., does not satisfy the fairness constraints. Intuitively, this gap describes the least amount of change needed for $h^{\mathrm{base}}$ to make it ‘’fair.’’
> > > >
> > > > The goal of Theorem 4 is to establish the statistical consistency of Algorithm 1: the solution of Algorithm 1 asymptotically (in the number of samples) achieves the lowest possible error of the dual optimization (and the theorem quantifies this convergence rate). Intuitively, Theorem 4 captures the rate at which the value of the "projection" optimization computed over finite samples converges to the optimal solution if the underlying data distribution was known exactly. This does not mean that $h$ converges to $h^{\mathrm{base}}$: in fact, they will be separated by a gap $D^*$ (in $f$-divergence), as mentioned above.
> > > >
> > > > In the final version, we will add a remark after Theorem 4 summarizing the discussion above and elucidating the important point you raised: the theorem does not imply that $h$ converges to $h^{\mathrm{base}}$.
> > > >
> > > > Thank you again! We would be happy to answer any additional follow-up questions.

---

> > > > > ### Author Response · Authors · 2022-08-05
> > > > > **A detailed derivation of linearization of statistical parity**
> > > > >
> > > > > Recall that a prediction $\hat{Y}$ satisfies statistical parity (SP) if it is independent of the group attribute $S$, i.e., $P_{\hat{Y}|S=a}(c’)=P_{\hat{Y}}(c’)$ for every $(a,c’) \in [A]\times [C]$. A relaxed notion of SP is ‘approximate independence’ of $\hat{Y}$ and $S$: $|P_{\hat{Y}|S=a}(c’)/P_{\hat{Y}}(c’)-1|\le \alpha$ for some small $\alpha \ge 0$ and all $(a,c’) \in [A]\times [C]$. Using $P_{\hat{Y}|S}=P_{\hat{Y},S}/P_S$ and rearranging, the above inequality is equivalent to
> > > > > $$\pm P_{\hat{Y},S}(c’,a)/P_S(a)-(\alpha \pm 1)P_{\hat{Y}}(c') \le 0.$$
> > > > > We expand via conditioning $\hat{Y}$ on $X$, and $S$ on $(X,Y)$. Recall that $h_{c’}(X) = P_{\hat{Y}|X=x}(c’)$ and $s_a(X,c)=P_{S|X=x,Y=c}(a)$ by definition, and that we have a Markov chain $(Y,S)-X-\hat{Y}$; hence, $P\_{\hat{Y}}(c') = \mathbb{E}\_X[h\_{c'}(X)]$ and $P\_{\hat{Y},S}(c',a) = \mathbb{E}\_X[\sum\_{c\in [C]} s\_a(X,c)h\_c^{\mathrm{base}}(X)h\_{c’}(X)]$.
> > > > > Thus, we can write approximate SP as
> > > > > $$\mathbb{E}\_X\left[\left(\pm P\_S(a)^{-1}\sum\_{c\in [C]} s\_a(X,c) h\_{c}^{\mathrm{base}}(X) - (\alpha\pm 1) \right) h\_{c'}(X) \right] \le 0.$$
> > > > > We denote $\boldsymbol{h}(X) = [h\_1(X), h\_2(X), \cdots, h\_C(X)]^\top$, and for $(\delta,a,c')\in ${0, 1}$\times [A]\times [C]$, denote
> > > > > $$
> > > > > \boldsymbol{g}^{(\delta,a,c')}(X) := \left( (-1)^\delta P\_S(a)^{-1}\sum\_{c\in [C]} s\_a(X,c) h\_{c}^{\mathrm{base}}(X) - (\alpha + (-1)^{\delta}) \right) \mathbf{e}\_{c'},
> > > > > $$
> > > > > where $\mathbf{e}\_{c'}$, $c'=1, \cdots, C$ are the standard bases in $\mathbb{R}^C$.
> > > > > Then, for each pair $(\delta, a, c') \in $ {0, 1} $\times[A]\times [C]$, we have a linear constraint $\mathbb{E}\_X[\boldsymbol{g}^{(\delta,a,c')}(X)^T\boldsymbol{h}(X)] \le \mathbf{0}$.
> > > > > Since there are $K = 2AC$ possible triplets $(\delta, a, c)$, we convert the SP constraint into $K$ linear constraints.

---

> > > > > > ### Comment · Reviewer_5rHq · 2022-08-05
> > > > > > **Thanks for the elaboration.**
> > > > > >
> > > > > > Thanks for the elaboration. It is now clear to me how the linear constraint is constructed.

---

> > > > > ### Comment · Reviewer_5rHq · 2022-08-05
> > > > > **Thanks for the detailed explanation.**
> > > > >
> > > > > Thanks for the detailed explanation.
> > > > >
> > > > > The implication of Theorem 4 is more clear to me now.
> > > > >
> > > > > I like the explanation that "Intuitively, this gap describes the least amount of change needed for  to make it ‘’fair.’’". In revision, it would be nice to elaborate on this interpretation and connect it to the definition of D* in formula (5).
> > > > >
> > > > > All my concerns have been addressed.

---

> > > > > > ### Author Response · Authors · 2022-08-05
> > > > > > **We are glad that all your concerns have been addressed!**
> > > > > >
> > > > > > Thank you for your positive feedback, we are glad that we addressed all of your concerns. We would sincerely appreciate it if you would kindly consider increasing your score. Thanks, and have a great weekend!

---

> > > > ### Comment · Reviewer_5rHq · 2022-08-05
> > > > **Thank you for the responses.**
> > > >
> > > > Thank you for the responses.
> > > >
> > > > My question on the linear constraint is fully addressed.
> > > >
> > > > For experiment, as the authors noticed, it is "not clear" if the 1v1 baseline would work. This means nobody has disproved it yet, right? :)  In fact, if authors could show (the first) empirical evidence that disproves this baseline, the motivation of this paper will be significantly strengthened -- that multi-class fairness is not as trivial to achieve as we think. That said, this should not be a necessary addition for the acceptance of the paper.

---

### Author Response · Authors · 2022-08-02
**Thank you all for your thoughtful comments!**

We thank all Reviewers for their time and effort! We are glad to see our paper was positively received, and the Reviewers found that: the paper is novel, rigorous, and well-written (all Reviewers); the topic of multi-class fairness is interesting (Reviewers 5rHq, 4NMe, 3JtH); FairProjection scales better for larger datasets (Reviewers 4NMe, RebA, 3JtH); the use of the new dataset, ENEM, is valuable (Reviewers 4NMe, 3JtH); and our work will spur further fair ML research under a more complex setting (Reviewers 4NMe, 3JtH). We also recognize that the Reviewers are busy handling multiple papers, so their thoughtful feedback is even more appreciated. Below, we address the main points and questions raised by each Reviewer and outline how we plan to update the paper accordingly; we will add the changes in the final, longer version (both in the main text and SM).

Please do follow up with us if you have additional suggestions and feedback that can further strengthen the paper. Thanks!

---

### Meta-Review · Area_Chair_DdB6 · 2022-08-21

**Recommendation:** Accept
**Confidence:** Certain

**Metareview:**

I recommend acceptance for this paper due to the uniformly positive reviews, which emphasize the relevance and novelty of the proposed method for fair multiclass classification. The method is well-supported experimentally, and the authors introduce to the ML literature a new benchmark classification task. During the discussion period, many of the reviewers concerns around baselines, related work, etc were addressed.

**Award:**

No

---

### Decision · Program_Chairs · 2022-09-14

Accept